# SELFREFLECT: CAN LLMS COMMUNICATE THEIR INTERNAL ANSWER DISTRIBUTION?

**Michael Kirchhof**
Apple

**Luca Füger**
Independent Researcher

**Adam Goliński**
Apple

**Eeshan Gunesh Dhekane**
Apple

**Arno Blaas**
Apple

**Seong Joon Oh**
Tübingen AI Center

**Sinead Williamson**
Apple

## ABSTRACT

The common approach to communicate a large language model's (LLM) uncertainty is to add a percentage number or a hedging word to its response. But is this all we can do? Instead of generating a single answer and then hedging it, an LLM that is fully transparent to the user needs to be able to reflect on its internal belief distribution and output a summary of all options it deems possible, and how likely they are. To test whether LLMs possess this capability, we develop the SelfReflect metric, an information-theoretic distance between a given summary and a distribution over answers. In interventional and human studies, we find that SelfReflect indicates even slight deviations, yielding a fine measure of faithfulness between a summary string and an LLM's actual internal distribution over answers. With SelfReflect, we make a resounding negative observation: modern LLMs are, across the board, incapable of revealing what they are uncertain about, neither through reasoning, nor chains-of-thoughts, nor explicit finetuning. However, we do find that LLMs are able to generate faithful summaries of their uncertainties if we help them by sampling multiple outputs and feeding them back into the context. This simple approach shines a light at the universal way of communicating LLM uncertainties whose future development the SelfReflect score enables. To support the development of this universal form of LLM uncertainties, we publish our metric under `https://github.com/apple/ml-selfreflect`.

## 1 INTRODUCTION

When large language models (LLMs) are uncertain about a response, either because the query is ambiguous or because they are factually unsure, they should indicate it. Consider the example in Fig. 1. The LLM's internal distribution comprises a variety of answers, so it is not enough to just output the greedy response. While existing uncertainty quantification approaches augment the greedy response (or any other single sample from the distribution) with a numerical measure of uncertainty (Aichberger et al., 2024; Fadeeva et al., 2023; Fomicheva et al., 2020; Malinin and Gales, 2020) or verbalize the confidence in the response (Lin et al., 2022; Yona et al., 2024), this offers limited insight into the model's beliefs: we do not see the full range of cities the LLM believes are plausible, nor the variety of supporting information (e.g., that Paris hosts the French government).

We believe we can do better than this. As motivation, consider the following comment on Gödel's proof on the incompleteness of number theory.

> *Gödel had the insight that a statement of number theory could be about a statement of number theory (possibly even itself), if only numbers could somehow stand for statements.*
>
> Hofstadter (1979)

Gödel's key idea was that statements of number theory are expressive of much more than just integers. The same holds for strings: An answer string $s$ generated by an LLM is expressive enough to describe

**User query** $q$: What is the main city of France?

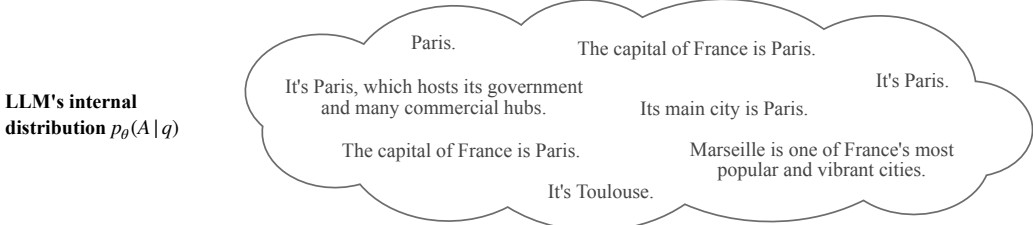

**LLM's internal distribution** $p_\theta(A \mid q)$

**Normal (greedy) answer:** 'The capital of France is Paris.'

**Numerical uncertainty:** ('The capital of France is Paris.', 75%)

**Verbalized uncertainty:** 'I'm very sure that the capital of France is Paris.'

**Self-reflective uncertainty:** 'I'm 75% sure that it's Paris, its capital and commercial hub, but it could also be Toulouse or Marseille.'

Figure 1: LLMs have internal answer distributions about user queries. Rather than just sampling an output, possibly combined with a percentage, LLMs should generate a string that is self-reflective of their internal distribution, summarizing all possibilities and which they find the most likely.

a *distribution over* all answer strings the LLM could generate. We can therefore use a single string $s$ to summarize the LLM's distribution $p_\theta(A \mid q)$ over responses $A$ to a query $q$. We see this in the "self-reflective uncertainty" example of Fig. 1: A single string conveys the relative degrees of belief in different cities, and covers the detailed facts of all answers of the distribution. Communicating uncertainty like this, through a string rather than a number, is a new paradigm for uncertainty quantification – so novel that there exists no way to benchmark it. Our contribution is thus twofold:

First, we define a *benchmark* that evaluates whether a given self-summary string faithfully represents an LLM's internal distribution over possible responses[1]. The underlying challenge here is to measure whether a single string *"carries the same information"* as a *distribution over* strings, in some information-theoretic sense that takes into account both mentioned facts and their relative likelihoods. Our theoretical analysis yields the SelfReflect metric. It scores how well a self-summary string is predictively sufficient of a distribution of answer strings. To ensure that this measure of faithfulness is robust in practice, we conduct controlled experiments on both free-form and closed-form question datasets. We find that the SelfReflect score precisely discriminates good from bad (and almost-good) summaries of answer distributions, and that it agrees with human judgements, in both cases outperforming other possible benchmark metrics such as LM judges and embedding distances.

Second, we use the SelfReflect metric to test whether 20 modern LLMs can generate self-reflective uncertainty strings. We make a resounding negative observation: Neither explicit prompting, nor reasoning, nor SFT and DPO fine-tuning enable an LLM to faithfully summarize its internal beliefs. Its output may have a summary-style format, but it mentions arbitrary possibilities, not those that the LLM actually believes in. It is, however, possible to give honest insights into the internal answer distributions by explicitly i.i.d. sampling an LLM and returning this back for summarization.

These findings mark but the start of this new avenue of uncertainty quantification, and, in extension, of fundamentally making LLMs aware of their internal uncertainties. We expect that future advances along our SelfReflect benchmark metric will unlock more honest and trustworthy LLM interactions.

## 2 RELATED WORK

### 2.1 UNCERTAINTY IN LLMS

Most work on uncertainty in LLMs associates a single numerical expression of uncertainty to a specific string like the greedily decoded response. Since LLMs are, in essence, probabilistic next-token classifiers, one can attempt to read their uncertainty off their token logits (Aichberger et al.,

---

[1]Throughout the paper, we use anthropomorphised language like "what the LLM deems possible" or "being honest". We do this only for brevety and giving intuitions. Technically, we always mean "summarizing the answers that an LLM could give if sampled multiple times".

2024; Fadeeva et al., 2023; Fomicheva et al., 2020; Malinin and Gales, 2020). These methods can be extended to longer LLM answers for example by searching for fact tokens and extracting their logits (Fadeeva et al., 2024) and made more human-readable by transforming the numeric uncertainty into a string like "I am very sure that..." (Lin et al., 2022; Yona et al., 2024). Still, these approaches quantify the uncertainty of only a single element of the LLM's internal distribution.

So how can the full uncertainty of the LLM's distribution be captured? Farquhar et al. (2024) cluster answers sampled from the LLM's internal distribution semantically and calculate an entropy over the clusters. This considers the full distribution over strings, but it still reduces the uncertainty to a single number and presents this number alongside a single string from the distribution. Moving towards richer uncertainty explications, Xu et al. (2024) generate multiple samples from an LLM, use GPT-4 to summarize the distribution of samples and train the LLM to output such summaries. Similarly, Yang et al. (2024b) train an LLM to output strings that delineate which facts it is uncertain about. This is arguably one of the richest ways to express an LLM's uncertainty. But both papers, focusing on the generation of summaries rather than on evaluation, use simple LM judges to rate the summary strings. As we show in Section 4.1, LM judges can not discern how faithfully a string reflects a distribution over strings beyond relatively simple good vs bad cases. Our SelfReflect gives a better-founded and more precise metric to compare whether a summary string contains the same information as the LLM's internal distribution, enabling to further develop this new avenue of LLM uncertainties.

## 2.2 SUMMARIZATION

Testing whether a summary of a long document is *good* has a long history in natural language processing (NLP) (Zhang et al., 2024). Summaries are traditionally rated in terms of consistency with the long document, relevance of the chosen information, and fluency and coherence of their sentences (Fabbri et al., 2021), as rated by humans or recently by LM judges (Jain et al., 2023). In modern LLM-generated summaries, fluency and coherence are usually granted, so that the focus lays on the consistency and relevance of the summary, in other words, whether it *contains the same information* as the long document. This fundamental question dates back to the Cloze test (Taylor, 1953). This test, originally designed for human language learners, masks out words from the long document and asks to fill them in. Summarization metrics like BLANC (Vasilyev et al., 2020) run this test twice, once when conditioning an NLP model on the summary and once without. If the summary contains correct information, the NLP model should fill in better words. The masked-out performance can be quantified either as an accuracy gain (Vasilyev et al., 2020) or, more softly, as a pseudo log-likelihood (Shin et al., 2019; Wang and Cho, 2019; Salazar et al., 2020; Kauf and Ivanova, 2023). Other recent metrics use masked-out tasks to estimate pointwise mutual information (Jung et al., 2024).

Since our SelfReflect metric also quantifies the quality of a summary, we base it off Cloze-like masked-out tasks. But there is a twist: The summary string $s$ does not summarize another string but a *distribution over* strings $p_\theta (A \mid q)$. This means we must go beyond comparing $s$ to a specific string $a \sim p_\theta (A \mid q)$, to quantifying how faithfully $s$ represents the density over the string space that $p_\theta (A \mid q)$ defines, i.e., to all possible answers and how likely they are. To this end, we re-think masked-out tasks from the lens of sufficient statistics in the following section.

## 3 DISTANCES BETWEEN SUMMARY STRINGS AND DISTRIBUTIONS OF STRINGS

Our main challenge is to find a distance that quantifies the extent to which a summary string *carries the same information as* an LLM's internal answer distribution. We build a theoretical foundation for sufficient statistics in string spaces in Section 3.1 and develop the SelfReflect metric in Section 3.2.

### 3.1 SUMMARIES AS PREDICTIVE SUFFICIENT STATISTICS

Suppose we have an LLM (which we denote LLM$_\theta$), prompted with a random query $Q$. We posit that this puts us in a state $\Theta_Q$, which allows us to sample random responses $B$. We are interested in summarizing this distribution over responses. Let $A^{(1:N)} := (A^{(1)}, \ldots, A^{(N)}) \in \mathcal{X}^N$ be a set of responses sampled from LLM$_\theta$, where $\mathcal{X}$ is the space of finite strings.[2] Consider a summarization

---

[2] These $N$ samples may be generated independently and identically to $B$, but we do not require this; for example, the distribution over subsequent answers could depend on the previous answers.

function $\psi : \mathcal{X}^N \longrightarrow \mathcal{X}$ that, given $A^{(1:N)}$, generates a summary $S := \psi(A^{(1:N)})$. What criteria should $\psi$ satisfy if its summaries are to exactly capture $\text{LLM}_\theta$'s distribution over $B$?

Continuing the example from Fig. 1, we can see that an ideal summary of $A^{(1:N)}$ should neither omit important details from the answer distribution nor add extra details. For example, a summary stating "The capital of France is Paris" would ignore the LLM's belief in Marseille or Toulouse, whereas a summary stating "The capital of France is Paris but for a period in history, it was Orléans" would be adding unfaithful details. The same holds for the relative likelihood of answers: the ideal summary should state that the capital of France is most likely Paris, and not Toulouse or Marseille, because this answer has a higher probability mass in the LLM's internal distribution. This indicates that an *ideal summary should capture exactly the same information about the answer distribution as that contained in the sampled answers*. We can formalize this in terms of mutual information,

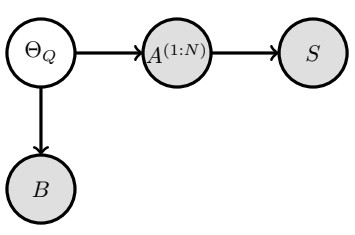

Figure 2: Graphical model for the sufficiency that SelfReflect quantifies.

**Definition 3.1** (Ideal summary). *An ideal summary $S$ of answers $A^{(1:N)}$ of an LLM satisfies*

$$\mathcal{I}\left\{A^{(1:N)}\,;B\right\} = \mathcal{I}\left\{S\,;B\right\} \tag{1}$$

*Here, $\mathcal{I}\{Y\,;Z\}$ denotes the mutual information between $Y$ and $Z$. Intuitively, for any subsequent answer $B$ from the LLM, the information about $B$ contained in $A^{(1:N)}$ is exactly captured by $S$.*

This definition is closely tied to the notion of predictive sufficiency (Lauritzen, 1974), whereby a statistic $T(X^{(1:N)})$ of observations $X^{(1:N)}$ is called sufficient if it satisfies $p\left(X \mid X^{(1:N)}\right) = p\left(X \mid T(X^{(1:N)})\right)$ for any subsequent observation $X$. In fact, we can reframe Definition 3.1:

**Proposition 3.1** (Connection to predictive sufficiency). *For an ideal summary $S$ of answers $A^{(1:N)}$,*

$$\mathcal{I}\left\{A^{(1:N)}\,;B\right\} = \mathcal{I}\left\{S\,;B\right\} \Longleftrightarrow p\left(B \mid A^{(1:N)}\right) = p\left(B \mid S\right) \tag{2}$$

*Intuitively, the ideal summary $S$ is a predictive-sufficient statistic of the answers $A^{(1:N)}$ for $B$.*

From Definition 3.1 and Proposition 3.1, we see that a measure of how much $p\left(B \mid A^{(1:N)}\right)$ diverges from $p\left(B \mid S\right)$ would be a good metric for measuring how faithfully $S$ reflects the sampled answers $A^{(1:N)}$. Towards this, we formulate a Cloze-task based on masked-token prediction that constitutes a simple yet equivalent characterization of the desired predictive sufficiency. Let $B_i$ denote the $i$th word of $B$ and let $B_{-i} := (B_j)_{j\neq i}$ denote all other words of the answer. We propose predicting the missing word $B_i$ from the rest of the words $B_{-i}$ with the extra context of either the sampled answers $A^{(1:N)}$ or their summary $S$. Identical behavior in this masked-token prediction task turns out to be equivalent to predictive sufficiency (and hence, Definition 3.1):

**Proposition 3.2** (Informal; towards the SelfReflect metric). *For answers $A^{(1:N)}$ and their summary $S$, under mild conditions on all involved distributions and support of $B$, we have:*

$$p\left(B \mid A^{(1:N)}\right) = p\left(B \mid S\right) \Longleftrightarrow \text{for all masking indices } i,\ p\left(B_i \mid A^{(1:N)}, B_{-i}\right) = p\left(B_i \mid S, B_{-i}\right) \tag{3}$$

Full details and proofs of Propositions 3.1 and 3.2 are given in Appendix A. Proposition 3.2 motivates us to measure the divergence between the distributions $p\left(B_i \mid S, B_{-i}\right)$ and $p\left(B_i \mid A^{(1:N)}, B_{-i}\right)$ as a tractable metric for the quality of a summary, forming the basis of the SelfReflect metric.

## 3.2 THE SELFREFLECT METRIC

Proposition 3.2 tells us we can use a sequence of masked-out tasks to quantify whether a summary $s$ contains the same information about $\text{LLM}_\theta$'s distribution $p_\theta\left(B \mid q\right)$ as a sequence of $N$ samples from that distribution. We approximate this task using a second judge LLM, $\text{LLM}_J$, to estimate the conditional distribution over masked-out words. Intuitively, irrespective of whether we show the sampled answers or their ideal summary, a judge LLM should predict the same masked tokens.

Concretely, we sample a new response $B$ at temperature 1 from $\text{LLM}_\theta$, mask out one word $B_i$, and ask $\text{LLM}_J$ to predict $B_i$ given the remainder of the answer $B_{-i}$, the query $q$, and either the summary

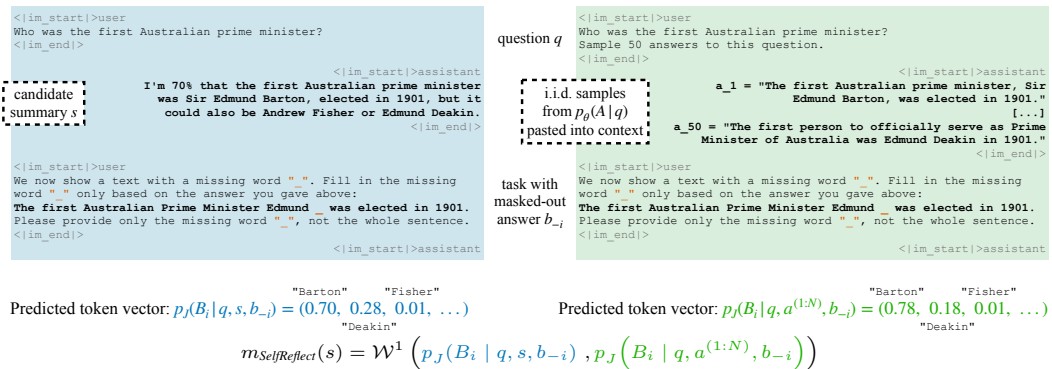

Figure 3: To test whether a summary string $s$ contains the same information as a set of samples $a^{(1:N)}$, SelfReflect prompts an LLM twice. First, it provides the summary as context; next, it provides the concatenated samples. SelfReflect then compares the resulting distributions via a masked-out task.

$s$ or a sequence $a^{(1:N)}$ of $N$ samples from $p_\theta\left(A^{(1:N)} \mid q\right)$, see Fig. 3. This yields two distributions $p_J\left(B_i \mid Q = q, A^{(1:N)} = a^{(1:N)}, B_{-i} = b_{-i}\right)$ and $p_J\left(B_i \mid Q = q, S = s, B_{-i} = b_{-i}\right)$, over the vocabulary space of $\text{LLM}_J$ which we compare using the 1-Wasserstein distance.[3] We marginalize over $B$ and index $i$ to satisfy the requirements of Proposition 3.2. Finally, we take the expectation over all summaries that a summarization strategy $\psi$ writes for each question in the dataset:

$$m_{\text{SelfReflect}}(\psi) = \mathbb{E}_{Q, A^{(1:N)}, B, i}\left[\mathcal{W}^1\Big(p_J(B_i \mid Q, \psi(Q), B_{-i}), p_J\Big(B_i \mid Q, A^{(1:N)}, B_{-i}\Big)\Big)\right] \quad (4)$$

Here, $\psi$ is any method that makes $\text{LLM}_\theta$ output a summary of its internal distribution in response to a query.[4] We put a set of $N = 50$ samples $A^{(1:N)}$ per query into the reference context of $\text{LLM}_J$ and estimate Eq. (4) via Monte Carlo sampling. We iterate over $M = 50$ samples of $B$ and each of their words $i$ (except stopwords) to create masked-out tasks. We repeat this over 1000 queries per dataset to average the final SelfReflect benchmark score. These are relatively conservative settings that take 67 minutes to compute on a node of 8 NVIDIA A100 GPUs. In Appendix B, we show that we can also reduce to 9 minutes or even below one minute if the goal is not to reach a benchmark metric precise to multiple digits but rapid development or reward signals. Further, literature notes that Cloze-like evaluations are often limited by synonyms (Kauf and Ivanova, 2023), so we post-hoc flatten $p_J$ with $\tau = 5$. We quantitatively find this improves discriminability since especially Instruction-tuned $\text{LLM}_J$ models otherwise place too much mass on one specific masked-out word. We discuss further design choices in Section 7.

We explore different choices of $\text{LLM}_J$ and find that SelfReflect is robust to the exact choice, see the quantitative results in Appendix C and the qualitative example in Appendix D. We find that Qwen 2.5 Instruct (Yang et al., 2024a) captures both textual details and the implicit relative certainties in summaries or concatenated samples in its context even when they are subtle. The 7B model provides results almost on par with the 72B model, so we choose it for efficiency.

## 4 CAN SELFREFLECT SCORES QUANTIFY HOW GOOD SUMMARIES ARE?

We now verify that the SelfReflect metric works as a benchmarking tool, based on three pillars: Distinguishing hand-crafted good and bad summaries on free-form questions, a simplified study on multiple-choice QA answer distributions, and a comparison to which summaries humans deem faithful. In all studies, we compare SelfReflect to several other possible benchmarking metrics.

---

[3]If $\text{LLM}_J$ is a black-box model that only returns the top-predicted word, i.e., $p_J$ are one-hot vectors, our 1-Wasserstein comparison simplifies into an accuracy that tests whether the two predicted words are equal. This is why we choose 1-Wasserstein distance over KL divergence, which is quantitatively equally good.

[4]While the link to sufficiency only holds if $\psi$ depends only on $a^{(1:N)}$, the metric is well-defined whether the summary generation involves taking samples in-between or generating a summary answer for $q$ in other ways.

Table 1: Given pairs of good and bad summaries, we measure how often the SelfReflect score, and other benchmark metrics, correctly assign a better score to the good summary to verify that they work as benchmarking metrics. We test multiple pairs of good and bad summaries, e.g., lacking possibilities or lacking details. Mean $\pm$ 95% confidence interval. Per-dataset results are in Appendix G.

| Metric | Good summaries vs bad summaries | Good vs almost-good | Detailed vs truncated | Verbalized uncertainty vs only majority answer | Verbalized vs or-concatenated | Percentage vs or-concatenated |
|---|---|---|---|---|---|---|
| Summarization | $93.33\%_{\pm 0.89\%}$ | $39.72\%_{\pm 1.87\%}$ | $53.05\%_{\pm 6.04\%}$ | $19.90\%_{\pm 5.66\%}$ | $58.12\%_{\pm 7.00\%}$ | $64.92\%_{\pm 6.77\%}$ |
| InfoSumm | $\mathbf{99.87\%}_{\pm 0.13\%}$ | $60.81\%_{\pm 1.87\%}$ | $49.24\%_{\pm 6.05\%}$ | $15.71\%_{\pm 5.16\%}$ | $27.75\%_{\pm 6.35\%}$ | $10.99\%_{\pm 4.44\%}$ |
| LM Judge | $98.33\%_{\pm 0.46\%}$ | $47.32\%_{\pm 1.91\%}$ | $59.92\%_{\pm 5.93\%}$ | $19.37\%_{\pm 5.60\%}$ | $34.55\%_{\pm 6.74\%}$ | $35.08\%_{\pm 6.77\%}$ |
| Opt. Transport | $80.16\%_{\pm 1.43\%}$ | $60.78\%_{\pm 1.87\%}$ | $39.69\%_{\pm 5.92\%}$ | $48.69\%_{\pm 7.09\%}$ | $52.88\%_{\pm 7.08\%}$ | $69.11\%_{\pm 6.55\%}$ |
| Embedding | $96.50\%_{\pm 0.66\%}$ | $65.49\%_{\pm 1.82\%}$ | $65.65\%_{\pm 5.75\%}$ | $10.99\%_{\pm 4.44\%}$ | $43.98\%_{\pm 7.04\%}$ | $36.65\%_{\pm 6.83\%}$ |
| SR-logl | $96.37\%_{\pm 0.67\%}$ | $85.90\%_{\pm 1.33\%}$ | $86.64\%_{\pm 4.12\%}$ | $58.12\%_{\pm 7.00\%}$ | $40.84\%_{\pm 6.97\%}$ | $49.21\%_{\pm 7.09\%}$ |
| SR-PMI | $88.40\%_{\pm 1.15\%}$ | $33.64\%_{\pm 1.81\%}$ | $53.44\%_{\pm 6.04\%}$ | $25.65\%_{\pm 6.19\%}$ | $14.14\%_{\pm 4.94\%}$ | $20.42\%_{\pm 5.72\%}$ |
| SR-sampling-free | $88.26\%_{\pm 1.15\%}$ | $54.85\%_{\pm 1.90\%}$ | $73.28\%_{\pm 5.36\%}$ | $38.74\%_{\pm 6.91\%}$ | $35.08\%_{\pm 6.77\%}$ | $38.22\%_{\pm 6.89\%}$ |
| SR-P(True) | $65.29\%_{\pm 1.70\%}$ | $81.91\%_{\pm 1.47\%}$ | $69.47\%_{\pm 5.58\%}$ | $\mathbf{87.96\%}_{\pm 4.62\%}$ | $71.73\%_{\pm 6.39\%}$ | $\mathbf{86.39\%}_{\pm 4.86\%}$ |
| SelfReflect | $98.77\%_{\pm 0.40\%}$ | $\mathbf{93.20\%}_{\pm 0.96\%}$ | $\mathbf{93.13\%}_{\pm 3.06\%}$ | $85.34\%_{\pm 5.02\%}$ | $\mathbf{72.77\%}_{\pm 6.31\%}$ | $80.10\%_{\pm 5.66\%}$ |

**Baselines.** While developing SelfReflect, we experimented with approaches from various roots for comparing a summary string $s$ to a set of strings $a^{(1:N)}$. First, we compare against multiple metrics from summarization literature that treat $a^{(1:N)}$ as a single document that is summarized by $s$. *Summarization* (Jain et al., 2023) uses the judge $LLM_J$ to rate the summary in terms of consistency, fluency, relevance, and coherence. *LM Judge* prompts $LLM_J$ to rate how well $s$ matches $a^{(1:N)}$ in one prompt, following the chain-of-thoughts prompt of Xu et al. (2024). *InfoSumm* (Jung et al., 2024) uses a masked-out task to estimate pointwise mutual-information between summary and document. Next, we turn to the neighboring field of calibration. Wang and Holmes (2024) argue that calibration can be seen as a distance to a centroid. We implement this in *Embedding* by comparing embeddings of $s$ to $a^{(1:N)}$. Finally, from an *Optimal transport* perspective (Peyré et al., 2019), we let $LLM_J$ split $s$ into a "distribution" over atomic statements and likelihoods, compute a pairwise entailment matrix and return the Earth Mover's distance to $p_\theta (A \mid q)$.

**Ablations.** We also ablate key characteristics of SelfReflect (SR). *SR-logl* replaces the Wasserstein distance over the whole logit vector with only the log probability assigned to the masked-out word given either context. *SR-PMI* (SelfReflect with Pointwise Mutual Information) even removes the one-by-one masked-out task and directly compares the log likelihoods of the full answers $A^{(1:N)}$; analogous to Proposition 3.1. *SR-sampling-free* uses the masked-out task, but compares the masked-out logit vectors given the summary to predictions of $LLM_\theta$ given $q$, instead of putting sampled answers into the context of $LLM_J$. *SR-P(True)* changes from a generative to a discriminative masked-out task, asking $LLM_J$ whether several candidates words fit or not (via the P(True) method of (Kadavath et al., 2022)), given either the summary or the samples. More details are in Appendix E.

### 4.1 STUDY 1: DISTINGUISHING GOOD FROM BAD AND ALMOST-GOOD SUMMARIES

We first conduct an interventional study to test whether summaries that we know are good are judged as better than summaries that we know are bad. To this end, we take $3 \times 1,000$ open-ended questions from Natural Questions (Kwiatkowski et al., 2019), TriviaQA (Joshi et al., 2017), and SimpleQA (Wei et al., 2024), and let Qwen 2.5 7B Instruct generate 50 answers each. We then give these answer distributions to Gemini 2.0 Flash and prompt it to generate *good* summaries, containing all possibilities, details, and relative likelihoods, and *bad* summaries, which alter key facts of the good summaries, but keep their remaining style (human-written summaries reach equivalent results in Appendix F). We then calculate which score SelfReflect gives to the summaries, and in how many percent of the good-bad pairs it correctly gives the good summary a better (lower) score than the bad one.

Table 1 shows that SelfReflect correctly discriminates good from bad in 98.77% of cases. But several other baseline metrics also score over 90%. So we make the task harder by comparing good to *almost-good* summaries, which only contain facts that are faithful to the answer distribution, but leave out some possibilities and details. SelfReflect gives the good summary a better score than the almost-good summaries in 93.2% of all questions. The other metrics, including the LM judge used in literature, can no longer distinguish these fine-grained quality differences and are thus not good for benchmarking. We ablate this multiple times, finding the SelfReflect score also correctly notices when a summary does not mention all written details, or when it does not mentions all options

but only the majority answer. Even when a summary mentions all options (*"It is ... or ... or ..."*), SelfReflect assigns a yet better score to a summary that also faithfully delineates in words or numbers which options are how likely. The SelfReflect score picks these subtle differences (last five columns) up better than all other benchmarking metrics, matched only in some cases by its own SR-P(True) ablation. Further, all these tests checked whether SelfReflect can distinguish the quality of *individual summaries*. In later benchmarks, which average over thousands of summaries per method, averaged SelfReflect scores will become even more exact by the law of large numbers, making SelfReflect a precise benchmarking tool that allows to iteratively develop summary-generating methods.

## 4.2 STUDY 2: DISTANCES OF MULTIPLE-CHOICE DISTRIBUTIONS

Next, we investigate SelfReflect in a narrower setup. We generate $2 \times 1,000$ answer distributions for MMLU (Hendrycks et al., 2021), a multiple-choice dataset with choices A, B, C, and D for each question, with Gemma 3 12B (non-Instruct) (Gemma Team et al., 2025), and Qwen 2.5 7B Instruct. Since MMLU is a multiple-choice dataset, we can sample the LLM multiple times to obtain a simple categorical distribution. We then create summaries that either talk about this distribution *"The answer is most likely C (54% sure), but it could also be B (32% sure) or A (14% sure)."* or that mention the most likely answer only, are overconfident, or give random percentages. This gives a range of different-

Table 2: Agreement (rank corr.) between SelfReflect, and others, and a special benchmark metric for MMLU. Mean $\pm$ 95%.

| Metric | Per Question | Whole Dataset |
|---|---|---|
| Summarization | $0.45_{\pm 0.03}$ | $0.80_{\pm 0.00}$ |
| LM Judge | $\mathbf{0.76}_{\pm 0.02}$ | $\mathbf{1.00}_{\pm 0.00}$ |
| Opt. Transport | $0.67_{\pm 0.02}$ | $0.82_{\pm 0.00}$ |
| Embedding | $-0.24_{\pm 0.04}$ | $0.19_{\pm 0.02}$ |
| SR-logl | $0.59_{\pm 0.03}$ | $\mathbf{1.00}_{\pm 0.00}$ |
| SR-PMI | $0.07_{\pm 0.03}$ | $0.20_{\pm 0.00}$ |
| SR-sampling-free | $0.51_{\pm 0.03}$ | $0.80_{\pm 0.00}$ |
| SR-P(True) | $0.57_{\pm 0.03}$ | $\mathbf{1.00}_{\pm 0.00}$ |
| SelfReflect | $0.65_{\pm 0.03}$ | $\mathbf{1.00}_{\pm 0.00}$ |

quality summaries that the benchmark metrics have to tell apart. The categorical setup of MMLU also allows to calculate a reference benchmark metric, namely the Wasserstein distance between the percentages mentioned in the summary and that of the real distribution. This lets us test if the SelfReflect metric and the other baselines agrees with the true distance in this special case. Specifically, we report the rank correlation between them and the reference metric.

Table 2 shows that most metrics have a positive rank correlation with the reference metric. The LM judge metric even slightly outperforms SelfReflect on this particular task, indicating that SelfReflect may be slightly noisy on individual questions when summaries contain exact probabilities. However, when we compute the average score across all questions, as it will later be used in the benchmark, SelfReflect, like two of its ablations and LM Judge, achieves a perfect agreement with the reference metric. This shows SelfReflects generic power as benchmark metric, even in this special case.

## 4.3 STUDY 3: DO THE RATINGS ALIGN WITH HUMAN RATINGS?

Finally, we assess whether SelfReflect scores are aligned with human judgements. We conduct a user study using 200 open-ended questions from the TriviaQA dataset (Joshi et al., 2017). For each question, we generate ten sample responses using Phi-4 (Abdin et al., 2024), and four summaries: a *good* summary and a *bad* summary, generated using Gemini 2.0 Flash as in Section 4.1; a *greedy* summary, i.e., the greedy response of Phi-4; and a Chain of Thought (*CoT*) summary, using Phi-4 to reason about possible answers and then summarize its reasoning. Note that the greedy and CoT summaries are not based on the actual samples. All prompts are provided in Appendix E. Raters were shown the question, the ten sample answers, and two of the summaries, and asked to choose which best summarized the set of samples. Each question/summary combination was evaluated by 5 raters. To assess agreement between human raters, we calculate Krippendorff's $\alpha$.[5] Alternative agreement metrics such as Cohen's kappa or Fleiss' kappa are not appropriate here since each rater only rates a subset of the combinations. We then calculate Krippendorff's $\alpha$ between the majority human preference and that of SelfReflect and other scores. Further details are in Appendix I.

As we see from Table 3, SelfReflect has the highest overall alignment with the majority human judgement ($\alpha = 0.690$). This is close to the inter-human alignment ($\alpha = 0.723$) and significantly higher than any of the competing methods or ablations. Looking into the individual summary types, we see all metrics other than SR-P(True) have good alignment with humans on the *bad* vs *good*, *bad* vs *greedy*, and *bad* vs *CoT* comparisons. However, the other metrics show poor agreement

---

[5] https://github.com/grrrr/krippendorff-alpha/tree/master

Table 3: Agreement of metrics with human preference (consensus over five raters) on a pairwise summary preference task, using Krippendorff's $\alpha$ (values in [-1, 1]; positive numbers indicate agreement). Also shown is Krippendorff's $\alpha$ between individual human raters. Mean $\pm$ 95% CI.

| | all | bad vs good | bad vs greedy | bad vs CoT | good vs greedy | good vs CoT | greedy vs CoT |
|---|---|---|---|---|---|---|---|
| Summarization | $0.480_{\pm 0.050}$ | $0.950_{\pm 0.046}$ | $0.910_{\pm 0.050}$ | $\mathbf{0.940}_{\pm 0.046}$ | $-0.211_{\pm 0.156}$ | $-0.067_{\pm 0.135}$ | $0.260_{\pm 0.121}$ |
| LM Judge | $0.517_{\pm 0.046}$ | $0.940_{\pm 0.048}$ | $\mathbf{0.920}_{\pm 0.058}$ | $0.930_{\pm 0.046}$ | $-0.063_{\pm 0.152}$ | $-0.015_{\pm 0.151}$ | $0.267_{\pm 0.128}$ |
| Opt. Transport | $0.487_{\pm 0.047}$ | $0.850_{\pm 0.076}$ | $0.779_{\pm 0.085}$ | $0.679_{\pm 0.104}$ | $0.098_{\pm 0.155}$ | $0.265_{\pm 0.132}$ | $0.191_{\pm 0.146}$ |
| Embeddings | $0.435_{\pm 0.047}$ | $0.750_{\pm 0.081}$ | $0.799_{\pm 0.087}$ | $0.477_{\pm 0.125}$ | $-0.363_{\pm 0.136}$ | $0.331_{\pm 0.135}$ | $\mathbf{0.490}_{\pm 0.121}$ |
| SR-PMI | $0.436_{\pm 0.053}$ | $0.820_{\pm 0.081}$ | $0.890_{\pm 0.067}$ | $0.769_{\pm 0.080}$ | $-0.246_{\pm 0.156}$ | $0.029_{\pm 0.147}$ | $0.246_{\pm 0.114}$ |
| SR-sampling-free | $0.530_{\pm 0.045}$ | $0.829_{\pm 0.076}$ | $0.870_{\pm 0.071}$ | $0.799_{\pm 0.080}$ | $0.025_{\pm 0.143}$ | $0.241_{\pm 0.131}$ | $0.340_{\pm 0.141}$ |
| SR-P(True) | $-0.032_{\pm 0.052}$ | $-0.029_{\pm 0.138}$ | $-0.335_{\pm 0.124}$ | $-0.474_{\pm 0.120}$ | $0.311_{\pm 0.147}$ | $0.409_{\pm 0.125}$ | $-0.024_{\pm 0.143}$ |
| SelfReflect | $\mathbf{0.690}_{\pm 0.036}$ | $\mathbf{0.990}_{\pm 0.015}$ | $0.850_{\pm 0.066}$ | $0.850_{\pm 0.070}$ | $\mathbf{0.489}_{\pm 0.131}$ | $\mathbf{0.599}_{\pm 0.103}$ | $0.329_{\pm 0.125}$ |
| Human vs human | $0.723_{\pm 0.027}$ | $0.988_{\pm 0.013}$ | $0.906_{\pm 0.035}$ | $0.871_{\pm 0.048}$ | $0.441_{\pm 0.075}$ | $0.636_{\pm 0.064}$ | $0.452_{\pm 0.069}$ |

with humans on the more nuanced *good* vs *greedy* and *good* vs *CoT*. For all pairs of summary type, SelfReflect is close to inter-human agreement and either the most aligned with the majority human preference, or has overlapping 95% confidence intervals with the most aligned metric.

## 5 CAN LLMs GENERATE SELF-REFLECTIVE RESPONSES?

Now that we have a metric to benchmark how well summaries summarize the distribution of LLM answers, we explore the performance of different summarization methods, that is: can one (somehow) make LLMs reflect on and summarize their own internal distributions?

### 5.1 EXPERIMENTAL SETUP

We distinguish two broad categories of methods: A) *Sample & summarize*: draw multiple independent samples from the LLM, and then feed them back to the LLM to summarize them, B) *Single-decoding*: methods which utilize only one decoding, requiring the LLM to reflect on its internal distribution on its own. We consider three single-decoding methods: a) *Basic*: a prompt asking the LLM for a summary of all possible answer options; b) *CoT*: a prompt inducing chain-of-thoughts reasoning about the possible answers and then summarizing them; c) *Greedy*: Simply return the greedy-decoding answer without summarizing all possibilities; we use this as a baseline. The *Greedy* baseline is, in fact, strong: On questions where a model has a unimodal distribution on a specific answer, *Greedy* is in fact the best possible summary of this distribution and achieves a competitive SelfReflect score. To account for this, we report the percentage of answers where we observe such "$p_\theta (A \mid q)$ *unimodal*" distributions per LLM. We evaluate all summarization methods via the SelfReflect score on $3 \times 1000$ randomly chosen questions from Natural Questions, SimpleQA, and TriviaQA (retrieval-augmented generation experiments on HotpotQA are in Appendix M). We use the same LLM to sample the answers to the question and generate the summaries in order to assess whether LLMs can access and describe their *own* internal distributions. We publish all benchmarking code upon publication. More details are in Appendix J.

### 5.2 RESULT: LLMs CAN ONLY ACCESS THEIR INTERNAL DISTRIBUTIONS WITH SOME HELP

As we see in Table 4, *Sample & summarize* is able to create summaries that faithfully reflect the model's internal uncertainty, consistently outperforming the *Greedy* baseline. In fact, its score matches that of humans asked to summarize samples from an LLM distribution, with humans achieving $90 \cdot 10^{-3}$ when summarizing Qwen 2.5 72B Instruct answer distributions and *Sample & summarize* achieving $88 \cdot 10^{-3}$ on the data-split of Appendix F. However, *Sample & summarize* helps the LLM in so far that it explicitly samples it i.i.d. and provides the samples back as context to summarize.

It is of particular interest if we can generate such self-reflective outputs without needing to sample in-between, for runtime and elegancy. Table 4 unveils a resounding negative result: No single-decoding methods is able to out-perform the *Greedy* baseline, corroborating that LLMs are not able to fully verbalize their own uncertainty by themselves, despite our best efforts to optimize the prompts.

Maybe this task is too complex for an instruction-tuned LLM. We thus turn to reasoning models, asking them to reason about all possibilities and then output a summary. But Table 5 reinforces our

Table 4: SelfReflect score $\downarrow$ ($\times 10^{-3}$ for readability). The results in small font are relative to *Greedy*. $p_\theta(A|q)$ *unimodal* is the proportion of questions for which the LLM always gives the same answer.

| Model | $p_\theta(A\mid q)$ unimodal | Single-decoding methods | | | Sample & summarize | |
|---|---|---|---|---|---|---|
| | | Greedy | Basic | CoT | $N=10$ | $N=20$ |
| Qwen2.5 0.5B Instruct (Yang et al., 2024a) | 7% | 96 | $95_{-1}$ | $94_{-2}$ | $96_{-0}$ | $96_{-0}$ |
| Qwen2.5 1.5B Instruct (Yang et al., 2024a) | 17% | 94 | $94_{-0}$ | $92_{-2}$ | $87_{-7}$ | $87_{-7}$ |
| Qwen2.5 3B Instruct (Yang et al., 2024a) | 27% | 97 | $99_{+2}$ | $99_{+2}$ | $91_{-6}$ | $89_{-8}$ |
| Qwen2.5 7B Instruct (Yang et al., 2024a) | 36% | 96 | $99_{+3}$ | $101_{+5}$ | $91_{-5}$ | $90_{-6}$ |
| Qwen2.5 14B Instruct (Yang et al., 2024a) | 52% | 92 | $97_{+5}$ | $99_{+7}$ | $86_{-6}$ | $85_{-7}$ |
| Qwen2.5 32B Instruct (Yang et al., 2024a) | 49% | 96 | $102_{+6}$ | $105_{+9}$ | $91_{-5}$ | $91_{-5}$ |
| Qwen2.5 72B Instruct (Yang et al., 2024a) | 50% | 91 | $94_{+3}$ | $96_{+5}$ | $85_{-6}$ | $84_{-7}$ |
| Phi 4 14B (Abdin et al., 2024) | 36% | 92 | $92_{-0}$ | $93_{+1}$ | $85_{-7}$ | $84_{-8}$ |
| Ministral 8B Instruct 2410 (Jiang et al., 2024) | 25% | 107 | $106_{-1}$ | $105_{-2}$ | $101_{-6}$ | $100_{-7}$ |
| Llama 3.1 70B Instruct (Meta AI, 2024a) | 51% | 92 | $92_{-0}$ | $95_{+3}$ | $87_{-5}$ | $87_{-5}$ |
| Llama 3.3 70B Instruct (Meta AI, 2024b) | 63% | 94 | $98_{+4}$ | $104_{+10}$ | $89_{-5}$ | $88_{-6}$ |
| Llama 4 Scout 17B 16e Instruct (Meta AI, 2025) | 53% | 91 | $96_{+5}$ | $101_{+10}$ | $88_{-3}$ | $87_{-4}$ |
| Gemma 3 1B Instruct (Gemma Team et al., 2025) | 26% | 116 | $129_{+13}$ | $129_{+13}$ | $117_{+1}$ | $111_{-5}$ |
| Gemma 3 4B Instruct (Gemma Team et al., 2025) | 52% | 108 | $124_{+16}$ | $128_{+20}$ | $101_{-7}$ | $100_{-8}$ |
| Gemma 3 12B Instruct (Gemma Team et al., 2025) | 59% | 105 | $116_{+11}$ | $121_{+16}$ | $102_{-3}$ | $101_{-4}$ |
| Gemma 3 27B Instruct (Gemma Team et al., 2025) | 71% | 100 | $113_{+13}$ | $120_{+20}$ | $97_{-3}$ | $96_{-4}$ |
| Generation time (seconds) | | 1.56 | 1.59 | 2.48 | 3.65 | 4.50 |
| Length (characters) | | 104.79 | 195.12 | 303.09 | 174.70 | 219.22 |

Table 5: SelfReflect score $\downarrow$ ($\times 10^{-3}$) of RLVR models averaged over TriviaQA, NQ & SimpleQA. *Greedy* is generated w/o reasoning. *Basic* and *Sample & Summarize* reason and output a summary.

| Model | Single-decoding methods | | Sample & Summarize | |
|---|---|---|---|---|
| | Greedy | Reasoning | $N=10$ | $N=20$ |
| QwQ 32B (Qwen Team, 2025b) | 96 | $105_{+9}$ | $91_{-5}$ | $90_{-6}$ |
| DeepSeek R1 Distill Qwen 2.5 32B (DeepSeek-AI et al., 2025) | 96 | $108_{+12}$ | $91_{-5}$ | $90_{-6}$ |
| Qwen3 32B (Reasoning enabled) (Qwen Team, 2025a) | 93 | $96_{+3}$ | $86_{-7}$ | $85_{-8}$ |
| Qwen3 8B (Reasoning enabled) (Qwen Team, 2025a) | 103 | $104_{+1}$ | $90_{-13}$ | $89_{-14}$ |
| Generation time (seconds) | 1.96 | 3.60 | 6.99 | 8.57 |
| Length (characters) | 107.56 | 224.98 | 287.31 | 350.98 |

negative result. Reasoning models do not perform any better. Qualitatively, summaries produced by reasoning models are similar to the instruction-tuned LLMs with *CoT* prompts: they list possibilities, as prompted, but these possibilities are not faithful to the LLM's actual internal distribution.

Last, we attempt to explicitly train LLMs to output self-summaries. We take a dataset of 10,000 *Sample & summarize* summaries from TriviaQA or Natural Questions (SimpleQA is too small) as good examples and perform supervised finetuning (SFT) and/or direct preference optimization (DPO, against 10,000 greedy answers as negative examples) on a Qwen 3 8B non-reasoning model. Appendix L shows that SFT reduces the SelfReflect score on the train data but neither on held-out validation questions from the same dataset nor on out-of-domain questions from the other dataset. This suggests that the model memorizes individual summaries rather than learning a general mechanism for accessing and summarizing its internal distribution. These experiments show that generating self-reflective summaries that are faithful to the model's internal uncertainty is a non-trivial new challenge.

### 5.3 IF IT IS NOT FAITHFUL, THEN WHICH ANSWERS DOES CHAIN-OF-THOUGHTS LIST?

To understand our resounding negative result further, we compare summaries and the true internal distributions. As example case, we use *CoT* summaries from Qwen2.5 72B Instruct, our largest model.

We first test whether *CoT* correctly captures the *spread* of the answer distribution, i.e., whether it focuses on a single answer when the true distribution is unimodal and includes multiple options when the true distribution is multimodal. We let Gemini 2.0 Flash classify whether the *CoT* summaries and $a^{(1:N)}$ are certain (only mentioning one answer option) or uncertain (mentioning semantically different options, see also Appendix K). Figure 4 shows that for 36% of the questions, its summary is uncertain even when the answer distribution samples are not, meaning it suggests multiple answers

options that do not have high probability under the true distribution. The same holds in reverse; in fact, the cross-table reveals that *CoT* generates certain or uncertain summaries nearly independently of whether the model's internal distribution is actually certain or uncertain.

Second, we investigate the possibilities mentioned in a summary. The most important possibility to cover is the ground-truth answer, so we use it as an anchor for this analysis. Following the best practices of Santilli et al. (2025), we measure the RougeL-Recall on Natural Questions' short answers, i.e., the longest substring of the true answer that appears in a summary, as percentage of the true answer's length. We find that *Greedy* answers have an average overlap of 59.5% with the true answers. *Basic* summaries have 62.0%, *CoT* summaries 64.0%, and *Sample & Summarize* summaries 65.6%. Evaluating with an LM Judge instead of RougeL-Recall shows the same trend, rating that 71.3%, 72.2%, 74.1%, and 76.0% of the summaries include the true answer. In other words, summaries of the LLM's internal distributions are going in a promising direction in that they cover the true answer more often, they are just not faithful to the model's internal uncertainties (yet).

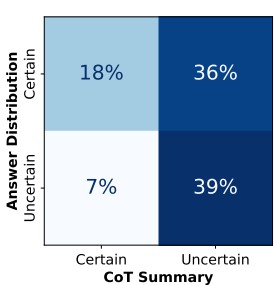

Figure 4: How often Qwen2.5 72B Instruct answer distributions span multiple possibilities vs how often their CoT summaries do.

## 6 OUTLOOK

We present SelfReflect, a metric that judges how faithfully a single string represents a distribution over output strings. SelfReflect is intended to guide the field on a new avenue of expressing uncertainties: Developing methods to make LLMs honestly describe all possible answers to a prompt in one string. We have seen in our benchmark that this is a hard task, but a solution to this problem would be a fundamental building block in many applications: It provides a human-interpretable account of LLM uncertainty, which can be useful in building appropriate trust in the LLM's outputs. The string can also be fed back to the LLM, for example to reason about follow-up questions when a user query is ambiguous. Listing all output possibilities is also a core necessity for conformal approaches, which are popular for classification but less explored for LLMs where the span of possible outputs is not immediately available. Finally, an accurate description of a distribution can also be recast into a numeric uncertainty value, thus generalizing traditional numeric and verbalized uncertainties.

## 7 LIMITATIONS AND DESIGN CHOICES OF THE SELFREFLECT SCORE

To outline the limitations of our work, we first note that 1-Wasserstein-based SelfReflect scores are not directly interpretable without baselines. A simplified version, like the percentage of equal top-predicted words using either summary or answer samples, would give more standardized values in [0,1]. However, we found that such an approach is less sensitive to differences in good vs *almost*-good summaries, rendering it less useful as a benchmark metric.

Second, seen from a summarization literature perspective, our SelfReflect metric intends to capture whether a summary faithfully represents the information of the model distribution. It does not intend to capture how short a summary is, so that concatenating 50 i.i.d. sampled answers as an adversarial summary would probably optimize the SelfReflect score without being a useful summary. From summarization literature we know that this is an orthogonal aspect that is better captured in a second metric like the summary length, so we appeal to always report summary lengths and qualitative samples along with the SelfReflect metric, as we do in this paper's results tables.

Third, we repeat that the faithfulness we measure is with respect to an LLM's *subjective* uncertainty. We intentionally did not develop SelfReflect to quantify objective truthfulness, with the outlook that larger LLMs approximate their training datasets better and better, such that more faithful summaries of subjective uncertainties will ultimately lead to better objective uncertainties.

REPRODUCIBILITY STATEMENT

We intend to lay a foundation for a new avenue of communicating uncertainties with our work, and enable future researchers to contribute to it. Thus, we publish code to compute SelfReflect scores for arbitrary LLMs and summary-generating methods to enable standardized benchmarking. Besides code, we have added all prompts used throughout our experiments in the appendix, as well as all hyperparameters, and exemplary SelfReflect computations broken down to the word-level.

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

APPENDIX CONTENTS

# A SELFREFLECT AND PREDICTIVE SUFFICIENCY: PROPOSITIONS AND PROOFS

In this appendix, we provide details of the propositions from the main text and their proofs. We begin with the definition of predictive sufficiency and provide a proof of its two equivalent characterizations in the context of the SelfReflect metric. We then prove an equivalence between solving the masked-token prediction task of the SelfReflect metric and the desired predictive sufficiency of the summary, providing a theoretical foundation for the design of the SelfReflect metric.

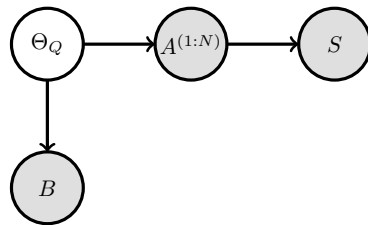

Figure 5: The graphical model for the setting of SelfReflect metric. The figure is reproduced from Figure 2 of the main text for the sake of better readability of the formalization that follows.

## A.1 SETUP, NOTATIONS, AND ASSUMPTIONS

Recall that prompting a given LLM with question $Q$ puts it in state $\Theta_Q$, from which we sample $N$ answers $A^{(1:N)}$. A summarization mechanism function $\psi$ generates the summary of these answers as $S = \psi(A^{(1:N)})$. For developing the SelfReflect metric, we generate another sample $B$ from the same state $\Theta_Q$ and require an ideal summary $S$ to capture all the information about $B$ that is captured by the samples $A^{(1:N)}$. Now, we formalize this setup of the SelfReflect metric by setting the notation, listing the assumptions of the setup, and providing their justifications.

SETUP AND NOTATION

1. Firstly, Figure 2 shows the graphical model of this setup, which we also reproduce here in Figure 5 for better readability. In this graphical model, observed variables are shaded gray, which includes the sampled answers $A^{(1:N)}$, their summary $S$, and a subsequent answer $B$, whereas unobserved/latent variables are unshaded, which includes the LLM state $\Theta_Q$.

2. We will use upper-case non-boldface letters (like $B$ or $S$) to represent random variables/vectors and the corresponding lower-case non-boldface letters (like $b$ or $s$) to represent particular samples from their underlying distributions.

3. For a random variable $Y$, the sampling of a particular value $y$ will be denoted as $y \sim Y$ or $y \in \text{supp}(Y)$, where $\text{supp}(Y)$ represents the support of the random variable $Y$.

4. Let $\mathbf{V}$ denote a finite vocabulary of words (or tokens), which is used to generate questions, the corresponding answers, and their summaries.

5. Let $Q$ denote the random variable for a question.

6. Prompting the given LLM with this question $Q$ is assumed to put it in state, which is represented with the random variable $\Theta_Q$. From this state, we can sample multiple answers, which are then used to define the SelfReflect metric.

7. The random variables $A^{(1:N)} := (A^{(1)}, \cdots, A^{(N)})$ are used to denote the $N$ answers sampled from the LLM in state $\Theta_Q$. These samples may be sampled in an i.i.d. manner but we do not necessitate this. In fact, one can sample each answer $A^{(n)}$ conditioned on all previous samples $A^{(1:n-1)}$ as well. We allow for this generality because throughout our derivation, we will always consider these answers jointly as $A^{(1:N)}$.

8. A summarization mechanism inputs the sampled answers and generates their summary $S$.

9. Suppose $B$ denote a subsequent sample from the LLM in the same state $\Theta_Q$. For the SelfReflect metric, we require an idea summary $S$ of sampled answers $A^{(1:N)}$ to capture all information about this subsequent answer $B$.

ASSUMPTIONS

1. The support of question $Q$ is assumed to be the set of all finite-length sentences generated from $\mathbf{V}$, which we denote by $\mathcal{X}$.

2. The support of each $A^{(i)}$ is also assumed to be $\mathcal{X}$, the set of all finite-length sentences generated from vocabulary $\mathbf{V}$.

3. The summarization mechanism that inputs the sampled answers $A^{(1:N)}$ and generates their summary $S$ is assumed to be a function $\psi$. Formally, $\psi : \mathcal{X}^N \longrightarrow \mathcal{X}$ inputs any $N$ sampled answers $A^{(1:N)}$ from the LLM and generates their summary $S$ as $S := \psi(A^{(1:N)})$. Note that the support of the summary $S$, will be a subset of the set of all finite-length sentences, i.e., $\mathrm{supp}\,(S) \subseteq \mathcal{X}$. This condition models our setup sufficiently well, where we have a candidate summary $S$ per set of answers $A^{(1:N)}$. However, we acknowledge that it is a restrictive condition in that it doesn't allow for modeling a conditional distribution over all summaries given the answers. Generalizing our SelfReflect metric for this case or proving its generality in this case is an interesting direction for future work.

4. We define the support of the subsequent new answer $B$ to be the set $\mathcal{X}_L := \mathbf{V}^L$ of all possible sentences from the vocabulary $\mathbf{V}$ that are of length $L$. Despite being slightly restrictive, this assumption is not unreasonable; all LLMs have a maximum context length, which can be viewed as an upper limit on the length of the answer $B$. Also, sentences with smaller lengths are usually padded to achieve the maximum context length.

5. Throughout our derivations, we will assume all required marginal and conditional distributions to be strictly positive. This assumption is reasonable for our setting because in practice, we would be implementing corresponding distributions using the given LLM. For instance, $p\,(W)$ would represent the probability of sentence $W$ under the given LLM. Further, $p\,(Y \mid Z)$ would represent the probability of sentence $Y$ when the LLM is prompted with the context $Z$. Since the LLMs generate distribution over the entire vocabulary $\mathbf{V}$, all the conditional distributions will have strictly positive values, albeit extremely small in certain cases.

## A.2 PREDICTIVE SUFFICIENCY AND EQUIVALENT CHARACTERIZATIONS

Now, having set the notations and assumptions, we define the notion of sufficiency and connect it with the definition of an ideal summary.

**Definition A.1** (Bayesian and Predictive Sufficiency (Bernardo and Smith, 2009)). *Consider a distribution parameterized in terms of a parameter $\phi$. Let $X^{(1:M)}$ denote $M$ (i.i.d.) samples from this distribution. A statistic (function) $T(X^{1:M})$ is called a **Bayesian sufficient statistic** of samples $X^{(1:M)}$ for $\phi$ if and only if we have: $p\left(\phi \mid X^{(1:M)} = x^{(1:M)}\right) = p\left(\phi \mid T(X^{(1:M)}) = t(x^{(1:M)})\right)$. On the other hand, it is called a **predictive sufficient statistic** of samples $X^{(1:M)}$ if and only if we have: $p\left(X = x \mid X^{(1:M)} = x^{(1:M)}\right) = p\left(X = x \mid T(X^{(1:M)}) = t(x^{(1:M)})\right)$ for any subsequent sample $X$ (with concrete value $x \in \mathrm{supp}\,(X)$) from the same distribution.*

Note that our Definition 3.1 of an ideal summary is closely related to predictive sufficiency as defined in Definition A.1. However, it turns out that Bayesian and predictive sufficiency notions are not exactly equivalent. In light of this, our reason for defining an ideal summary to be predictive sufficient, rather than Bayesian sufficient, is as follows. An LLM trained on a huge corpus of data contains information about a wide array of aspects. However, through the summary, we are interested in capturing only those aspects of the state $\Theta_Q$ of the LLM that are related to answering the given question $Q$. For this, requiring the summary to be predictive sufficient serves the purpose precisely.

Now, in the context of the Definition A.1 of predictive sufficiency, Definition 3.1 of ideal summary, and the graphical model of Figure 5, we prove Proposition 3.1, which asserts the equivalence in the information theoretic and conditional distribution based formulations of the ideal summary. We begin by proving a lemma about the graphical model of Figure 5.

**Lemma A.1** (Conditioning on $A^{(1:N)}$ and $S$). *Under the graphical model given in Figure 5, we have:*

$$p\left(B \mid A^{(1:N)}, S\right) = p\left(B \mid A^{(1:N)}\right)$$

*Proof.* Consider the following manipulations:

$$
\begin{aligned}
p\left(B \mid A^{(1:N)}, S\right) &=^{(1)} \int_\theta d\theta \, p\left(B, \Theta_Q = \theta \mid A^{(1:N)}, S\right) \\
&=^{(2)} \int_\theta d\theta \, \frac{p\left(\Theta_Q = \theta, B, A^{(1:N)}, S\right)}{p\left(A^{(1:N)}, S\right)} \\
&=^{(3)} \int_\theta d\theta \, \frac{p\left(\Theta_Q = \theta\right) \cdot p\left(B \mid \Theta_Q = \theta\right) \cdot p\left(A^{(1:N)} \mid \Theta_Q = \theta\right) \cdot p\left(S \mid A^{(1:N)}\right)}{p\left(A^{(1:N)}\right) \cdot p\left(S \mid A^{(1:N)}\right)} \\
&=^{(4)} \int_\theta d\theta \, \frac{p\left(\Theta_Q = \theta\right) \cdot p\left(B \mid \Theta_Q = \theta\right) \cdot p\left(A^{(1:N)} \mid \Theta_Q = \theta\right)}{p\left(A^{(1:N)}\right)} \\
&=^{(5)} \int_\theta d\theta \, \frac{p\left(\Theta_Q = \theta, B, A^{(1:N)}\right)}{p\left(A^{(1:N)}\right)} \\
&=^{(6)} \int_\theta d\theta \, p\left(B, \Theta_Q = \theta \mid A^{(1:N)}\right) =^{(7)} p\left(B \mid A^{(1:N)}\right)
\end{aligned}
\tag{5}
$$

Here, steps $(2), (5), (6)$ follow from chain rule. Step $(4)$ follows by cancellation of the common terms. Steps $(1), (7)$ follows from integrating out variable $\Theta_Q$. Step $(3)$ follows from the graphical model of Figure 5. Finally, an analogous derivation would follow by replacing integration with summation in the case of $\Theta_Q$ being a discrete variable. $\qquad\square$

Now, we prove Proposition 3.1 establishing the equivalence of the information theoretic and conditional distribution based formulations of the desired predictive sufficiency.

**Theorem A.1** (Connection of SelfReflect to Predictive Sufficiency). *Consider the graphical model given in Figure 5. Under this graphical model, for ideal summary $S$ of answers $A^{(1:N)}$,*

$$
\mathcal{I}\left\{A^{(1:N)}; B\right\} = \mathcal{I}\left\{S; B\right\} \iff p\left(B \mid A^{(1:N)}\right) = p\left(B \mid S\right)
$$

*Proof.* Consider following steps:

$$
\begin{aligned}
\mathcal{I}\left\{A^{(1:N)}; B\right\} = \mathcal{I}\left\{S; B\right\} &\iff^{(1)} \mathbb{E}_{A^{(1:N)}, B}\left[\log \frac{p\left(A^{(1:N)}, B\right)}{p\left(A^{(1:N)}\right) \cdot p(B)}\right] = \mathbb{E}_{S, B}\left[\log \frac{p(S, B)}{p(S) \cdot p(B)}\right] \\
&\iff \mathbb{E}_{B, A^{(1:N)}, S}\left[\log \frac{p\left(A^{(1:N)}, B\right) \cdot p(S)}{p(S, B) \cdot p\left(A^{(1:N)}\right)}\right] = 0 \iff^{(2)} \mathbb{E}_{B, A^{(1:N)}, S}\left[\log \frac{p\left(B \mid A^{(1:N)}\right)}{p(B \mid S)}\right] = 0 \\
&\iff^{(3)} \mathbb{E}_{B, A^{(1:N)}, S}\left[\log \frac{p\left(B \mid A^{(1:N)}, S\right)}{p(B \mid S)}\right] = 0 \iff^{(4)} \mathcal{I}\left\{A; A^{(1:N)} \mid S\right\} = 0 \\
&\iff^{(5)} p\left(B, A^{(1:N)} \mid S\right) = p(B \mid S) \cdot p\left(A^{(1:N)} \mid S\right) \\
&\iff^{(6)} p\left(B \mid A^{(1:N)}, S\right) = p(B \mid S) \iff^{(7)} p\left(B \mid A^{(1:N)}\right) = p(B \mid S)
\end{aligned}
\tag{6}
$$

Here, step $(1)$ follows from the definition of mutual information, steps $(2)$ and $(6)$ from chain rule, steps $(3)$ and $(7)$ from Lemma A.1, step $(4)$ from the definition of conditional mutual information, and step $(5)$ from the equality condition of conditional mutual information. For details on mutual information and conditional mutual information, we refer the reader to Cover (1999). $\qquad\square$

### A.3 SELFREFLECT METRIC AND EQUIVALENCE TO PREDICTIVE SUFFICIENCY

Now, we demonstrate that the masked-token prediction task of SelfReflect is equivalent to the above notion of predictive sufficiency. For the SelfReflect metric, we consider the random variable $B$ for a new subsequent sample from the LLM in state $\Theta_Q$ and dissect it in terms of its words. In particular, we have: $B \equiv (B_1, \cdots, B_L)$, where $L$ is length of the sentence $B$ (which, as we saw, could be chosen to be the maximum context length for the LLM). Here, $B_i$ represents the random variable for the $i-$th word of the sentence $B$ for each value of $i \in \{1, \cdots, L\}$. For each $i$, we use

the shorthand notation $B_{-i}$ to represent the variable for all the words in the sentence $B$ except for $B_i$, i.e., $B_{-i} := (B_1, \cdots, B_{i-1}, B_{i+1}, \cdots, B_L) = (B_j)_{j \neq i}$. Note that $B_\ell$, which represents the $\ell$−th word of sentence $B$, is not to be confused with $A^{(k)}$, which represents the $k$−th sampled answer from the LLM. For each $B_i$, its support is going to be the vocabulary $\mathbf{V}$ and the supports of $B_{-i}$ and $B$ are $\mathbf{V}^{L-1}$ and $\mathbf{V}^L \equiv \mathcal{X}_L$ respectively. With this setup, we can prove Proposition 3.2, which asserts that under assumptions from subsection A.1, SelfReflect metric provides an equivalent formulation of the desired predictive sufficiency of ideal summary $S$. This is done as follows.

**Theorem A.2** (SelfReflect Metric and Predictive Sufficiency). *Suppose all involved conditionals are modeled via the given LLM and hence, are strictly positive. Then, we have:*

$$p\left(B \mid A^{(1:N)}\right) = p\left(B \mid S\right) \iff \text{for all masking indices } i, \ p\left(B_i \mid A^{(1:N)}, B_{-i}\right) = p\left(B_i \mid S, B_{-i}\right) \quad (7)$$

*Proof.* ($\Longrightarrow$) Suppose we are given that $p\left(B \mid A^{(1:N)}\right) = p\left(B \mid S\right)$. Consider the following steps:

$$p\left(B \mid A^{(1:N)}\right) = p\left(B \mid S\right) \Longrightarrow p\left(B_1, \cdots, B_L \mid A^{(1:N)}\right) = p\left(B_1, \cdots, B_L \mid S\right)$$

$$\Longrightarrow \sum_{b_i \in \mathbf{V}} p\left(B_1, \cdots, B_i = b_i, \cdots, B_L \mid A^{(1:N)}\right) = \sum_{b_i \in \mathbf{V}} p\left(B_1, \cdots, B_i = b_i, \cdots, B_L \mid S\right)$$

$$\Longrightarrow^{(1)} p\left(B_{-i} \mid A^{(1:N)}\right) = p\left(B_{-i} \mid S\right) \tag{8}$$

Here, step $(1)$ follows from integrating out variable $B_i$. Combining this result with the premise gives:

$$p\left(B \mid A^{(1:N)}\right) = p\left(B \mid S\right), \ p\left(B_{-i} \mid A^{(1:N)}\right) = p\left(B_{-i} \mid S\right)$$

$$\Longrightarrow \frac{p\left(B \mid A^{(1:N)}\right)}{p\left(B_{-i} \mid A^{(1:N)}\right)} = \frac{p\left(B \mid S\right)}{p\left(B_{-i} \mid S\right)} \Longrightarrow^{(1)} p\left(B_i \mid A^{(1:N)}, B_{-i}\right) = p\left(B_i \mid S, B_{-i}\right) \tag{9}$$

Here, step $(1)$ follows because $B$ is formed of the $i$−th word $B_i$ and the rest of the words $B_{-i}$. Since we can carry out these steps for any index $i$, we prove the forward direction of the theorem.

($\Longleftarrow$) Now, to prove the converse, suppose we are given that for all masking indices $i$, we have: $p\left(B_i \mid A^{(1:N)}, B_{-i}\right) = p\left(B_i \mid S, B_{-i}\right)$ and we have to prove that $p\left(B \mid A^{(1:N)}\right) = p\left(B \mid S\right)$. Since this is an equality of the random variables, we prove the equality of random variables by proving it for any and all choices of the samples of those random variables. Note that this works because of the assumption of summary mechanism $S$ being a function of $A^{(1:N)}$, which allows us to use the given condition as well as prove the desired result by assuming particular instantiations of $A^{(1:N)} = \bar{a}^{(1:N)}$ and using the corresponding summary $S = \bar{s} := \psi(\bar{a}^{(1:N)})$. Pick any instantiations of sampled answers from their support as $a^{(1:N)} \sim A^{(1:N)}$. Since the summary mechanism is a function, it gives us a concrete sample $s = \psi(a^{(1:N)}) \in \mathcal{X}$. Now, suppose we want to prove the desired result for any particular given sample $b \sim B$ with $b := (b_1, \cdots, b_L) \in \mathbf{V}^L$. Consider a fixed sentence $b^* \in \mathbf{V}^L$ with $b^* := (b_1^*, \cdots, b_L^*)$. Now, we define a sequence of sentences as follows:

$$\begin{aligned} x^{(0)} &:= (b_1, b_2, \cdots, b_L) = b \in \mathbf{V}^L \\ x^{(1)} &:= (b_1^*, b_2, \cdots, b_L) \in \mathbf{V}^L \\ x^{(2)} &:= (b_1^*, b_2^*, \cdots, b_L) \in \mathbf{V}^L \\ &\vdots \\ x^{(L)} &:= (b_1^*, b_2^*, \cdots, b_L^*) = b^* \in \mathbf{V}^L \end{aligned} \tag{10}$$

Intuitively, we create a sequence of sentences where each subsequent sentence $x^{(i)}$ differs from the previous sentence and the next sentence in exactly one word and as we go from sentence $x^{(0)}$ to $x^{(L)}$, we change the given sentence $b$ to the fixed sentence $b^*$. Now, we consider the following

manipulations for $p\left(B = b \mid A^{(1:N)} = a^{(1:N)}\right)$:

$$p\left(B = b \mid A^{(1:N)} = a^{(1:N)}\right) = p\left(B = x^{(0)} \mid A^{(1:N)} = a^{(1:N)}\right)$$

$$=^{(1)} p\left(B = x^{(0)} \mid A^{(1:N)} = a^{(1:N)}\right) \cdot \prod_{\ell=1}^{L} \frac{p\left(B = x^{(\ell)} \mid A^{(1:N)} = a^{(1:N)}\right)}{p\left(B = x^{(\ell)} \mid A^{(1:N)} = a^{(1:N)}\right)}$$

$$=^{(2)} \left(\prod_{\ell=1}^{L} \frac{p\left(B = x^{(\ell-1)} \mid A^{(1:N)} = a^{(1:N)}\right)}{p\left(B = x^{(\ell)} \mid A^{(1:N)} = a^{(1:N)}\right)}\right) \cdot p\left(B = b^* \mid A^{(1:N)} = a^{(1:N)}\right) \quad (11)$$

In an exactly analogous way, we get following manipulations for $p\left(B = b \mid S = s\right)$:

$$p\left(B = b \mid S = s\right) = p\left(B = x^{(0)} \mid S = s\right)$$

$$=^{(1)} p\left(B = x^{(0)} \mid S = s\right) \cdot \prod_{\ell=1}^{L} \frac{p\left(B = x^{(\ell)} \mid S = s\right)}{p\left(B = x^{(\ell)} \mid S = s\right)}$$

$$=^{(2)} \left(\prod_{\ell=1}^{L} \frac{p\left(B = x^{(\ell-1)} \mid S = s\right)}{p\left(B = x^{(\ell)} \mid S = s\right)}\right) \cdot p\left(B = b^* \mid S = s\right) \quad (12)$$

Note that in both Equation 11 and Equation 12 above, step $(1)$ follows from multiplying and dividing by the same terms and step $(2)$ follows from rearranging the terms and recognizing $x^{(L)} = b^*$ by definition. Now, we consider the $\ell-$th term from the Equation 11 and simplify it as follows:

$$\frac{p\left(B = x^{(\ell-1)} \mid A^{(1:N)} = a^{(1:N)}\right)}{p\left(B = x^{(\ell)} \mid A^{(1:N)} = a^{(1:N)}\right)}$$

$$=^{(1)} \frac{p\left(B_1 = b_1^*, \cdots, B_{\ell-1} = b_{\ell-1}^*, B_\ell = b_\ell, B_{\ell+1} = b_{\ell+1}, \cdots, B_L = b_L \mid A^{(1:N)} = a^{(1:N)}\right)}{p\left(B_1 = b_1^*, \cdots, B_{\ell-1} = b_{\ell-1}^*, B_\ell = b_\ell^*, B_{\ell+1} = b_{\ell+1}, \cdots, B_L = b_L \mid A^{(1:N)} = a^{(1:N)}\right)}$$

$$=^{(2)} \frac{p\left(B_{-\ell} = (b_1^*, \cdots, b_{\ell-1}^*, b_{\ell+1}, \cdots, b_L) \mid A^{(1:N)} = a^{(1:N)}\right)}{p\left(B_{-\ell} = (b_1^*, \cdots, b_{\ell-1}^*, b_{\ell+1}, \cdots, b_L) \mid A^{(1:N)} = a^{(1:N)}\right)}$$

$$\times \frac{p\left(B_\ell = b_\ell \mid A^{(1:N)} = a^{(1:N)}, B_{-\ell} = (b_1^*, \cdots, b_{\ell-1}^*, b_{\ell+1}, \cdots, b_L)\right)}{p\left(B_\ell = b_\ell^* \mid A^{(1:N)} = a^{(1:N)}, B_{-\ell} = (b_1^*, \cdots, b_{\ell-1}^*, b_{\ell+1}, \cdots, b_L)\right)}$$

$$=^{(3)} \frac{p\left(B_\ell = b_\ell \mid A^{(1:N)} = a^{(1:N)}, B_{-\ell} = (b_1^*, \cdots, b_{\ell-1}^*, b_{\ell+1}, \cdots, b_L)\right)}{p\left(B_\ell = b_\ell^* \mid A^{(1:N)} = a^{(1:N)}, B_{-\ell} = (b_1^*, \cdots, b_{\ell-1}^*, b_{\ell+1}, \cdots, b_L)\right)} \quad (13)$$

Again, in an exactly analogous way, we simplify the $\ell-$th terms of Equation 12 as follows:

$$\frac{p\left(B = x^{(\ell-1)} \mid S = s\right)}{p\left(B = x^{(\ell)} \mid S = s\right)}$$

$$=^{(1)} \frac{p\left(B_1 = b_1^*, \cdots, B_{\ell-1} = b_{\ell-1}^*, B_\ell = b_\ell, B_{\ell+1} = b_{\ell+1}, \cdots, B_L = b_L \mid S = s\right)}{p\left(B_1 = b_1^*, \cdots, B_{\ell-1} = b_{\ell-1}^*, B_\ell = b_\ell^*, B_{\ell+1} = b_{\ell+1}, \cdots, B_L = b_L \mid S = s\right)}$$

$$=^{(2)} \frac{p\left(B_{-\ell} = (b_1^*, \cdots, b_{\ell-1}^*, b_{\ell+1}, \cdots, b_L) \mid S = s\right)}{p\left(B_{-\ell} = (b_1^*, \cdots, b_{\ell-1}^*, b_{\ell+1}, \cdots, b_L) \mid S = s\right)}$$

$$\times \frac{p\left(B_\ell = b_\ell \mid S = s, B_{-\ell} = (b_1^*, \cdots, b_{\ell-1}^*, b_{\ell+1}, \cdots, b_L)\right)}{p\left(B_\ell = b_\ell^* \mid S = s, B_{-\ell} = (b_1^*, \cdots, b_{\ell-1}^*, b_{\ell+1}, \cdots, b_L)\right)}$$

$$=^{(3)} \frac{p\left(B_\ell = b_\ell \mid S = s, A_{-\ell} = (b_1^*, \cdots, b_{\ell-1}^*, b_{\ell+1}, \cdots, b_L)\right)}{p\left(B_\ell = b_\ell^* \mid S = s, B_{-\ell} = (b_1^*, \cdots, b_{\ell-1}^*, b_{\ell+1}, \cdots, b_L)\right)} \quad (14)$$

In both these simplifications, step $(1)$ follows from the definition of the sentences $x^{(\ell-1)}, x^{(\ell)}$, step $(2)$ follows from chain rule, and step $(3)$ follows from canceling the common terms. However, given

equality $p\left(B_i \mid A^{(1:N)}, B_{-i}\right) = p\left(B_i \mid S, B_{-i}\right)$ for all masking locations $i$ implies that for all $\ell$:

$$p\left(B_\ell = b_\ell \mid A^{(1:N)} = a^{(1:N)}, B_{-\ell} = (b_1^*, \cdots, b_{\ell-1}^*, b_{\ell+1}, \cdots, b_L)\right)$$
$$= p\left(B_\ell = b_\ell \mid S = s, B_{-\ell} = (b_1^*, \cdots, b_{\ell-1}^*, b_{\ell+1}, \cdots, b_L)\right) \text{ and} \tag{15}$$

$$p\left(B_\ell = b_\ell^* \mid A^{(1:N)} = a^{(1:N)}, B_{-\ell} = (b_1^*, \cdots, b_{\ell-1}^*, b_{\ell+1}, \cdots, b_L)\right)$$
$$= p\left(B_\ell = b_\ell^* \mid S = s, B_{-\ell} = (b_1^*, \cdots, b_{\ell-1}^*, b_{\ell+1}, \cdots, b_L)\right) \tag{16}$$

$$\implies \frac{p\left(B_\ell = b_\ell \mid A^{(1:N)} = a^{(1:N)}, B_{-\ell} = (b_1^*, \cdots, b_{\ell-1}^*, b_{\ell+1}, \cdots, b_L)\right)}{p\left(B_\ell = b_\ell^* \mid A^{(1:N)} = a^{(1:N)}, B_{-\ell} = (b_1^*, \cdots, b_{\ell-1}^*, b_{\ell+1}, \cdots, b_L)\right)}$$
$$= \frac{p\left(B_\ell = b_\ell \mid S = s, B_{-\ell} = (b_1^*, \cdots, b_{\ell-1}^*, b_{\ell+1}, \cdots, b_L)\right)}{p\left(B_\ell = b_\ell^* \mid S = s, B_{-\ell} = (b_1^*, \cdots, b_{\ell-1}^*, b_{\ell+1}, \cdots, b_L)\right)}$$

$$\implies \frac{p\left(B = x^{(\ell-1)} \mid A^{(1:N)} = a^{(1:N)}\right)}{p\left(B = x^{(\ell)} \mid A^{(1:N)} = a^{(1:N)}\right)} = \frac{p\left(B = x^{(\ell-1)} \mid S = s\right)}{p\left(B = x^{(\ell)} \mid S = s\right)} \text{ for all } \ell \in \{1, \cdots, L\}. \tag{17}$$

Combining this with Equation 11 and Equation 12, we get an interesting result:

$$\frac{p\left(B = b \mid A^{(1:N)} = a^{(1:N)}\right)}{p\left(B = b \mid S = s\right)}$$
$$= \frac{(\prod_{\ell=1}^{L} \frac{p\left(B = x^{(\ell-1)} \mid A^{(1:N)} = a^{(1:N)}\right)}{p\left(B = x^{(\ell)} \mid A^{(1:N)} = a^{(1:N)}\right)}) \cdot p\left(B = b^* \mid A^{(1:N)} = a^{(1:N)}\right)}{(\prod_{\ell=1}^{L} \frac{p\left(B = x^{(\ell-1)} \mid S = s\right)}{p\left(B = x^{(\ell)} \mid S = s\right)}) \cdot p\left(B = b^* \mid S = s\right)}$$
$$=^{(1)} \frac{p\left(B = b^* \mid A^{(1:N)} = a^{(1:N)}\right)}{p\left(B = b^* \mid S = s\right)} \tag{18}$$

Here, step $(1)$ follows from canceling equal terms in both the numerator and the denominator. What Equation 18 implies is that given $A^{(1:N)} = a^{(1:N)}$, thereby giving $S = s := \psi(a^{(1:N)})$, the ratio $\frac{p\left(B=b|A^{(1:N)}=a^{(1:N)}\right)}{p(B=b|S=s)}$ equals the ratio $\frac{p\left(B=b^*|A^{(1:N)}=a^{(1:N)}\right)}{p(B=b^*|S=s)}$ for any and all values of $b \in \mathbf{V}^L$, thereby making it a constant $c := c(a^{(1:N)})$ (a constant that dependents on $a^{(1:N)}$). Now, we can integrate out $B$ and obtain the value of this constant as follows:

$$\text{For all } b \in \mathbf{V}^L, \frac{p\left(B = b \mid A^{(1:N)} = a^{(1:N)}\right)}{p\left(B = b \mid S = s\right)} = c(a^{(1:N)})$$
$$\implies 1 = \sum_{b \in \mathbf{V}^L} p\left(B = b \mid A^{(1:N)} = a^{(1:N)}\right) = \sum_{b \in \mathbf{V}^L} c(a^{(1:N)}) \cdot p\left(B = b \mid S = s\right)$$
$$= c(a^{(1:N)}) \cdot \sum_{b \in \mathbf{V}^L} p\left(B = b \mid S = s\right) = c(a^{(1:N)}) \cdot 1 = c(a^{(1:N)}) \tag{19}$$

This proves that in fact $c(a^{(1:N)}) = 1$, which gives that for all $b \sim B$, we have: $p\left(B = b \mid A^{(1:N)} = a^{(1:N)}\right) = p\left(B = b \mid S = s\right)$. Since this result holds for all $b \sim B$, we can write the corresponding result with the underlying random variable as: $p\left(B \mid A^{(1:N)} = a^{(1:N)}\right) = p\left(B \mid S = s\right)$. However, since this result holds for any sample choice of $A^{(1:N)} = a^{(1:N)}$ (and corresponding $S = s := \psi(a^{(1:N)})$), we get the desired results involving all underlying random variables: $p\left(B \mid A^{(1:N)}\right) = p\left(B \mid S\right)$. This proves the reverse direction of the equivalence. $\square$

## A.4 MODELING WITH LLM: FROM DERIVATION TO IMPLEMENTATION

Now, having proved the equivalence of the basis of the SelfReflect metric and the desired predictive sufficiency of summary, we show the connection with the exact definition of the SelfReflect metric. Suppose we are given with a question $Q = q \in \mathcal{X}$, which is shown to an LLM labeled $\text{LLM}_\theta$. This puts $\text{LLM}_\theta$ in a state $\Theta_Q = \theta_q$, from which we sample answers $A^{(1:N)} = a^{(1:N)}$, and a subsequent sample $B = b \in \mathbf{V}^L$. Now, to calculate the SelfReflect metric, the core idea is that conditional distributions of the form $p\left(Y \mid Z\right)$ involved in the theoretical considerations above are modeled by

prompting the judge $\text{LLM}_J$ with context $Z$ and checking the probability of $Y$. In our implementation, this $\text{LLM}_J$ will be temperature-scaled with temperature $\tau = 5$ as mentioned in the main text in order to flatten its distribution and make it consider more synonyms. Then, we build the prompt of $\text{LLM}_J$ by including the question $Q = q$ and either the samples $A^{(1:N)} = a^{(1:N)}$ or their summary $S = s := \psi(a^{(1:N)})$, along with a description $t$ of the masked-token prediction task to tell the $\text{LLM}_J$ judge what it needs to do. We then mask each word of $B = b$ one by one to obtain the masked word $B_m = b_m \in \mathbf{V}$ and the rest of the sentence $B_{-m} = b_{-m} \in \mathbf{V}^{L-1}$. Then, we model the required conditional distributions that appear in the derivation using the $\text{LLM}_J$ judge as follows:

$$p\left(B_m = b_m \mid A^{(1:N)} = a^{(1:N)}, B_{-m} = b_{-m}\right)$$

$$:= p_{\text{LLM}_J}\left(B_m = b_m \mid Q = q, A^{(1:N)} = a^{(1:N)}, t, B_{-m} = b_{-m}\right), \text{ and} \qquad (20)$$

$$p\left(B_m = b_m \mid S = s, B_{-m} = b_{-m}\right)$$

$$:= p_{\text{LLM}_J}\left(B_m = b_m \mid Q = q, S = s, t, B_{-m} = b_{-m}\right)$$

This modeling along with Theorem A.2 demonstrates the efficacy of SelfReflect metric:

**Corollary A.1** (Efficacy of SelfReflect Metric). *For any question $Q$, for all masking indices $m$,*

$$\mathcal{W}^1(p_{LLM_J}\left(B_m \mid Q, A^{(1:N)}, t, B_{-m}\right), p_{LLM_J}\left(B_m \mid Q, S, t, B_{-m}\right)) = 0$$

$$\Longleftrightarrow^{(1)} p_{LLM_J}\left(B_m \mid Q, A^{(1:N)}, t, B_{-m}\right) = p_{LLM_J}\left(B_m \mid Q, S, t, B_{-m}\right)$$

$$\Longleftrightarrow^{(2)} p\left(B_m \mid A^{(1:N)}, B_{-m}\right) = p\left(B_m \mid S, B_{-m}\right) \qquad (21)$$

$$\Longleftrightarrow^{(3)} p\left(B \mid A^{(1:N)}\right) = p\left(B \mid S\right)$$

$$\Longleftrightarrow^{(4)} \mathcal{I}\left\{A^{(1:N)}; B\right\} = \mathcal{I}\left\{S; B\right\}$$

*Proof.* Step (4) follows from Theorem A.1, step (3) follows from Theorem A.2, step (2) follows from modeling in Equation 20, and step (1) follows from the fact that the $\mathcal{W}^1$ ($1-$Wasserstein) distance between two distributions is 0 if and only if the distributions are identical. $\qquad \square$

DISCUSSION

We conclude this section by discussing two important points about our derivation.

1. Firstly, LLMs are known to behave significantly better with careful design of prompts (Sahoo et al., 2024). Thus, in our modeling of Equation 20, one may try to optimize the prompting template and the task description $t$ in order to further obtain sharper versions of the SelfReflect metric. In this aspect, note that our derivation does not provide a mechanism for optimizing for the prompt template or task description $t$. In fact, irrespective of this detail, the derivation holds true.

2. Secondly, we state the assumptions required for the derivation, as stated in Appendix A.1, are needed for establishing the connection of SelfReflect metric with the notion of predictive sufficiency. However, these are not needed for defining, implementing, or using the SelfReflect metric. Users may find our SelfReflect metric useful even in cases where one or more of the assumptions are loosened. Also, further generalizing the SelfReflect metric in cases where the assumptions are loosened or proving that the current formulation holds in those scenarios remains an interesting direction for future theoretical work.

# B  CONVERGENCE OF THE SELFREFLECT METRIC

## B.1  REDUCING BOTH $N$ AND $M$

In the main paper, we evaluate SelfReflect on $1000$ questions per dataset with $N = M = 50$ conditioning and masked-out answers. This is based on a convergence analysis that we present in this section. We use Qwen 2.5 72B Instruct and Natural Questions as an example and calculate the average SelfReflect score across an increasing number of questions and conditioning and masked-out answers in Figs. 6 to 10. The question is how many questions are needed to arrive at a stable average score.

It can be seen in Fig. 6 that at $N = M = 50$, the SelfReflect score converges at $1000$ questions, our setup for the paper. One can of course reduce $N$ and $M$, which will roughly linearly reduce the runtime required to compute the score. However, when for example reducing to $N = M = 20$ questions in Fig. 7, convergence to the final value sets in only at about $2500$ questions, which linearly increases the runtime, so that the runtime advantage vanishes. If one allows the score to be a bit less converged, for example in development rather than in reporting test results, we suggest to use $N = M = 10$ and 500 questions. This reduces the runtime to calculate SelfReflect to 9 minutes on a node with 8 A100 GPUs, compared to the 67 minutes of $N = M = 50$ and 1000 questions.

The only real outlier to these trends is $N = M = 1$. Here, it is especially important that $N = 1$, i.e., in the context of the answer distribution prompt, there is only a single response. In this case, the ideal summary is actually to return exactly this response rather than a summary of the distribution. Hence, in Fig. 10, *Greedy* obtains a better SelfReflect score than *Sample & Summarize*. This underlines the importance of why SelfReflect uses *multiple* samples from the answer distribution in the context.

## B.2  REDUCING ONLY $M$

We can study this effect further by reducing only the number of answers that we compute masked-out tasks over, $M$, and keeping the number of reference distribution answers in the LM judge context constant at $N = 50$. The results, and a comparison to all baselines, is shown in Table 6. We observe that, similar as in the previous section, $M$ can be reduced down to $M = 5$ without losing much of the ability to detect fine-grained quality differences. At $M \in \{1, 2\}$, performance degrades slightly on good vs almost-good and percentage vs or-concatenated. But it remains above the baselines and unlike in the previous experiment, does not degrade in verbalized vs only majority answer. This confirms that the collapse in the previous experiment was due to lowering the number of answers in the reference distribution to $N = 1$, in which case the task becomes a top-1 matching task rather than a distribution matching task.

Table 6: We lower the compute spent to calculate the SelfReflect score by reducing the number of masked-out task answers $M$, keeping the number of reference distribution answers $N = 50$ constant. Mean $\pm$ 95% confidence interval. It can be seen that SelfReflect's performance stays roughly untouched from $N = 50$ down to $N = 5$, and stays above baselines even for $N \in \{1, 2\}$. Qwen 2.5 7B Instruct distributions over questions from the Natural Questions dataset.

| Metric | Good summaries vs bad summaries | Good vs almost-good | Detailed vs truncated | Verbalized uncertainty vs only majority answer | Verbalized vs or-concatenated | Percentage vs or-concatenated |
|---|---|---|---|---|---|---|
| Summarization | $97.40\%_{\pm 0.99\%}$ | $38.70\%_{\pm 3.02\%}$ | $53.55\%_{\pm 7.85\%}$ | $11.57\%_{\pm 5.70\%}$ | $57.02\%_{\pm 8.82\%}$ | $65.29\%_{\pm 8.48\%}$ |
| LM Judge | $98.33\%_{\pm 0.46\%}$ | $47.32\%_{\pm 1.91\%}$ | $59.92\%_{\pm 5.93\%}$ | $19.37\%_{\pm 5.60\%}$ | $34.55\%_{\pm 6.74\%}$ | $35.08\%_{\pm 6.77\%}$ |
| Opt. Transport | $80.16\%_{\pm 1.43\%}$ | $60.78\%_{\pm 1.87\%}$ | $39.69\%_{\pm 5.92\%}$ | $48.69\%_{\pm 7.09\%}$ | $52.88\%_{\pm 7.08\%}$ | $69.11\%_{\pm 6.55\%}$ |
| Embedding | $96.50\%_{\pm 0.66\%}$ | $65.49\%_{\pm 1.82\%}$ | $65.65\%_{\pm 5.75\%}$ | $10.99\%_{\pm 4.44\%}$ | $43.98\%_{\pm 7.04\%}$ | $36.65\%_{\pm 6.83\%}$ |
| SelfReflect $M = 1$ | $99.00\%_{\pm 0.62\%}$ | $94.23\%_{\pm 1.48\%}$ | $96.13\%_{\pm 3.04\%}$ | $84.30\%_{\pm 6.48\%}$ | $71.07\%_{\pm 8.08\%}$ | $78.51\%_{\pm 7.32\%}$ |
| SelfReflect $M = 2$ | $99.60\%_{\pm 0.39\%}$ | $96.12\%_{\pm 1.23\%}$ | $98.06\%_{\pm 2.17\%}$ | $90.91\%_{\pm 5.12\%}$ | $77.69\%_{\pm 7.42\%}$ | $73.55\%_{\pm 7.86\%}$ |
| SelfReflect $M = 5$ | $99.70\%_{\pm 0.34\%}$ | $97.90\%_{\pm 0.91\%}$ | $98.71\%_{\pm 1.78\%}$ | $90.91\%_{\pm 5.12\%}$ | $71.90\%_{\pm 8.01\%}$ | $80.99\%_{\pm 6.99\%}$ |
| SelfReflect $M = 10$ | $99.80\%_{\pm 0.28\%}$ | $98.22\%_{\pm 0.84\%}$ | $98.06\%_{\pm 2.17\%}$ | $92.56\%_{\pm 4.68\%}$ | $76.03\%_{\pm 7.61\%}$ | $82.64\%_{\pm 6.75\%}$ |
| SelfReflect $M = 20$ | $99.80\%_{\pm 0.28\%}$ | $98.64\%_{\pm 0.74\%}$ | $98.06\%_{\pm 2.17\%}$ | $95.04\%_{\pm 3.87\%}$ | $71.07\%_{\pm 8.08\%}$ | $84.30\%_{\pm 6.48\%}$ |
| SelfReflect $M = 50$ | $99.90\%_{\pm 0.28\%}$ | $98.74\%_{\pm 0.71\%}$ | $98.06\%_{\pm 2.17\%}$ | $95.04\%_{\pm 3.87\%}$ | $74.38\%_{\pm 7.78\%}$ | $83.47\%_{\pm 6.62\%}$ |

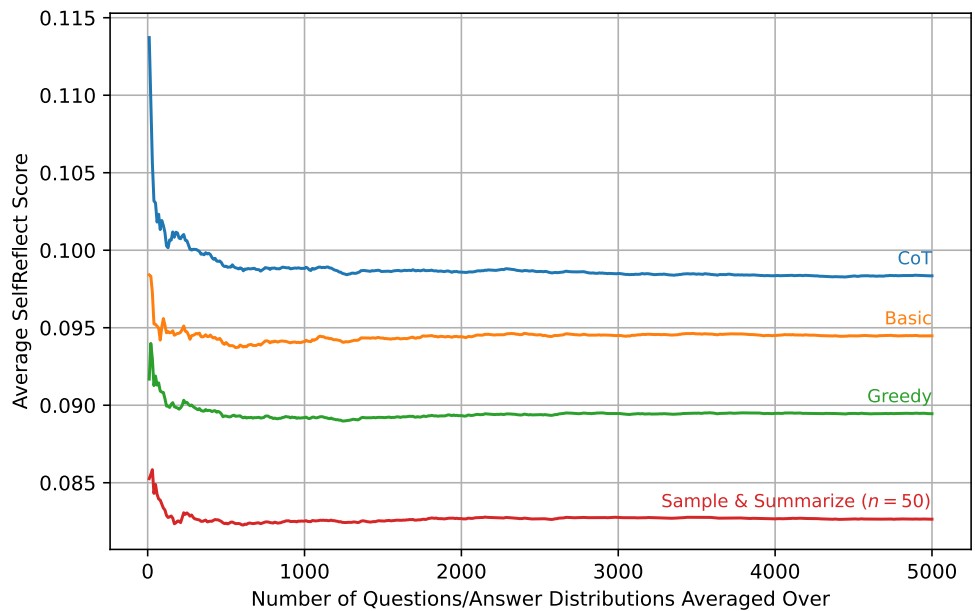

Figure 6: Convergence of the SelfReflect score with $N = M = 50$ and an increasing number of queries we evaluate on. Answer Distributions of Qwen 2.5 72B Instruct on Natural Questions.

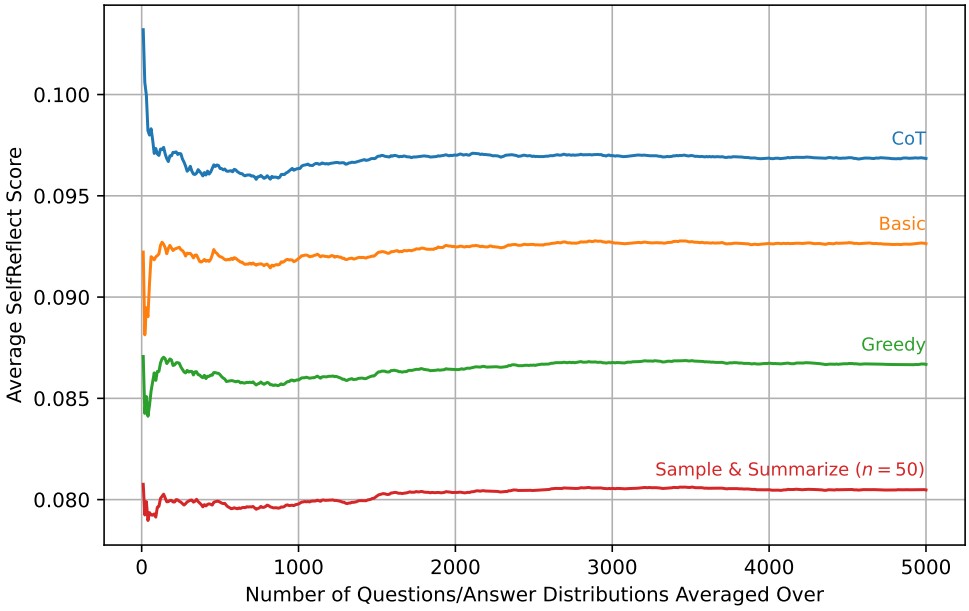

Figure 7: Convergence of the SelfReflect score with $N = M = 20$ and an increasing number of queries we evaluate on. Answer Distributions of Qwen 2.5 72B Instruct on Natural Questions.

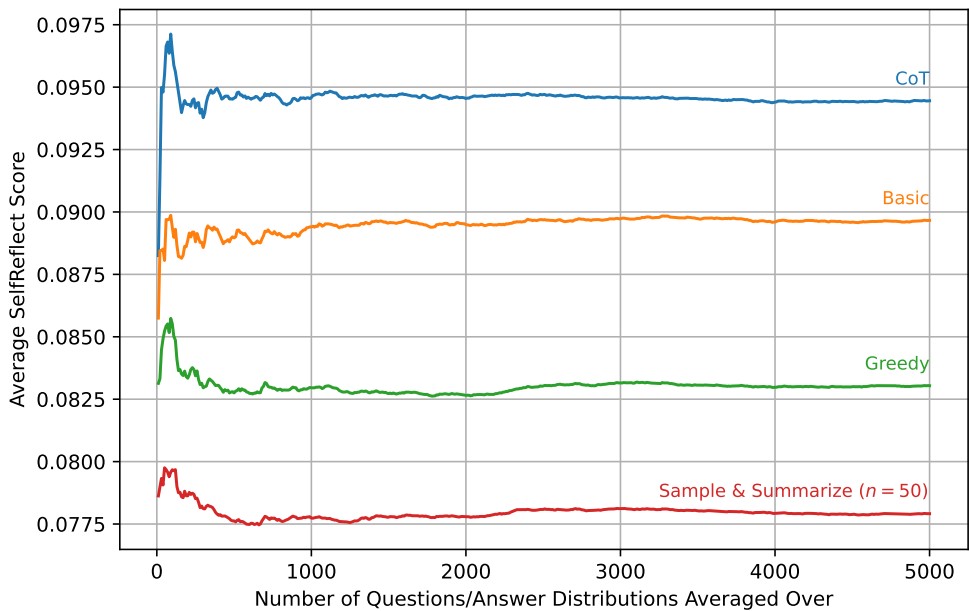

Figure 8: Convergence of the SelfReflect score with $N = M = 10$ and an increasing number of queries we evaluate on. Answer Distributions of Qwen 2.5 72B Instruct on Natural Questions.

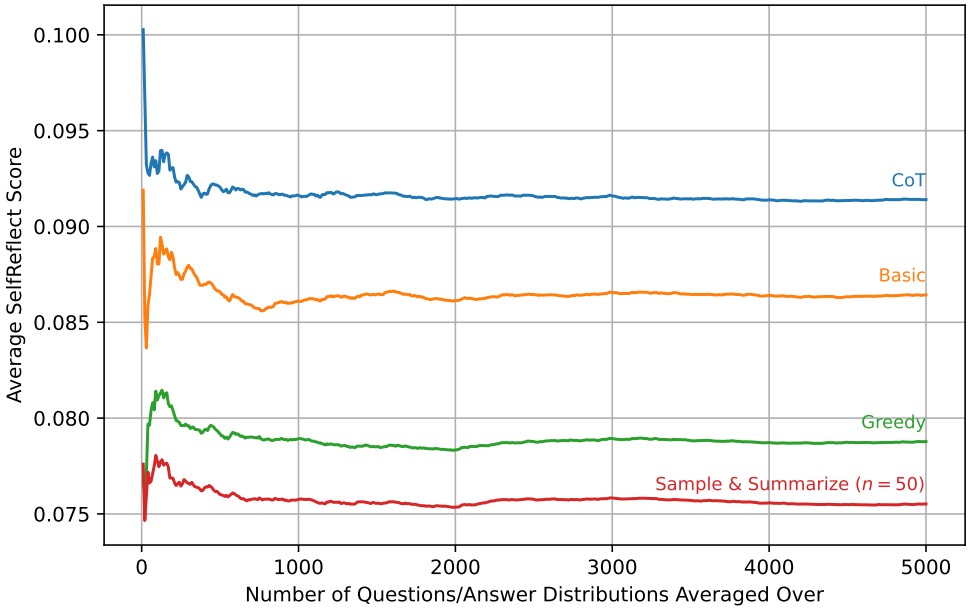

Figure 9: Convergence of the SelfReflect score with $N = M = 5$ and an increasing number of queries we evaluate on. Answer Distributions of Qwen 2.5 72B Instruct on Natural Questions.

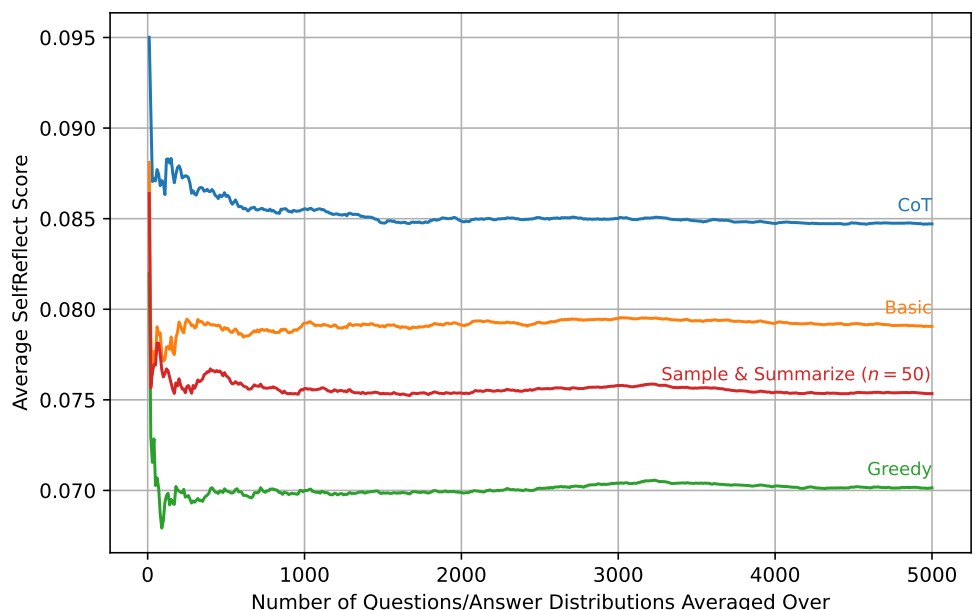

Figure 10: Convergence of the SelfReflect score with $N = M = 1$ and an increasing number of queries we evaluate on. Answer Distributions of Qwen 2.5 72B Instruct on Natural Questions.

## C   WHICH LLM$_J$ JUDGE TO USE TO GENERATE SELFREFLECT LOGITS

Table 7: To find out which LLM judge produces the best logits, we test how often SelfReflect correctly distinguishes a good (top) from a bad (bottom) summarywith different possible judges LLM$_J$ that calculate the SelfReflect metric, across different LLM's LLM$_\theta$ whose answer distributions are being summarized. Automatically generated summaries on Natural Questions, following Table 1. Results for Phi 4 14B as a judge for Llama 3.1 8B Instruct are pending and will be added.

| LLM$_\theta$ | LLM$_J$ | Good summaries vs bad summaries | Good vs almost-good | Detailed vs truncated | Verbalized uncertainty vs only majority answer | Verbalized vs or-concatenated | Percentage vs or-concatenated |
|---|---|---|---|---|---|---|---|
| Llama 3.1 8B Instruct | Llama 3.1 8B Instruct | 99.73%$_{\pm0.37\%}$ | 96.13%$_{\pm1.38\%}$ | 94.92%$_{\pm3.96\%}$ | 97.39%$_{\pm2.91\%}$ | 80.00%$_{\pm7.31\%}$ | 87.83%$_{\pm5.98\%}$ |
| Phi 4 14B | Llama 3.1 8B Instruct | 99.75%$_{\pm0.49\%}$ | 97.50%$_{\pm1.53\%}$ | 100.00%$_{\pm0.00\%}$ | 96.30%$_{\pm5.03\%}$ | 51.85%$_{\pm13.33\%}$ | 66.67%$_{\pm12.57\%}$ |
| Qwen2.5 7B Instruct | Llama 3.1 8B Instruct | 99.70%$_{\pm0.34\%}$ | 94.10%$_{\pm1.46\%}$ | 100.00%$_{\pm0.00\%}$ | 97.52%$_{\pm2.77\%}$ | 47.93%$_{\pm8.90\%}$ | 80.99%$_{\pm6.99\%}$ |
| Llama 3.1 8B Instruct | Phi 4 14B | 99.87%$_{\pm0.26\%}$ | 94.93%$_{\pm1.57\%}$ | 94.92%$_{\pm3.96\%}$ | 99.13%$_{\pm1.70\%}$ | 87.83%$_{\pm5.98\%}$ | 86.09%$_{\pm6.32\%}$ |
| Phi 4 14B | Phi 4 14B | 100.00%$_{\pm0.00\%}$ | 94.25%$_{\pm2.28\%}$ | 94.44%$_{\pm5.33\%}$ | 48.15%$_{\pm13.33\%}$ | 59.26%$_{\pm13.11\%}$ | 59.26%$_{\pm13.11\%}$ |
| Qwen2.5 7B Instruct | Phi 4 14B | 99.70%$_{\pm0.34\%}$ | 93.10%$_{\pm1.57\%}$ | 98.71%$_{\pm1.78\%}$ | 95.04%$_{\pm3.87\%}$ | 59.50%$_{\pm8.75\%}$ | 75.21%$_{\pm7.69\%}$ |
| Llama 3.1 8B Instruct | Qwen2.5 7B Instruct | 100.00%$_{\pm0.00\%}$ | 95.73%$_{\pm1.45\%}$ | 95.76%$_{\pm3.64\%}$ | 95.65%$_{\pm3.73\%}$ | 80.87%$_{\pm7.19\%}$ | 85.22%$_{\pm6.49\%}$ |
| Phi 4 14B | Qwen2.5 7B Instruct | 99.25%$_{\pm0.85\%}$ | 96.75%$_{\pm1.74\%}$ | 98.59%$_{\pm2.74\%}$ | 94.44%$_{\pm6.11\%}$ | 70.37%$_{\pm12.18\%}$ | 77.78%$_{\pm11.09\%}$ |
| Qwen2.5 7B Instruct | Qwen2.5 7B Instruct | 99.80%$_{\pm0.28\%}$ | 94.20%$_{\pm1.45\%}$ | 98.06%$_{\pm2.17\%}$ | 95.04%$_{\pm3.87\%}$ | 74.38%$_{\pm7.78\%}$ | 83.47%$_{\pm6.62\%}$ |
| Llama 3.1 8B Instruct | Qwen2.5 72B Instruct | 99.87%$_{\pm0.26\%}$ | 96.13%$_{\pm1.38\%}$ | 97.46%$_{\pm2.84\%}$ | 99.13%$_{\pm1.70\%}$ | 86.96%$_{\pm6.15\%}$ | 78.26%$_{\pm7.54\%}$ |
| Phi 4 14B | Qwen2.5 72B Instruct | 98.75%$_{\pm1.09\%}$ | 97.50%$_{\pm1.53\%}$ | 98.59%$_{\pm2.74\%}$ | 96.30%$_{\pm5.03\%}$ | 72.22%$_{\pm11.95\%}$ | 55.56%$_{\pm13.25\%}$ |
| Qwen2.5 7B Instruct | Qwen2.5 72B Instruct | 99.80%$_{\pm0.28\%}$ | 94.40%$_{\pm1.43\%}$ | 99.35%$_{\pm1.27\%}$ | 99.17%$_{\pm1.62\%}$ | 75.21%$_{\pm7.69\%}$ | 66.94%$_{\pm8.38\%}$ |

A mandatory component to calculate the SelfReflect metric is a judge LLM$_J$ that predicts which masked-out words are possible, given either a summary or a concatenation of samples. This judge needs to be able to "understand" both the details of the answer and the probabilistic aspect of this task, all the while not overwriting its context information with its own world knowledge when making the prediction. The choice of the judge can thus be seen as a hyperparameter to be optimized to produce SelfReflect scores that are as discriminative as possible between good and bad and almost-good summaries. We test four different judges in this section, Llama 3.1 8B Instruct, Phi 4 14B, Qwen 2.5 7B Instruct (which we ultimately use in the paper), and Qwen 2.5 72B Instruct. We generate answer distributions on Natural Questions for different LLM$_\theta$ (Llama 3.1 8B Instruct, Phi 4 14B, and Qwen 2.5 7B Instruct), then use Gemini 2.0 to generate summaries like in Section 4.1, and calculate how often SelfReflect correctly tells apart good from bad (or almost-good) summaries.

Table 7 shows that SelfReflect is very robust to the choice of the judge LLM: All judges can tell apart good from bad summaries in almost all cases. In particular, there is also no indication of a "home-bias", i.e., that a judge would perform better in judging answer distributions that it sampled itself. This, along with the fact that especially bad summaries, which explicitly introduce statements that are wrong and go against the judge's world knowledge, are almost always judged as worse than good summaries, shows that there is no world-knowledge leakage. We attribute this to LLMs' abilities to predict from their context, and to the fact that SelfReflect runs its prediction both conditional on the summary and conditional on the answer distribution, so that should there be any world knowledge leakage, it would likely be equal and removed.

To make the choice of which LLM judge to use, we pay particular attention to the last three columns of Table 7: Comparing a verbalized or percentage uncertainty answer to an or-concatenated answer is among the most subtle challenges and tests whether the judge correctly infers the relative probabilities in both the answer distributions and the summaries, even when they are not explicit. Here we see that the Qwen family sets itself slightly off Phi 4 and Llama 3.1. Within the Qwen family, the 7B model is within the confidence interval of the 72B model (with a mean result better for percentage vs or-concatenated, and worse for the other two), so we use it in the main paper due to its lower inference cost. We note that we also tried using a Qwen 2.5 0.5B Instruct judge, however, this small model was not able to tell apart good from bad summaries. Finally, we note that there exists a research opportunity in developing an LLM judge specialized to perform the SelfReflect judging, either to compress the 7B model into a smaller and faster one, or to improve the last bits of performance on challenging cases. However, we decide against this in this paper, since a specialized model would increase the complexity of our method and add a dependency on a particular model (-checkpoint), which is likely to be outdated soon in the fast-moving field of LLMs.

## D    EXAMPLE OF SELFREFLECT SCORES PER MASKED-OUT WORD

To deepen the understanding of how the SelfReflect score judges summaries, we provide a worked example. We break down the SelfReflect score to the penalty it gives to each masked-out word. To simplify this educational example, we use only $N = M = 7$ samples and make the answers in the conditioning of the prompt equal to the masked-out test answers.

The question posed to the LLM is *"Who received the first Nobel Prize in physics?"*. As can be seen below, the LLM's answer distribution includes Wilhelm Conrad Röntgen as most likely answer, as well as Hendrik Antoon Lorentz and Pieter Zeeman or Henri Becquerel as additional possibilities, and details on their work. Let us now first look at how SelfReflect judges a relatively bad summary of this distribution which just returns the greedy answer *"Wilhelm Conrad Röntgen received the first Nobel Prize in Physics."*. Overall, SelfReflect assigns this bad summary a distance of 0.102 (or taken ×1000 like in Table 4: 102). This score is due to SelfReflect detecting that Hendrik Antoon Lorentz and Pieter Zeeman or Henri Becquerel are not predictable from the summary, and neither the details of the works, as we can see in the per-word penalties below (darker red = higher penalty).

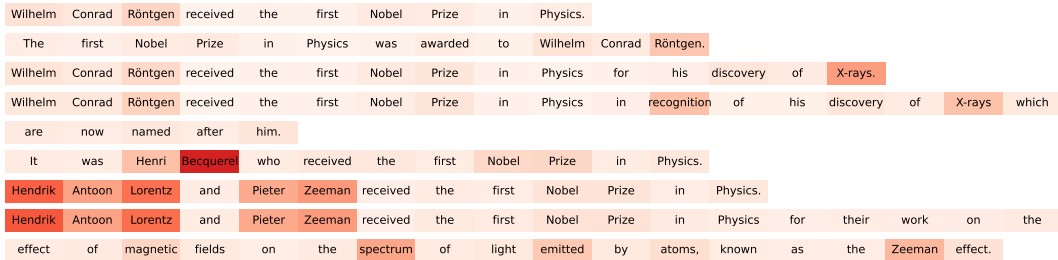

Figure 11: SelfReflect per-word penalties on how far the prediction of each masked-out word based on the summary *"Wilhelm Conrad Röntgen received the first Nobel Prize in Physics."* differs from the prediction based on the samples from the internal distribution. Total penalty: 0.102.

We can now improve this summary by adding the two other possibilities, namely *"It's most likely that Wilhelm Conrad Röntgen received the first Nobel Prize in Physics. But the laureates could also have been Hendrik Antoon Lorentz and Pieter Zeeman or Henri Becquerel."*. With this better summary, SelfReflect correctly removes the penalty on Hendrik Antoon Lorentz, Pieter Zeeman, and Henri Becquerel. But it correctly still penalizes the summary for not mentioning the details of any of the works. This results in an overall score of 0.084 (or 84).

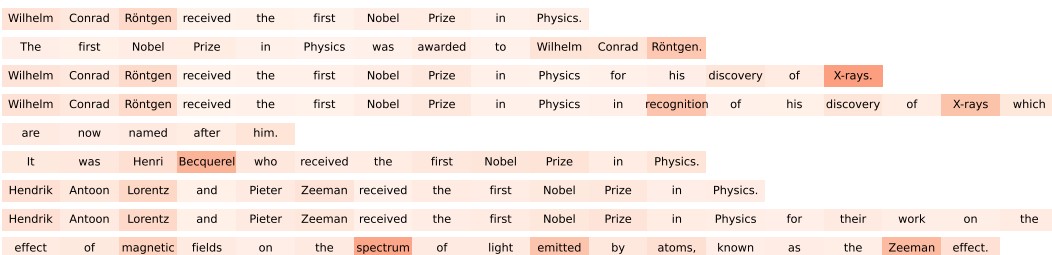

Figure 12: SelfReflect per-word penalties on how far the prediction of each masked-out word based on the summary *"It's most likely that Wilhelm Conrad Röntgen received the first Nobel Prize in Physics. But the laureates could also have been Hendrik Antoon Lorentz and Pieter Zeeman or Henri Becquerel."* differs from the prediction based on the samples from the internal distribution. Total penalty: 0.084.

Having added all answer possibilities, we can now add details mentioned in the individual answers. As a good summary, we give *"It's most likely that Wilhelm Conrad Röntgen received the first Nobel Prize in Physics in recognition of his discovery of X-rays which are now named after him. But the laureates could also have been Hendrik Antoon Lorentz and Pieter Zeeman or Henri Becquerel."*. SelfReflect removes the penalty on X-rays, which the summary mentions. The remaining penalty of

0.078 (or 78) is due to the summary still not mentioning the details on the Zeeman effect, plus some remaining noise mostly on the names.

| Wilhelm | Conrad | Röntgen | received | the | first | Nobel | Prize | in | Physics. | | | | | | |
|---------|--------|---------|----------|-----|-------|-------|-------|----|----------|---|---|---|---|---|---|
| The | first | Nobel | Prize | in | Physics | was | awarded | to | Wilhelm | Conrad | Röntgen. | | | | |
| Wilhelm | Conrad | Röntgen | received | the | first | Nobel | Prize | in | Physics | for | his | discovery | of | X-rays. | |
| Wilhelm | Conrad | Röntgen | received | the | first | Nobel | Prize | in | Physics | in | recognition | of | his | discovery | of | X-rays | which |
| are | now | named | after | him. | | | | | | | | | | | |
| It | was | Henri | Becquerel | who | received | the | first | Nobel | Prize | in | Physics. | | | | |
| Hendrik | Antoon | Lorentz | and | Pieter | Zeeman | received | the | first | Nobel | Prize | in | Physics. | | | |
| Hendrik | Antoon | Lorentz | and | Pieter | Zeeman | received | the | first | Nobel | Prize | in | Physics | for | their | work | on | the |
| effect | of | magnetic | fields | on | the | spectrum | of | light | emitted | by | atoms, | known | as | the | Zeeman | effect. |

Figure 13: SelfReflect per-word penalties on how far the prediction of each masked-out word based on the summary *"It's most likely that Wilhelm Conrad Röntgen received the first Nobel Prize in Physics in recognition of his discovery of X-rays which are now named after him. But the laureates could also have been Hendrik Antoon Lorentz and Pieter Zeeman or Henri Becquerel."* differs from the prediction based on the samples from the internal distribution. Total penalty: 0.078.

These examples demonstrate that SelfReflect punishes summaries for the correct reasons: Either when they don't mention all possibilities or all details of the actual internal answer distribution. We have seen in Sections 4.1 and 4.2 that SelfReflect also correctly punishes deviations from the relative frequencies. To this end, let us modify the second summary which previously had a score of 0.084 (or 84) and state that Henri Becquerel was the most likely first Nobel laureate, which is in conflict with the LLM's internal answer distribution: *"It's most likely that Henri Becquerel received the first Nobel Prize in Physics. But the laureates could also have been Hendrik Antoon Lorentz and Pieter Zeeman or Wilhelm Conrad Röntgen."*. This correctly leads to higher penalties on Wilhelm Conrad Röntgen and Henri Becquerel because both of their implied probabilities are off (while keeping the same penalties on Hendrik Antoon Lorentz and Pieter Zeeman, as well as the in both cases unmentioned details on their works) and worsens the SelfReflect score to 0.092 (or 92).

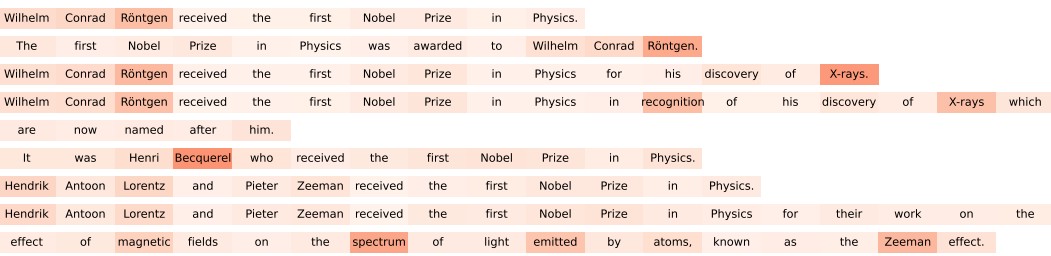

Figure 14: SelfReflect per-word penalties on how far the prediction of each masked-out word based on the summary *"It's most likely that Henri Becquerel received the first Nobel Prize in Physics. But the laureates could also have been Hendrik Antoon Lorentz and Pieter Zeeman or Wilhelm Conrad Röntgen."* (note that Henri Becquerel is in fact not the most likely; it is Wilhelm Conrad Röntgen) differs from the prediction based on the samples from the internal distribution. Total penalty: 0.092.

This demonstrates that SelfReflect's score works as intended, not only on the dataset or question level as studied in the main paper, but also on a word-level granularity. This example is a regular case, one of the 95%+ (see Table 1) where SelfReflect correctly scores the summaries. We note, however, that there are around 5% of questions where it does not score correctly. In most of these cases, the scores of a good and a slightly worse summary are very close to one another and the mis-decision is mostly due to noise. We thus recommend to run SelfReflect over 1000 questions per dataset, as noted in Appendix B and the main paper, in order to smoothen out some of the remaining noise.

# E    IMPLEMENTATION DETAILS

## E.1    SELFREFLECT SCORE

To calculate the SelfReflect score, in every masked-out task we run the two prompts in Fig. 3 through a judge LLM, which is by default Qwen 2.5 7B-Instruct. This makes the judge predict the logits over the vocabulary size for the current token of the fill-in word. If a fill-in word consists of multiple tokens, where we add the tokens of the true fill-in word one after another into the autoregressive context of the assistant answer. Given the two fill-in token vectors conditioned either on the summary or on the concatenated answers, we apply a temperature of 5 to flatten it. This is in order to give some weight to synonyms, since instruct-tuned LM judges otherwise would give nearly probability 1 to only one possible token (in which case SelfReflect would still be valid, but simplify into comparing whether the two contexts lead to predicting the exactly same word). We found that a temperature of $\tau = 5$ improves the SelfReflect score, making it able to discern good from almost-good summaries more often on a validation dataset. We then softmax the flattened logit vectors and calculate the 1-Wasserstein distance between the log probability vectors. Since these are categorical vectors, the 1-Wasserstein distance simplifies into the $L_1$ distance, times $0.5$. We repeat this over all tokens of a masked-out word, then over all masked-out words of each of the $M = 50$ answers (that are not stopwords), then across all 1000 questions of a dataset. The global average gives the SelfReflect score.

## E.2    SR SAMPLING-FREE SCORE

The sampling-free ablation of the SelfReflect metric also gives two prompts to a judge LLM to calculate the masked-in task. The difference is that the prompt which in SelfReflect contains the sampled answers does not contain sampled answers. Instead it just gives the question and then the masked-out task.

## E.3    SR-PMI SCORE

The PMI ablation of the SelfReflect metric uses no masked out task. Instead it poses the question, gives either the summary or the sampled answers as background information in context, and then measures the logit vectors assigned to each token of each of the $M = 50$ answers. In other words, the answers are not given word-by-word with masked-out tasks, but measured as one full answer. As for SelfReflect, we then calculate the 1-Wasserstein distance between the flattened logit vectors and average.

## E.4    SR-P(TRUE) SCORE

In the P(True) ablation of SelfReflect, we turn the generative masked-out task into a discriminative one. We first generate three candidate words to fill in the masked word: One is the true masked word, one is a word sampled from the masked-out task prompt given the summary and the last is a word sampled from the masked-out task prompt given the distribution samples. With these candidate fill-in words, we then run two prompts, one conditional on the summary and one on the answers, to let the judge LLM predict how likely they fit in, see Fig. 15. As in the normal SelfReflect, this gives a distribution over the vocabulary size, concentrated on "True" and "False" tokens. We then compare the two flattened logit vectors via the 1-Wasserstein distance and average as in the original SelfReflect.

## E.5    EMBEDDING SCORE

We compare the gte-Qwen2-7B-instruct (Li et al., 2023) embedding of the summary to the embedding of the samples of the distribution, normalize them and take the inner product to form cosine distances. We average over all samples. The reason why we select this particular embedding model is that at the time of submission it was the best-performing open-source model on the MTEB benchmark (Enevoldsen et al., 2025).

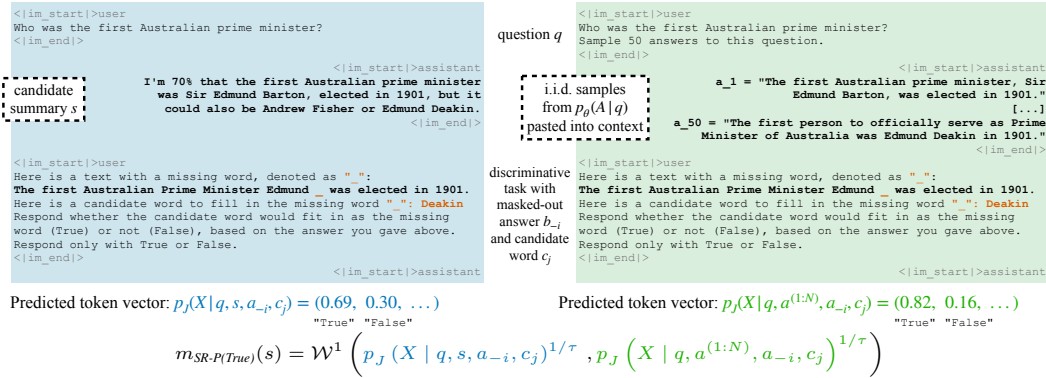

Predicted token vector: $p_J(X|q, s, a_{-i}, c_j) = (0.69, \ 0.30, \ \dots)$

Predicted token vector: $p_J(X|q, a^{(1:N)}, a_{-i}, c_j) = (0.82, \ 0.16, \ \dots)$

$$m_{SR\text{-}P(True)}(s) = \mathcal{W}^1 \left( p_J \left( X \mid q, s, a_{-i}, c_j \right)^{1/\tau}, p_J \left( X \mid q, a^{(1:N)}, a_{-i}, c_j \right)^{1/\tau} \right)$$

Figure 15: The P(True) ablation of SelfReflect adds a candidate word $c_j$ into the context and asks the judge LLM to classify whether this word fits as masked-out word or not. It compares the probability vectors predicted given either the summary or the concatenated samples.

## E.6 SUMMARIZATION SCORE

We follow the prompts of Jain et al. (2023) that prompt an LLM to judge a summary in terms of consistency, fluency, relevance, and coherence with a few-shot example. We then normalize all scores to [0, 1] and average them to get the summarization score.

---

**Prompt for the 'Summarization' metric in Table 1 to judge fluency.**

```
Fluency measures the quality of individual sentences, and whether
they are well-written and grammatically correct.  Rate the summary
of a given text on a scale of 0 to 1 on fluency.

Here are some examples:  <4 few-shot examples>

Now here is the summary whose fluency you are supposed to rate:
Summary:  {summary}

Fluency:
```

---

**Prompt for the 'Summarization' metric in Table 1 to judge coherence.**

```
Rate the following summaries on a scale from 0 to 1 on coherence,
with a higher value corresponding to higher coherence.  Coherence
is a collective quality of all sentences.  To score highly on
it, the summary should be well-structured and well-organized.  It
should not just be a heap of related information, but should build
from sentence to sentence to form a coherent body of information
about the topic.

Here are some examples:  <4 few-shot examples>

Now here is the summary whose coherence you are supposed to rate:
Summary:  {summary}

Coherence:
```

---

**Prompt for the 'Summarization' metric in Table 1 to judge consistency.**

```
Consistency measures whether the details in the summary reproduce
the facts present in the text accurately.  Rate the summary of
```

```
given text on a scale from 0 to 1 on consistency.

Here are some examples:  <4 few-shot examples>

Now here is the text and summary whose consistency you are supposed
to rate:
Text:  We received many answers to our question '{question}'.  Here
they are:
x_1 = '{answer}'
...
x_{n_answers} = '{answer}'

Summary:  {summary}

Consistency:
```

**Prompt for the 'Summarization' metric in Table 1 to judge relevance.**

```
Relevance is the quality of a summary to capture important
information from a reference text.  Rate the summary on a scale
from 0 to 1 on relevance.

Here are some examples:  <4 few-shot examples>

Now here is the text and summary whose relevance you are supposed
to rate:
Text:  We received many answers to our question '{question}'.  Here
they are:
x_1 = '{answer}'
...
x_{n_answers} = '{answer}'

Summary:  {summary}

Relevance:
```

## E.7   LM JUDGE SCORE

We follow Xu et al. (2024) to build a metric that asks an LM judge to chain-of-thoughts think and rate how well a summary matches a distribution of answers, including a few-shot example. The prompt is shown below.

**Prompt for the 'LM Judge' metric in Table 1.**

```
Your task is to analyze whether a summarized answer correctly
contains all the possibilities that len(answers) individual answers
to a question mention.
Note that some individual answers occur more often than other
individual answers.  You should output a score from 0 to 10,
indicating whether the summarized answer mentions all possibilities
and whether it correctly outlines which are the most often
appearing individual answers and which appear less often.  A higher
score is means the summarized answer matches the distribution of
individual answers better.
Also note that some individual answers may be factually wrong.
Do not correct those, just report how good the summarized answer
matches the individual answers.
You should first provide your reasoning for how well the summarized
answer matches the distribution over individual answers, and then
assign a score based on this reasoning.  The output should be in
```

```
the following format:

Reason:  [REASON]
Score:   [SCORE]

Here is an example:
Question:  <Example question>
Individual answers:
<Example answer samples>
Summarized answer:  <Example summary>

Then your output can be:
Reason:  The summarized answer mentions the most likely possibility,
and it also correctly mentions that this is the most likely one.
For other possibilities, it mentions Wilhelm Conrad Röntgen, but
does not mention that he got the award for his discovery of x-rays,
which the individual answers do mention.  It also does not mention
the possibility of Hendrik Antoon Lorentz and Pieter Zeeman, which
the individual answers mention.
Score:  8

Now consider the following case:
{question}
Individual answers:
x_1 = '{answer}'
...
x_{n_answers} = '{answer}'

Summarized answer:  '{summary}'

Please provide the reason and the score of how good the summarized
answer matches the distribution of individual answers.
```

## E.8  OPTIMAL TRANSPORT SCORE

The optimal transport metric consists of two steps. First, we break down a summary into a distribution of statements and their probabilities. This is done with the following prompt.

---

**Prompt for the 'Optimal transport' metric in Table 1 to split a summary into core statements and their probabilities.**

```
Question:  {question}

Here is some background information.  This background information
defines a distribution of possible answers you can later sample
from:
{summary}

Now, split this distribution up into its mutually fundamental
statements and the explicitly or implicitly connected
probabilities.
Split it up such that each statement is mutually exclusive and the
probabilities sum to 1.
Include an 'I don't know' statement with the remaining percentage
if the background information explicitly mentions not being
certain.
Return a json file with a list of dictionaries, where in every
dictionary the first key is called 'prob' and includes the
numerical probability and the second key is 'statement' and
includes a string of the fundamental statement.
```

---

In the second step, we use an NLI model to calculate an entailment probability in [0, 1] between how much each sample answer entails each statement and vice versa. We multiply $(1 - \text{entailment probability})$ of both directions to get an distance score for each sample answer and statement. This defines a distance matrix with the statements as rows and the sample answers as columns. Besides a distance matrix, optimal transport also requires marginals for both rows and columns. For the rows, we use the probabilities assigned to the statements in the above prompt. For the columns, we assign each individual sample answer a uniform probability. We then compute the earth movers distance using Flamary et al. (2021). This matches sample answers to summary statements in such a manner that the marginals are preserved and that overall all pairs in sum have the smallest possible distance. The resulting overall distance then tells how far the answer samples are from the summary.

## E.9 INFOSUMM SCORE

InfoSumm (Jung et al., 2024) is a metric from summarization literature to compare the pointwise mutual information between a summary and a document string based on masked-out tasks, similar to SelfReflect in spirit but with some notable differences in the estimator. InfoSumm expects to have a summary string $s$ and a long document, which corresponds to concatenated answers from the sampled distribution $a^{(1:N)}$ in our case. They then estimate PMI once via saliency and once via faithfulness. Saliency predicts masked words of the long document with an LLM judge that is given the summary, and faithfulness predicts masked words of the summary with an LLM judge given the long document.[6] After removing terms in the denominator that are independent of the summary in question, this gives two log likelihood terms:

$$\texttt{saliency\_score} := \log\left(p_{\text{LLM}_J}\left(A_m^{(1:N)} = a_m^{(1:N)} \mid Q = q, A_{-m}^{(1:N)} = a_{-m}^{(1:N)}, s\right)\right), \quad (22)$$

$$\texttt{faithfulness\_score} := \log\left(p_{\text{LLM}_J}\left(S_m = s_m \mid Q = q, S_{-m} = s_{-m}, A^{(1:N)} = a^{(1:N)}\right)\right), \quad (23)$$

where $x_m$ denotes masked-out words and $x_{-m}$ the remaining non-masked text. To make the masked-out task for saliency less ambiguous to the LLM judge (since there may be duplicated answers in $a_m^{(1:N)}$), we compute the saliency score per answer:

$$\texttt{saliency\_score} := \frac{1}{N}\sum_{n=1}^{N}\log\left(p_{\text{LLM}_J}\left(A_m^n = a_m^n \mid Q = q, A_{-m}^n = a_{-m}^n, s\right)\right). \quad (24)$$

In the original paper, these terms are thresholded and used as filters to create train datasets. In our application as a score, we add them up and multiply by $(-1)$ to ensure that lower is better. This gives the InfoSumm baseline:

$$\texttt{InfoSumm\_score} := -(\texttt{saliency\_score} + \texttt{faithfulness\_score}). \quad (25)$$

We integrate this over the masked-out tasks of all $M$ answers as we do for SelfReflect.

## E.10 LICENSING INFORMATION

Table 8 contains licensing information for models used in this paper.

---

[6]There is no differentiation between answers for the conditioning $a$ and for masked-out tasks $b$ in their setup, because they never condition one on the other.

Table 8: Licencing information for models used in this work

| LLM | License | Reference |
|---|---|---|
| DeepSeek R1 Distill Qwen 2.5 32B | MIT | DeepSeek-AI et al. (2025) |
| Gemma 3 family | https://ai.google.dev/gemma/terms | Gemma Team et al. (2025) |
| Gemini 2.0 Flash | Apache 2.0 | |
| Ministral 8B Instruct 2410 | https://mistral.ai/static/licenses/MRL-0.1.md | Jiang et al. (2024) |
| Llama 3.1 70B Instruct | https://www.llama.com/llama3_1/license/ | Meta AI (2024a) |
| Llama 3.3 70B Instruct | https://www.llama.com/llama3_3/license/ | Meta AI (2024b) |
| Llama 4 Scout 17b 16e Instruct | https://www.llama.com/llama4/license/ | Meta AI (2025) |
| Phi-4 | MIT | Abdin et al. (2024) |
| gte Qwen 2 7B Instruct | Apache-2.0 | Li et al. (2023) |
| Qwen 2.5 family | Apache-2.0 | Yang et al. (2024a) |
| Qwen 3 family | Apache-2.0 | Qwen Team (2025a) |
| QwQ 32B | Apache-2.0 | Qwen Team (2025b) |

# F  RATING GOOD AND BAD SUMMARIES WRITTEN BY HUMANS

Table 9: Mirroring Table 1, we compare how often our SelfReflect metric, and other possible metrics, discriminates good from bad summaries of answer distributions. In this version, the summaries are written by humans rather than by Gemini, on a disjoint set of questions. Mean $\pm$ 95% interval. Confidence intervals are larger than in Table 1 because we have less manually written summaries of answer distributions than the automated ones in Table 1.

| Metric | Good summaries vs bad summaries | Good vs almost-good | Detailed vs truncated | Verbalized uncertainty vs only majority answer | Verbalized vs or-concatenated | Percentage vs or-concatenated |
|---|---|---|---|---|---|---|
| Summarization | $98.20\%_{\pm1.43\%}$ | $60.18\%_{\pm5.25\%}$ | $70.00\%_{\pm20.08\%}$ | $24.14\%_{\pm7.93\%}$ | $55.17\%_{\pm18.10\%}$ | $51.72\%_{\pm18.19\%}$ |
| LM Judge | $98.50\%_{\pm1.30\%}$ | $54.19\%_{\pm5.34\%}$ | $55.00\%_{\pm21.80\%}$ | $34.48\%_{\pm17.30\%}$ | $37.93\%_{\pm17.66\%}$ | $24.14\%_{\pm15.58\%}$ |
| Opt. Transport | $78.14\%_{\pm4.43\%}$ | $57.19\%_{\pm5.31\%}$ | $10.00\%_{\pm13.15\%}$ | $58.62\%_{\pm17.93\%}$ | $79.31\%_{\pm14.74\%}$ | $72.41\%_{\pm16.27\%}$ |
| Embedding | $74.85\%_{\pm4.65\%}$ | $44.01\%_{\pm5.32\%}$ | $60.00\%_{\pm21.47\%}$ | $20.69\%_{\pm14.74\%}$ | $41.38\%_{\pm17.93\%}$ | $13.79\%_{\pm12.55\%}$ |
| SR-PMI | $85.33\%_{\pm3.79\%}$ | $60.18\%_{\pm5.25\%}$ | $75.00\%_{\pm18.98\%}$ | $3.45\%_{\pm6.64\%}$ | $24.14\%_{\pm15.58\%}$ | $31.03\%_{\pm16.84\%}$ |
| SR-sampling-free | $92.22\%_{\pm2.87\%}$ | $80.24\%_{\pm4.27\%}$ | $75.00\%_{\pm18.98\%}$ | $62.07\%_{\pm17.66\%}$ | $62.07\%_{\pm17.66\%}$ | $58.62\%_{\pm17.93\%}$ |
| SR-P(True) | $47.90\%_{\pm5.36\%}$ | $58.38\%_{\pm5.29\%}$ | $35.00\%_{\pm20.90\%}$ | $\mathbf{96.55\%}_{\pm6.64\%}$ | $82.76\%_{\pm13.75\%}$ | $79.31\%_{\pm14.74\%}$ |
| SelfReflect | $\mathbf{99.70\%}_{\pm0.59\%}$ | $\mathbf{94.61\%}_{\pm2.42\%}$ | $\mathbf{95.00\%}_{\pm9.55\%}$ | $86.21\%_{\pm12.55\%}$ | $\mathbf{93.10\%}_{\pm9.22\%}$ | $\mathbf{82.76\%}_{\pm13.75\%}$ |

In Table 1 in the main paper, we use Gemini 2.0 Flash to generate various types of good and bad summaries from sampled answers. We choose an automated LLM approach because it is more scalable (with accordingly preciser 95% intervals) and reproducible than manual annotation. For reproducibility, we also report the prompts below. To ensure the quality of the results, we have, however, also replicated the experiments where we wrote the summaries manually for 334 questions of the Natural Questions dataset, on a disjoint split from those in Table 1. Table 9 reports the results on how often SelfReflect, and other metrics, rated good summaries as better than their worse counterparts. The results are analogous to those in the main paper in that SelfReflect scores the highest on all metrics except on one where the P(True) ablation achieves a slightly better result.

```
Gemini 2.0 Flash prompt to generate 'good' summaries in Table 1.

Below, you are given {n_answers} individual answers to the question
'{question}'.

Your goal is to summarize the {n_answers} answers into one answer.
    • The summarized answer should mention the main possibilities
      mentioned by the {n_answers} answers.  If a possibility is
      mentioned only once, it can be skipped so that the summary
      remains concise.
    • If some possibilities are mentioned much more often than
      others, delineate which possibilities are more often found in
      the others by using words like "most likely" and "could also
      be".
```

```
          • The format of the summarized answer should be the same as
            each individual answer.  Provide only the answer, as if it
            were part of the {n_answers} answers, without statements like
            "The answers include...".
          • Similarly, the summarized answer should use the same wording
            as the original answers.  If the original answer always uses
            "is situated", then use "is situated" and not "is located".
          • The summarized answer should reflect what the {n_answers}
            answers deem possible.  They can contain factually wrong
            options.  Do not correct those, just report the possibilities
            as they are given in the answers.
Here are the {n_answers} answers:
x_1 = '{answer}'
...
x_{n_answers} = '{answer}'

Please provide the summarized answer.
```

**Gemini 2.0 Flash prompt to generate an 'almost-good' summary from a 'good' summary in Table 1. Also used to generate 'truncated' from 'detailed' summaries.**

```
Below, you are given an answer to the question '{question}'.

Your goal is to shorten the answer.
      • If the answer mentions multiple possibilities, only return
        the main possibility.
      • If the answer includes a main answer and details, remove the
        details.
      • The shortened answer should have the same format as the
        original answer.  If the original answer uses full sentences,
        the shortened answer should also use a full sentence.
      • The shortened answer should use the same wording as the
        original answers.  If the original answer always uses "is
        situated", then use "is situated" and not "is located".
      • The answer can contain factually wrong options.  Do not
        correct those, just shorten what the answer says, even if it
        is factually wrong.
Original answer:  {good_summary}

Please provide the shortened answer.
```

**Gemini 2.0 Flash prompt to generate a 'bad' summary from a 'good' summary in Table 1.**

```
Below, you are given a response to the question '{question}'.

Your goal is to change the answer.
      • The answer should generally stay close to the original
        answer, with only some key factual terms changed.
      • The answer might already be factually wrong.  But the goal is
        still to change the key facts, so that the changed answer is
        different from the original one.
      • The changed answer should have the same format as the
        original answer.  The structure should remain the same, only
        keywords should be exchanged.
      • The changed answer should also use the same wording as the
        original answers for any non-factual words.  If the original
```

```
        answer always uses "is situated", then use "is situated" and
        not "is located".
Original answer:  {good_summary}

Please provide the changed answer.
```

For verbalized, percentage, or-concatenated and majority answers, we first use a prompt to cluster the answer distribution into clusters of statements and which answers belong to which cluster statement:

**Gemini 2.0 Flash prompt to cluster samples from an answer distribution into a list of cluster representatives and cluster memberships.**

```
Below, you are given {n_answers} individual answers to the question
'{question}'.  These {n_answers} answers can be seen as samples from
an answer distribution.  Your goal is to cluster the distribution
in two steps:

First step:  Find the clusters and their representatives.
    • Each cluster contains a set of answers that are essentially
      the same.  This means they may vary in the level of detail,
      but their primary answer should be the same.
    • Different clusters should be mutually exclusive answers.
    • There are at least two clusters.
    • The answer can contain factually wrong options.  Do not
      correct them, just cluster the answers as they are.
    • Output a json file with each entry giving the "cluster_id"
      (cluster_1, cluster_2, ...), and a "representative_answer",
      copy-pasted from the answers below.
Second step:  Match the answers to their clusters.
    • Match each of the {n_answers} individual answers to one
      cluster representative.
    • Output a json file with each entry giving the "cluster_id"
      (cluster_1, cluster_2, ...)  and the "cluster_members", a
      list of [x_1, x_26, ...].
Here are the {n_answers} answers:
x_1 = '{answer}'
...
x_{n_answers} = '{answer}'

Please output the two json files, one after another.  Each json
file should start with ``json
```

We then count how many member each cluster has (manually in code as opposed to asking the LLM since this increases accuracy), and provide lists of the representative answers of each cluster and their relative frequencies to build the percentage and or-concatenated summaries. We sort the resulting list frequency. For the majority answer, we directly return the representative answer of the highest-likely cluster. The verbalized uncertainty summary is built by removing the percentages in their brackets from the percentage summary.

**Gemini 2.0 Flash prompt to generate 'percentage-uncertainty' summaries in Table 1.**

```
Below, you are given list of answers with their probabilities to
the question '{question}'.

Your goal is to stitch these answers together into one sentence.
    • The sentence should have the structure 'It is most likely
      that <Answer A> (<probability of Answer A>% sure), but it
```

```
          could also be <Answer B> (<probability of Answer B>% sure) or
          <Answer C> (<probability of Answer C>% sure) or ...'

      • Stick to the original wording of the answers as much as
        possible, but you can add small words so that the sentence
        becomes a grammatically coherent sentence.

      • The answer can contain factually wrong options.  Do not
        correct those, just stitch together the answer options, even
        if it is factually wrong.
List of answers:
[
  {
  'prob':  0.72,
  'statement':  ...
  },
  {
  'prob':  0.22,
  'statement':  ...
  }
]

Please provide the coherent sentence.
```

**Gemini 2.0 Flash prompt to generate 'or-concatenated' summaries in Table 1.**

```
Below, you are given list of answers with their probabilities to
the question '{question}'.

Your goal is to stitch these answers together into one sentence.

      • The sentence should have the structure 'Either <Answer A> or
        <Answer B> or <Answer C> or ...'

      • The sentence should be grammatically coherent.

      • Stick to the original wording of the answers as much as
        possible.

      • The answer can contain factually wrong options.  Do not
        correct those, just stitch together the answer options, even
        if it is factually wrong.
List of answers:
[
  {
  'prob':  0.72,
  'statement':  ...
  },
  {
  'prob':  0.22,
  'statement':  ...
  }
]

Please provide the coherent sentence.
```

## G How well does SelfReflect distinguish good from bad summaries

Table 10: How well does SelfReflect, and other metrics, discriminate between good and bad summaries of answer distributions. For each column, the second summary lacks textual details or misrepresents relative probabilities. Mean $\pm$ 95% confidence interval. Results per dataset, where Table 1 shows results averaged across datasets.

| Metric | Good summaries vs bad summaries | Good vs almost-good | Detailed vs truncated | Verbalized uncertainty vs only majority answer | Verbalized vs or-concatenated | Percentage vs or-concatenated |
|---|---|---|---|---|---|---|
| **Natural Questions** | | | | | | |
| Summarization | $98.70\%_{\pm 0.70\%}$ | $40.67\%_{\pm 3.12\%}$ | $53.55\%_{\pm 7.85\%}$ | $11.57\%_{\pm 5.70\%}$ | $57.02\%_{\pm 8.82\%}$ | $65.29\%_{\pm 8.48\%}$ |
| LM Judge | $99.10\%_{\pm 0.59\%}$ | $50.21\%_{\pm 3.17\%}$ | $65.16\%_{\pm 7.50\%}$ | $24.79\%_{\pm 7.69\%}$ | $33.88\%_{\pm 8.43\%}$ | $38.02\%_{\pm 8.65\%}$ |
| Opt. Transport | $91.89\%_{\pm 1.69\%}$ | $56.50\%_{\pm 3.15\%}$ | $45.16\%_{\pm 7.83\%}$ | $48.76\%_{\pm 8.91\%}$ | $47.11\%_{\pm 8.89\%}$ | $70.25\%_{\pm 8.15\%}$ |
| Embedding | $94.49\%_{\pm 1.41\%}$ | $31.34\%_{\pm 2.94\%}$ | $47.10\%_{\pm 7.86\%}$ | $5.79\%_{\pm 4.61\%}$ | $46.28\%_{\pm 8.88\%}$ | $38.02\%_{\pm 8.65\%}$ |
| SR-logl | $99.10\%_{\pm 0.59\%}$ | $90.04\%_{\pm 1.90\%}$ | $90.97\%_{\pm 4.51\%}$ | $66.94\%_{\pm 8.38\%}$ | $39.67\%_{\pm 8.72\%}$ | $49.59\%_{\pm 8.91\%}$ |
| SR-PMI | $90.89\%_{\pm 1.78\%}$ | $47.90\%_{\pm 3.17\%}$ | $69.68\%_{\pm 7.24\%}$ | $23.97\%_{\pm 7.61\%}$ | $9.09\%_{\pm 5.12\%}$ | $16.53\%_{\pm 6.62\%}$ |
| SR-sampling-free | $96.30\%_{\pm 1.17\%}$ | $74.42\%_{\pm 2.77\%}$ | $82.58\%_{\pm 5.97\%}$ | $39.67\%_{\pm 8.72\%}$ | $29.75\%_{\pm 8.15\%}$ | $30.58\%_{\pm 8.21\%}$ |
| SR-P(True) | $55.86\%_{\pm 3.08\%}$ | $74.95\%_{\pm 2.75\%}$ | $62.58\%_{\pm 7.62\%}$ | $92.56\%_{\pm 4.68\%}$ | $69.42\%_{\pm 8.21\%}$ | $85.95\%_{\pm 6.19\%}$ |
| SelfReflect | $99.90\%_{\pm 0.20\%}$ | $98.74\%_{\pm 0.71\%}$ | $98.06\%_{\pm 2.17\%}$ | $95.04\%_{\pm 3.87\%}$ | $74.38\%_{\pm 7.78\%}$ | $83.47\%_{\pm 6.62\%}$ |
| **SimpleQA** | | | | | | |
| Summarization | $85.40\%_{\pm 2.19\%}$ | $36.36\%_{\pm 3.03\%}$ | $50.00\%_{\pm 17.89\%}$ | $26.32\%_{\pm 19.80\%}$ | $63.16\%_{\pm 21.69\%}$ | $68.42\%_{\pm 20.90\%}$ |
| LM Judge | $96.30\%_{\pm 1.17\%}$ | $50.31\%_{\pm 3.15\%}$ | $46.67\%_{\pm 17.85\%}$ | $10.53\%_{\pm 13.80\%}$ | $31.58\%_{\pm 20.90\%}$ | $31.58\%_{\pm 20.90\%}$ |
| Opt. Transport | $73.50\%_{\pm 2.74\%}$ | $68.70\%_{\pm 2.92\%}$ | $16.67\%_{\pm 13.34\%}$ | $42.11\%_{\pm 22.20\%}$ | $63.16\%_{\pm 21.69\%}$ | $68.42\%_{\pm 20.90\%}$ |
| Embedding | $97.80\%_{\pm 0.91\%}$ | $81.61\%_{\pm 2.44\%}$ | $83.33\%_{\pm 13.34\%}$ | $10.53\%_{\pm 13.80\%}$ | $42.11\%_{\pm 22.20\%}$ | $36.84\%_{\pm 21.69\%}$ |
| SR-logl | $91.50\%_{\pm 1.73\%}$ | $84.09\%_{\pm 2.30\%}$ | $100.00\%_{\pm 0.00\%}$ | $36.84\%_{\pm 21.69\%}$ | $31.58\%_{\pm 20.90\%}$ | $31.58\%_{\pm 20.90\%}$ |
| SR-PMI | $84.00\%_{\pm 2.27\%}$ | $25.21\%_{\pm 2.74\%}$ | $33.33\%_{\pm 16.87\%}$ | $26.32\%_{\pm 19.80\%}$ | $21.05\%_{\pm 18.33\%}$ | $26.32\%_{\pm 19.80\%}$ |
| SR-sampling-free | $79.20\%_{\pm 2.52\%}$ | $36.47\%_{\pm 3.03\%}$ | $60.00\%_{\pm 17.53\%}$ | $15.79\%_{\pm 16.40\%}$ | $31.58\%_{\pm 20.90\%}$ | $52.63\%_{\pm 22.45\%}$ |
| SR-P(True) | $72.10\%_{\pm 2.78\%}$ | $87.50\%_{\pm 2.08\%}$ | $83.33\%_{\pm 13.34\%}$ | $84.21\%_{\pm 16.40\%}$ | $63.16\%_{\pm 21.69\%}$ | $78.95\%_{\pm 18.33\%}$ |
| SelfReflect | $96.60\%_{\pm 1.12\%}$ | $91.63\%_{\pm 1.74\%}$ | $100.00\%_{\pm 0.00\%}$ | $68.42\%_{\pm 20.90\%}$ | $68.42\%_{\pm 20.90\%}$ | $73.68\%_{\pm 19.80\%}$ |
| **TriviaQA** | | | | | | |
| Summarization | $95.90\%_{\pm 1.23\%}$ | $43.02\%_{\pm 3.64\%}$ | $53.25\%_{\pm 11.14\%}$ | $37.25\%_{\pm 13.27\%}$ | $58.82\%_{\pm 13.51\%}$ | $62.75\%_{\pm 13.27\%}$ |
| LM Judge | $99.60\%_{\pm 0.39\%}$ | $39.35\%_{\pm 3.60\%}$ | $54.55\%_{\pm 11.12\%}$ | $9.80\%_{\pm 8.16\%}$ | $37.25\%_{\pm 13.27\%}$ | $29.41\%_{\pm 12.51\%}$ |
| Opt. Transport | $75.10\%_{\pm 2.68\%}$ | $55.71\%_{\pm 3.66\%}$ | $37.66\%_{\pm 10.82\%}$ | $50.98\%_{\pm 13.72\%}$ | $62.75\%_{\pm 13.27\%}$ | $66.67\%_{\pm 12.94\%}$ |
| Embedding | $97.20\%_{\pm 1.02\%}$ | $89.42\%_{\pm 2.26\%}$ | $96.10\%_{\pm 4.32\%}$ | $23.53\%_{\pm 11.64\%}$ | $39.22\%_{\pm 13.40\%}$ | $33.33\%_{\pm 12.94\%}$ |
| SR-logl | $98.50\%_{\pm 0.75\%}$ | $82.79\%_{\pm 2.78\%}$ | $72.73\%_{\pm 9.95\%}$ | $45.10\%_{\pm 13.66\%}$ | $47.06\%_{\pm 13.70\%}$ | $54.90\%_{\pm 13.66\%}$ |
| SR-PMI | $90.30\%_{\pm 1.83\%}$ | $25.95\%_{\pm 3.23\%}$ | $28.57\%_{\pm 10.09\%}$ | $29.41\%_{\pm 12.51\%}$ | $23.53\%_{\pm 11.64\%}$ | $27.45\%_{\pm 12.25\%}$ |
| SR-sampling-free | $89.30\%_{\pm 1.92\%}$ | $53.60\%_{\pm 3.67\%}$ | $59.74\%_{\pm 10.95\%}$ | $45.10\%_{\pm 13.66\%}$ | $49.02\%_{\pm 13.72\%}$ | $50.98\%_{\pm 13.72\%}$ |
| SR-P(True) | $67.90\%_{\pm 2.89\%}$ | $83.64\%_{\pm 2.72\%}$ | $77.92\%_{\pm 9.26\%}$ | $78.43\%_{\pm 11.29\%}$ | $80.39\%_{\pm 10.90\%}$ | $90.28\%_{\pm 8.16\%}$ |
| SelfReflect | $99.80\%_{\pm 0.28\%}$ | $87.87\%_{\pm 2.40\%}$ | $80.52\%_{\pm 8.85\%}$ | $68.63\%_{\pm 12.73\%}$ | $70.59\%_{\pm 12.51\%}$ | $74.51\%_{\pm 11.96\%}$ |

## H MMLU tests of the SelfReflect metric per dataset

Table 11: Rank Correlations between how SelfReflect, and others, rank good/overconfident/random summaries of MMLU multiple-choice summaries versus how a metric specialized for this task does. Mean $\pm$ 95% CI. Per Q means we check the rank correlation on each individual question and then average the rank correlations globally, Avg means we let the approaches calculate the average score across all good/confident/random summaries of the 1000 questions and then rank them based on the average (like in a benchmark).

| Metric | Gemma 3 12B | | Qwen 2.5 7B Instruct | |
|---|---|---|---|---|
| | Per Q | Avg | Per Q | Avg |
| Summarization | $0.44_{\pm 0.03}$ | $0.80_{\pm 0.00}$ | $0.54_{\pm 0.06}$ | $0.80_{\pm 0.00}$ |
| LM Judge | $\mathbf{0.80}_{\pm 0.02}$ | $\mathbf{1.00}_{\pm 0.00}$ | $0.56_{\pm 0.06}$ | $0.72_{\pm 0.01}$ |
| Opt. Transport | $0.69_{\pm 0.02}$ | $0.88_{\pm 0.01}$ | $\mathbf{0.60}_{\pm 0.06}$ | $0.81_{\pm 0.00}$ |
| Embedding | $0.29_{\pm 0.03}$ | $0.18_{\pm 0.02}$ | $0.02_{\pm 0.08}$ | $0.40_{\pm 0.00}$ |
| SR-logl | $0.58_{\pm 0.03}$ | $1.00_{\pm 0.00}$ | $0.64_{\pm 0.06}$ | $1.00_{\pm 0.00}$ |
| SR-PMI | $-0.03_{\pm 0.03}$ | $-0.20_{\pm 0.00}$ | $0.29_{\pm 0.08}$ | $0.47_{\pm 0.01}$ |
| SR-sampling-free | $0.57_{\pm 0.03}$ | $0.83_{\pm 0.00}$ | $-0.16_{\pm 0.09}$ | $-0.42_{\pm 0.01}$ |
| SR-P(True) | $0.66_{\pm 0.03}$ | $\mathbf{1.00}_{\pm 0.00}$ | $0.08_{\pm 0.07}$ | $0.20_{\pm 0.00}$ |
| SelfReflect | $0.66_{\pm 0.03}$ | $\mathbf{1.00}_{\pm 0.00}$ | $0.56_{\pm 0.07}$ | $\mathbf{0.93}_{\pm 0.01}$ |

# I   USER STUDY DETAILS

User studies were carried out using TryRating, with five raters per task. Raters were allowed to rate as many tasks as they wanted. All raters were US-based English speakers, and were paid $18/hr.

The users were presented with the instructions shown in Fig. 16, which included two examples with hand-crafted summaries (for space reasons, we include only one summary here).

---

**Task Guidelines**

You will be shown a question, and ten survey responses to that question.

You will then be shown two possible summaries of the survey responses. Your task is to pick the summary that best summarizes the ten individual responses.

Some things to consider:

- Does the summary reflect all viewpoints in the responses? Is it clear which viewpoints are more or less prevalent? A great summary reflects all viewpoints, and makes clear which are more or less common viewpoints. A good summary reflects all viewpoints, but might not indicate which are more or less common. An OK summary only reflects some viewpoints. A bad summary does not reflect any of the viewpoints.
- Does the summary accurately capture the viewpoints? A great summary accurately captures the information in each respose. A poor summary either excludes key information, or adds information that is not mentioned in the survey responses.
- Is the summary easy to read? If two summaries are equally informative about the ten survey answers, the better summary will be the one that is easier to read and understand.

Do **not** consider any prior knowledge you have about the question! Your task is not to assess which summary is a correct answer to the question, it is to assess which summary best represents the sampled responses (which might not correctly answer the question).

Here are a few examples:

---

**Question: where do peaches come from in the us?**

- Peaches come from several states in the US, primarily Georgia, California, and South Carolina.
- Peaches in the U.S. primarily come from California, Georgia, and South Carolina.
- Peaches primarily come from California, followed by South Carolina and Georgia.
- Peaches in the U.S. primarily come from California, Georgia, and South Carolina.
- Peaches are primarily grown in states like Georgia, California, and South Carolina in the US.
- Peaches primarily come from California and Georgia in the USA.
- Peaches in the U.S. primarily come from Georgia, California, and South Carolina.
- Peaches in the US primarily come from California, Georgia, and South Carolina.
- Peaches are primarily produced in California, which is the leading producer in the U.S.
- Peaches primarily come from California, which is the leading producer in the U.S.

|  **Summary 1** | **Summary 2** |
|---|---|
| Peaches primarily come from California, followed by Georgia and South Carolina. | Peaches primarily come from California. |
| Summary 1 | Summary 2 |

In the example above, Summary 1 is the better summary. It identifies all states mentioned in the responses, while indicating that California is the primary choice. In contrast, Summary 2 does not mention South Carolina or Georgia.

---

Figure 16: Instructions for user study (truncated; actual instructions contained a second example, which we have cut here for space).

To ensure quality responses, we constructed twenty "golden answer" tasks, where the summaries were manually constructed to either fit the definition of "good", "nearly good", or "bad" summaries, as described in Section 4.1. Ten of these questions were given as an entrance exam, with raters required to answer the golden answer in 80% of tasks to proceed. The remaining ten questions were periodically included as verification checks. A total of 215 raters passed the entrance exam and contributed ratings.

Confidence intervals were calculated using 100 bootstrapped samples.

# J  AUTOMATIC SUMMARY GENERATION

## J.1  EXPERIMENTAL DETAILS

In this section, we denote sampling parameters as `{T=1, topp=1, topk=None, minp=0}`.

**Non-reasoning/RLHF models**   For the models in Table 4, the same model is used for both generating the answers, and generating the summaries. We sample answers with `{T=1, topp=1, topk=None, minp=0}`. For summaries generation, we use greedy decoding, i.e., `{T=0, topp=1, topk=None, minp=0}`.

**Reasoning/RLVR models**   For the models in Table 5, we also want to sample the answers without reasoning and make use of the reasoning only for the summaries generation. For Qwen3, it is possible to suppress the reasoning with `tokenizer.apply_chat_template(...,  enable_thinking=False)`. For QwQ-32B and DeepSeek-R1-Distill-Qwen2.5-32B, we did not find a way to suppress reasoning and hence we sample the answers from Qwen2.5-32B-Instruct, which is the RLHF model which served as a base model for the RLVR training. We sample answers with `{T=1, topp=1, topk=None, minp=0}`.

For the *Greedy* summary generation, we use non-reasoning greedy decoding with the same model as for the answers generation, i.e., for rows QwQ-32B and DeepSeek-R1-Distill-Qwen2.5-32B, we use Qwen2.5-32B-Instruct.

For the *Basic* and *Sample & Summarize* summaries generation, we use the reasoning mode of each respective model. Unlike for RLHF models, we stray away from using greedy decoding for summary generation, because the creators of the reasoning models we use warn that the use of greedy decoding with reasoning "as it can lead to performance degradation and endless repetitions". Hence, for each model we use the respective recommended sampling parameters available on their respective HuggingFace model card.

## J.2  PROMPTS USED

---

**Prompt for the *basic* summary generation method in Section 5.**

```
Please respond to the following question '{question}'.

Your goal is to summarize all possible answers to this question:
     • If there are multiple possible answers, the summarized answer
       should mention the main possible answers.  However, you do
       not have to list possibilities that are too unlikely.
     • If some possibilities are more likely than others, delineate
       which possibilities are more more likely by using words like
       "most likely" and "could also be".
     • The format of the summarized answer should be the same as a
       normal answer.
     • If there is only clear answer to the question, just provide
       that answer, without hedging across possibilities.
Please provide the summarized answer.
```

---

**Prompt for the *CoT* summary generation method in Section 5.**

```
Please respond to the following question '{question}'.

Your goal is to first reason about all possible answers to this
question and then summarize them into a final answer:
     • Reflect on whether there are multiple possible answers to
       this question.
```

---

```
    • If there are multiple possible answers, the summarized answer
      should mention the main possible answers.  However, you do
      not have to list possibilities that are too unlikely.

    • If some possibilities are more likely than others, delineate
      which possibilities are more more likely by using words like
      "most likely" and "could also be".

    • The format of the summarized answer should be the same as a
      typical answer and be stand-alone.

    • If there is only clear answer to the question, just provide
      that answer, without hedging across possibilities.

The output should be in the following format:
Reasoning:  [REASONING ABOUT WHICH POSSIBLITIES THERE ARE AND HOW
LIKELY THEY ARE]
Summary:  [SUMMARIZED ANSWER]

Please provide the reasoning and then the summarized answer.
```

## J.3 RESULTS PER DATASET

Table 12: SelfReflect score $\downarrow$ ($\times 10^{-3}$, rounded for readability) for the SimpleQA dataset, averaged across 1000 questions. The results in small font are relative to *Greedy*.

| Model | $p_\theta(A \mid q)$ unimodal | Single-decoding methods | | | Sample & summarize | |
|---|---|---|---|---|---|---|
| | | Greedy | Basic | CoT | $N = 10$ | $N = 20$ |
| Qwen2.5 0.5B Instruct (Yang et al., 2024a) | 1% | 98 | $97_{-1}$ | $95_{-3}$ | $99_{+1}$ | $99_{+1}$ |
| Qwen2.5 1.5B Instruct (Yang et al., 2024a) | 2% | 98 | $97_{\ 0}$ | $93_{-5}$ | $90_{-8}$ | $89_{-8}$ |
| Qwen2.5 3B Instruct (Yang et al., 2024a) | 5% | 101 | $101_{\ 0}$ | $99_{-1}$ | $93_{-8}$ | $91_{-9}$ |
| Qwen2.5 7B Instruct (Yang et al., 2024a) | 15% | 98 | $102_{+4}$ | $102_{+3}$ | $93_{-5}$ | $92_{-6}$ |
| Qwen2.5 14B Instruct (Yang et al., 2024a) | 23% | 96 | $101_{+5}$ | $102_{+7}$ | $88_{-8}$ | $87_{-9}$ |
| Qwen2.5 32B Instruct (Yang et al., 2024a) | 17% | 103 | $108_{+5}$ | $110_{+8}$ | $95_{-8}$ | $94_{-8}$ |
| Qwen2.5 72B Instruct (Yang et al., 2024a) | 18% | 95 | $99_{+3}$ | $100_{+5}$ | $88_{-8}$ | $87_{-9}$ |
| Phi 4 (Abdin et al., 2024) | 3% | 99 | $99_{\ 0}$ | $97_{-2}$ | $89_{-10}$ | $87_{-12}$ |
| Ministral 8B Instruct 2410 (Jiang et al., 2024) | 1% | 117 | $116_{\ 0}$ | $114_{-2}$ | $109_{-7}$ | $107_{-9}$ |
| Llama 3.1 70B Instruct (Meta AI, 2024a) | 16% | 97 | $97_{\ 0}$ | $97_{+1}$ | $90_{-6}$ | $90_{-7}$ |
| Llama 3.3 70B Instruct (Meta AI, 2024b) | 29% | 100 | $103_{+3}$ | $113_{+13}$ | $92_{-8}$ | $91_{-9}$ |
| Llama 4 Scout 17B 16e Instruct (Meta AI, 2025) | 20% | 95 | $100_{+5}$ | $103_{+8}$ | $89_{-6}$ | $87_{-7}$ |
| Gemma 3 1B Instruct (Gemma Team et al., 2025) | 17% | 123 | $134_{+11}$ | $135_{+11}$ | $124_{\ 0}$ | $118_{-6}$ |
| Gemma 3 4B Instruct (Gemma Team et al., 2025) | 25% | 118 | $135_{+16}$ | $138_{+20}$ | $108_{-10}$ | $106_{-12}$ |
| Gemma 3 12B Instruct (Gemma Team et al., 2025) | 35% | 117 | $129_{+12}$ | $135_{+18}$ | $112_{-5}$ | $112_{-5}$ |
| Gemma 3 27B Instruct (Gemma Team et al., 2025) | 49% | 109 | $124_{+16}$ | $134_{+25}$ | $103_{-5}$ | $101_{-7}$ |

Table 13: SelfReflect score $\downarrow$ ($\times 10^{-3}$, rounded for readability) for the Natural Questions dataset, averaged across 1000 questions. The results in small font are relative to *Greedy*.

| Model | $p_\theta(A \mid q)$ unimodal | Single-decoding methods | | | Sample & summarize | |
|---|---|---|---|---|---|---|
| | | Greedy | Basic | CoT | $N = 10$ | $N = 20$ |
| Qwen2.5 0.5B Instruct (Yang et al., 2024a) | 5% | 92 | $90_{-1}$ | $90_{-2}$ | $92_{\ 0}$ | $92_{\ 0}$ |
| Qwen2.5 1.5B Instruct (Yang et al., 2024a) | 13% | 90 | $91_{+1}$ | $89_{-1}$ | $85_{-5}$ | $84_{-6}$ |
| Qwen2.5 3B Instruct (Yang et al., 2024a) | 30% | 95 | $96_{+1}$ | $97_{+2}$ | $88_{-7}$ | $87_{-8}$ |
| Qwen2.5 7B Instruct (Yang et al., 2024a) | 32% | 94 | $98_{+4}$ | $101_{+7}$ | $90_{-5}$ | $89_{-5}$ |
| Qwen2.5 14B Instruct (Yang et al., 2024a) | 56% | 91 | $97_{+6}$ | $100_{+9}$ | $86_{-5}$ | $85_{-6}$ |
| Qwen2.5 32B Instruct (Yang et al., 2024a) | 50% | 94 | $100_{+6}$ | $104_{+10}$ | $89_{-5}$ | $89_{-5}$ |
| Qwen2.5 72B Instruct (Yang et al., 2024a) | 48% | 89 | $94_{+5}$ | $98_{+9}$ | $84_{-5}$ | $83_{-6}$ |
| Phi 4 (Abdin et al., 2024) | 36% | 89 | $88_{-1}$ | $92_{+3}$ | $83_{-6}$ | $82_{-7}$ |
| Ministral 8B Instruct 2410 (Jiang et al., 2024) | 17% | 101 | $99_{-1}$ | $99_{-2}$ | $95_{-6}$ | $94_{-7}$ |
| Llama 3.3 70B Instruct (Meta AI, 2024b) | 68% | 91 | $96_{+5}$ | $104_{+13}$ | $86_{-5}$ | $85_{-6}$ |
| Llama 4 Scout 17B 16e Instruct (Meta AI, 2025) | 59% | 90 | $96_{+6}$ | $104_{+14}$ | $88_{-2}$ | $87_{-4}$ |
| Gemma 3 1B Instruct (Gemma Team et al., 2025) | 22% | 113 | $126_{+13}$ | $127_{+14}$ | $113_{+1}$ | $108_{-5}$ |
| Gemma 3 4B Instruct (Gemma Team et al., 2025) | 56% | 106 | $123_{+17}$ | $128_{+22}$ | $100_{-6}$ | $99_{-7}$ |
| Gemma 3 12B Instruct (Gemma Team et al., 2025) | 57% | 103 | $118_{+14}$ | $121_{+17}$ | $99_{-5}$ | $99_{-5}$ |
| Gemma 3 27B Instruct (Gemma Team et al., 2025) | 72% | 100 | $116_{+15}$ | $121_{+21}$ | $98_{-3}$ | $97_{-3}$ |

Table 14: SelfReflect score $\downarrow$ ($\times 10^{-3}$, rounded for readability) for the TriviaQA dataset, averaged across 1000 questions. The results in small font are relative to *Greedy*.

| Model | $p_\theta(A \mid q)$ unimodal | Single-decoding methods | | | Sample & summarize | |
|---|---|---|---|---|---|---|
| | | Greedy | Basic | CoT | $N = 10$ | $N = 20$ |
| Qwen2.5 0.5B Instruct (Yang et al., 2024a) | 15% | 97 | $96_{-1}$ | $96_{-1}$ | $98_{+1}$ | $98_{\ 0}$ |
| Qwen2.5 1.5B Instruct (Yang et al., 2024a) | 36% | 93 | $94_{+1}$ | $93_{\ 0}$ | $87_{-6}$ | $87_{-7}$ |
| Qwen2.5 3B Instruct (Yang et al., 2024a) | 45% | 97 | $99_{+2}$ | $100_{+2}$ | $91_{-6}$ | $90_{-7}$ |
| Qwen2.5 7B Instruct (Yang et al., 2024a) | 60% | 95 | $98_{+3}$ | $100_{+5}$ | $91_{-3}$ | $91_{-4}$ |
| Qwen2.5 14B Instruct (Yang et al., 2024a) | 76% | 88 | $93_{+5}$ | $94_{+6}$ | $85_{-3}$ | $84_{-4}$ |
| Qwen2.5 32B Instruct (Yang et al., 2024a) | 79% | 92 | $98_{+6}$ | $101_{+8}$ | $89_{-3}$ | $89_{-3}$ |
| Qwen2.5 72B Instruct (Yang et al., 2024a) | 85% | 87 | $89_{+3}$ | $90_{+4}$ | $84_{-3}$ | $83_{-4}$ |
| Phi 4 (Abdin et al., 2024) | 69% | 89 | $88_{-1}$ | $89_{\ 0}$ | $84_{-5}$ | $83_{-6}$ |
| Ministral 8B Instruct 2410 (Jiang et al., 2024) | 56% | 104 | $103_{\ 0}$ | $103_{-1}$ | $99_{-4}$ | $98_{-5}$ |
| Llama 3.1 70B Instruct (Meta AI, 2024a) | 82% | 89 | $88_{-1}$ | $89_{\ 0}$ | $85_{-4}$ | $85_{-5}$ |
| Llama 3.3 70B Instruct (Meta AI, 2024b) | 92% | 91 | $93_{+2}$ | $95_{+4}$ | $89_{-2}$ | $88_{-3}$ |
| Llama 4 Scout 17B 16e Instruct (Meta AI, 2025) | 81% | 89 | $92_{+2}$ | $96_{+6}$ | $87_{-2}$ | $86_{-3}$ |
| Gemma 3 1B Instruct (Gemma Team et al., 2025) | 40% | 112 | $127_{+14}$ | $125_{+12}$ | $113_{\ 0}$ | $108_{-4}$ |
| Gemma 3 4B Instruct (Gemma Team et al., 2025) | 76% | 101 | $114_{+13}$ | $118_{+17}$ | $96_{-5}$ | $94_{-6}$ |
| Gemma 3 12B Instruct (Gemma Team et al., 2025) | 84% | 95 | $103_{+8}$ | $107_{+13}$ | $94_{-1}$ | $93_{-1}$ |
| Gemma 3 27B Instruct (Gemma Team et al., 2025) | 93% | 91 | $99_{+8}$ | $104_{+13}$ | $91_{\ 0}$ | $91_{\ 0}$ |

Table 15: Runtime of generating different summaries in seconds per prompt per GPU, averaged across all three datasets with 1000 questions each. Note that some models are sharded across multiple GPUs – in this case, to represent their total computational requirements, we report the summed runtime of all GPUs, i.e., although a prompt may run through in one second using four GPUs, we will count it as four seconds total. In small font, relative comparisons w.r.t. *Greedy*.

| Model | Single-decoding methods | | | Sample & summarize |
|---|---|---|---|---|
| | Greedy | Basic | CoT | $N=10$ |
| Qwen2.5 0.5B Instruct (Yang et al., 2024a) | 0.93 | $1.26_{\times 1.35}$ | $1.26_{\times 1.36}$ | $1.79_{\times 1.93}$ |
| Qwen2.5 1.5B Instruct (Yang et al., 2024a) | 0.94 | $0.91_{\times 0.96}$ | $1.29_{\times 1.37}$ | $1.89_{\times 2.01}$ |
| Qwen2.5 3B Instruct (Yang et al., 2024a) | 0.84 | $0.85_{\times 1.02}$ | $1.00_{\times 1.20}$ | $1.82_{\times 2.18}$ |
| Qwen2.5 7B Instruct (Yang et al., 2024a) | 0.80 | $0.84_{\times 1.05}$ | $1.09_{\times 1.36}$ | $1.91_{\times 2.38}$ |
| Qwen2.5 14B Instruct (Yang et al., 2024a) | 0.96 | $1.01_{\times 1.05}$ | $1.33_{\times 1.38}$ | $2.44_{\times 2.53}$ |
| Qwen2.5 32B Instruct (Yang et al., 2024a) | 1.11 | $1.22_{\times 1.09}$ | $1.72_{\times 1.54}$ | $3.42_{\times 3.07}$ |
| Qwen2.5 72B Instruct (Yang et al., 2024a) | 1.68 | $1.63_{\times 0.97}$ | $3.48_{\times 2.07}$ | $4.41_{\times 2.62}$ |
| Phi 4 14B (Abdin et al., 2024) | 1.09 | $1.02_{\times 0.94}$ | $1.43_{\times 1.31}$ | $2.96_{\times 2.71}$ |
| Ministral 8B Instruct 2410 (Jiang et al., 2024) | 0.91 | $0.91_{\times 1.00}$ | $1.04_{\times 1.15}$ | $1.82_{\times 2.00}$ |
| Llama 3.1 70B Instruct (Meta AI, 2024a) | 4.16 | $4.45_{\times 1.07}$ | $10.58_{\times 2.54}$ | $10.59_{\times 2.55}$ |
| Llama 3.3 70B Instruct (Meta AI, 2024b) | 3.54 | $3.17_{\times 0.89}$ | $4.25_{\times 1.20}$ | $7.23_{\times 2.04}$ |
| Llama 4 Scout 17B 16e Instruct (Meta AI, 2025) | 3.45 | $3.92_{\times 1.13}$ | $5.67_{\times 1.64}$ | $9.05_{\times 2.62}$ |
| Gemma 3 1B Instruct (Gemma Team et al., 2025) | 0.89 | $0.86_{\times 0.97}$ | $0.93_{\times 1.04}$ | $1.74_{\times 1.96}$ |
| Gemma 3 4B Instruct (Gemma Team et al., 2025) | 0.95 | $0.93_{\times 0.98}$ | $1.09_{\times 1.15}$ | $1.96_{\times 2.06}$ |
| Gemma 3 12B Instruct (Gemma Team et al., 2025) | 1.12 | $1.14_{\times 1.02}$ | $1.54_{\times 1.38}$ | $2.43_{\times 2.17}$ |
| Gemma 3 27B Instruct (Gemma Team et al., 2025) | 1.32 | $1.36_{\times 1.03}$ | $2.26_{\times 1.71}$ | $3.09_{\times 2.34}$ |

Table 16: Character length of different summaries in seconds per prompt per GPU, averaged across all three datasets with 1000 questions each. In small font, relative comparisons w.r.t. *Greedy*.

| Model | Single-decoding methods | | | Sample & summarize |
|---|---|---|---|---|
| | Greedy | Basic | CoT | $N=10$ |
| Qwen2.5 0.5B Instruct (Yang et al., 2024a) | 130.10 | $699.93_{\times 5.38}$ | $896.85_{\times 6.89}$ | $155.39_{\times 1.19}$ |
| Qwen2.5 1.5B Instruct (Yang et al., 2024a) | 152.44 | $151.44_{\times 0.99}$ | $528.68_{\times 3.47}$ | $310.16_{\times 2.03}$ |
| Qwen2.5 3B Instruct (Yang et al., 2024a) | 81.55 | $168.60_{\times 2.07}$ | $254.03_{\times 3.11}$ | $274.03_{\times 3.36}$ |
| Qwen2.5 7B Instruct (Yang et al., 2024a) | 93.16 | $149.61_{\times 1.61}$ | $198.33_{\times 2.13}$ | $178.72_{\times 1.92}$ |
| Qwen2.5 14B Instruct (Yang et al., 2024a) | 92.93 | $170.49_{\times 1.83}$ | $245.97_{\times 2.65}$ | $203.53_{\times 2.19}$ |
| Qwen2.5 32B Instruct (Yang et al., 2024a) | 64.53 | $145.30_{\times 2.25}$ | $186.80_{\times 2.89}$ | $138.33_{\times 2.14}$ |
| Qwen2.5 72B Instruct (Yang et al., 2024a) | 97.30 | $157.04_{\times 1.61}$ | $247.99_{\times 2.55}$ | $177.61_{\times 1.83}$ |
| Phi 4 14B (Abdin et al., 2024) | 124.85 | $203.35_{\times 1.63}$ | $281.96_{\times 2.26}$ | $273.05_{\times 2.19}$ |
| Ministral 8B Instruct 2410 (Jiang et al., 2024) | 49.15 | $100.54_{\times 2.05}$ | $179.57_{\times 3.65}$ | $130.08_{\times 2.65}$ |
| Llama 3.1 70B Instruct (Meta AI, 2024a) | 168.64 | $267.63_{\times 1.59}$ | $499.54_{\times 2.96}$ | $225.05_{\times 1.33}$ |
| Llama 3.3 70B Instruct (Meta AI, 2024b) | 113.47 | $229.45_{\times 2.02}$ | $277.31_{\times 2.44}$ | $152.70_{\times 1.35}$ |
| Llama 4 Scout 17B 16e Instruct (Meta AI, 2025) | 132.81 | $275.57_{\times 2.07}$ | $220.68_{\times 1.66}$ | $214.80_{\times 1.62}$ |
| Gemma 3 1B Instruct (Gemma Team et al., 2025) | 22.66 | $38.91_{\times 1.72}$ | $183.15_{\times 8.08}$ | $61.71_{\times 2.72}$ |
| Gemma 3 4B Instruct (Gemma Team et al., 2025) | 26.45 | $80.66_{\times 3.05}$ | $113.46_{\times 4.29}$ | $101.16_{\times 3.83}$ |
| Gemma 3 12B Instruct (Gemma Team et al., 2025) | 28.51 | $102.76_{\times 3.60}$ | $174.70_{\times 6.13}$ | $70.67_{\times 2.48}$ |
| Gemma 3 27B Instruct (Gemma Team et al., 2025) | 37.77 | $137.62_{\times 3.64}$ | $192.39_{\times 5.09}$ | $70.97_{\times 1.88}$ |

# K  EXPERIMENT DETAILS OF COT DEEP DIVE

## K.1  RESULTS PER DATASET

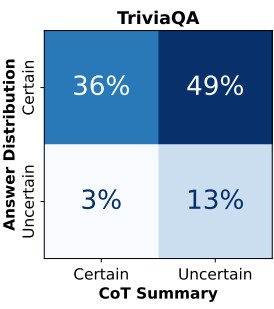 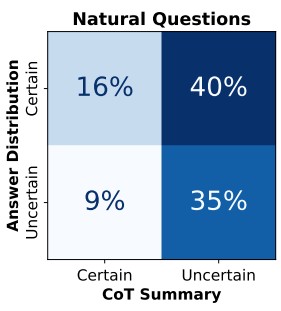 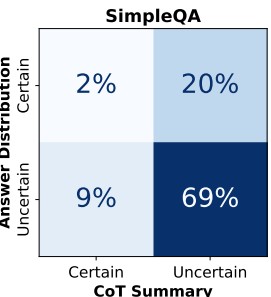

Figure 17: Confusion matrices between certainty of *CoT* summaries vs. actual answer distributions for Qwen2.5 72B Instruct. Judged by Gemini 2.0 Flash for 500 random questions per dataset.

We show the results per dataset in Fig. 17. By just looking at the marginals of the answer distribution, we can infer that the question difficulty increases from TriviaQA to Natural Questions to SimpleQA, as the number of questions with uncertain answer distributions increases from 16% to 44% to 78% of questions. However, the majority of the *CoT* summaries are uncertain for all three datasets (for 62%, 75%, and 89% of questions respectively), meaning that *CoT* is *underconfident* on TriviaQA and Natural Quesions—i.e., it suggests answers that do not have high probability under the true distribution. A balance is clearly needed. The *Greedy* method, by contrast, is overconfident. While it outperforms *CoT* on average, it underperforms *Sample & Summarize* since it will only present a single option. In a similar way, simply making *CoT* more confident would likely lead to detrimental effects on harder questions. Already as-is, there are relevant shares of questions in SimpleQA and Natural Questions where *CoT* is certain while the actual answer distribution is not (9% each). Generating self-reflective summaries that are truthful to the model's internal uncertainty thus remains a challenging task.

## K.2  PROMPTS USED TO CLASSIFY CERTAINTY VS. UNCERTAINTY

We used Gemini 2.0 Flash to judge if a summary mentions multiple semantically different answers or not, and likewise if samples from the distributions do so. In particular, we provide it with the corresponding context, and ask it to classify into three classes: A) Fully certain, B) not fully certain, C) Completely uncertain. We then aggregate B and C to form the general "uncertain" category (we found this to give better results than to directly give a binary task to Gemini, because it allows us to explicitly tell how to handle the "not fully certain" edge case). We manually verified that the classification is qualitatively correct. We show the prompts below, along with qualitative examples of how Gemini classified each summary and each answer distribution.

> Gemini 2.0 Flash prompt to classify *CoT* summary in Fig. 4. We call A. 'Certain' and group B. and C. into 'Uncertain'.
>
> ```
> Below, you are given an answer to the question '{question}'.
>
> Your goal is to classify which type of answer this is:
> A. The answer is certain, it only mentions one answer option.
> B. The answer is not fully certain.  It might mention one or two
> further answer options but judges them as less likely.
> C. The answer is very uncertain.  It mentions many mutually
> exclusive answer options, without a clear single most likely
> answer.
>
> Ignore differences in form and style.  You are only supposed to
> judge the answer semantically.
> ```

```
Here is the answer:  {CoT summary}

Please respond with the category of what type of this answer this
is.  Respond only with A, B, or C.
```

**Gemini 2.0 Flash prompt to classify answer distribution samples in Fig. 4. We call A. 'Certain' and group B. and C. into 'Uncertain'**

```
Below, you are given {n_answers} individual answers to the question
'{question}'.  These {n_answers} answers can be seen as samples from
an answer distribution.

Your goal is to classify which type of distribution this is:
A. The answers all do not contradict each other, up to one or two
that differ from the majority answer.
B. The answers give multiple mutually exclusive answer options, but
there is one answer option that is given in the majority of cases.
C. The answers give multiple mutually exclusive answer options, and
they are almost all different, without a clear majority answer.

The answers will have some natural variability.  Ignore differences
in form and style.  You are only supposed to judge if answer
options are semantically different.

Here are the {n_answers} answers:
x_1 = '{answer}'
...
x_{n_answers} = '{answer}'

Please respond with the category of what type of this distribution
this is.  Respond only with A, B, or C.
```

**Examples for bottom right quadrant: CoT classified as uncertain, answer distribution uncertain**

**Question:** What day, month, and year was the municipality of Valdivia, Antioquia, Colombia, founded?

**CoT summary:** "The most likely founding date of the municipality of Valdivia, Antioquia, Colombia, is July 16, 1679. However, some sources suggest it could also be August 15, 1679, or there might be earlier records of a settlement in the area."

**Answer distribution:**

- "Valdivia, Antioquia, Colombia, was founded on July 29, 1854.",
- "Valdivia, Antioquia, Colombia was founded on August 5, 1836.",
- "Valdivia, Antioquia, Colombia, was founded on July 7, 1861.",
- "Valdivia, Antioquia, Colombia, was founded on July 3, 1616.",
- "Valdivia, Antioquia, Colombia was founded on July 12, 1854.",
- "Valdivia, Antioquia, Colombia, was founded on January 27, 1616.",
- "Valdivia, Antioquia, Colombia was founded on August 5, 1645.",
- "Valdivia, Antioquia, Colombia, was founded on July 1, 1886.",
- "Valdivia, Antioquia, Colombia, was founded on May 19, 1849.",
- "Valdivia, Antioquia, Colombia, was founded on January 19, 1886."

**Question:** After the War of 1812, Edward William McBride (1791-1834) worked as what for the king's printer, John Cameron, on the York Gazette until April 1815?

**CoT summary:** "Edward William McBride most likely worked as a printer's assistant for John Cameron at the York Gazette, but he could also have been a typesetter or proofreader."

**Answer distribution:**

- "Edward William McBride worked as a printer for John Cameron on the York Gazette until April 1815.",
- "Edward William McBride worked as a printer for John Cameron on the York Gazette until April 1815.",
- "Edward William McBride worked as a printer for John Cameron on the York Gazette until April 1815.",
- "Edward William McBride worked as a printer for the king's printer, John Cameron, on the York Gazette until April 1815.",
- "Edward William McBride worked as a printer for John Cameron on the York Gazette until April 1815.",
- "Edward William McBride worked as a printer for John Cameron on the York Gazette until April 1815.",
- "Edward William McBride worked as a compositor for John Cameron on the York Gazette until April 1815.",
- "Edward William McBride worked as a compositor for John Cameron on the York Gazette until April 1815.",
- "Edward William McBride worked as a printer for John Cameron on the York Gazette until April 1815.",
- "Edward William McBride worked as a printer for John Cameron on the York Gazette until April 1815."

**Examples for top left quadrant: CoT classified as certain, answer distribution certain**

**Question:** who hosted and won the inagural world cup?

**CoT summary:** "Uruguay hosted and won the inaugural FIFA World Cup in 1930."

**Answer distribution:**

- "The inaugural World Cup was hosted by Uruguay in 1930, and Uruguay also won the tournament.",
- "The inaugural World Cup was hosted by Uruguay in 1930, and Uruguay also won the tournament.",
- "The inaugural World Cup was hosted by Uruguay in 1930, and Uruguay also won the tournament.",
- "The inaugural World Cup was hosted by Uruguay in 1930, and Uruguay also won the tournament.",
- "The inaugural World Cup was hosted by Uruguay in 1930, and Uruguay also won the tournament.",
- "The inaugural World Cup was hosted by Uruguay in 1930, and Uruguay also won the tournament.",
- "The inaugural World Cup was hosted by Uruguay in 1930, and Uruguay also won the tournament.",
- "The inaugural World Cup was hosted by Uruguay in 1930, and Uruguay also won the tournament.",
- "The inaugural World Cup was hosted by Uruguay in 1930, and Uruguay also won the tournament.",
- "The inaugural FIFA World Cup was hosted by Uruguay in 1930, and Uruguay also won the tournament."

**Question:** Who received the IEEE Frank Rosenblatt Award in 2019?

**CoT summary:** "Yann LeCun received the IEEE Frank Rosenblatt Award in 2019."

**Answer distribution:**

- "Yann LeCun received the IEEE Frank Rosenblatt Award in 2019.",
- "Yann LeCun received the IEEE Frank Rosenblatt Award in 2019.",
- "Yann LeCun received the IEEE Frank Rosenblatt Award in 2019.",
- "Yann LeCun received the IEEE Frank Rosenblatt Award in 2019.",
- "Yann LeCun received the IEEE Frank Rosenblatt Award in 2019.",
- "Yann LeCun received the IEEE Frank Rosenblatt Award in 2019.",
- "Yann LeCun received the IEEE Frank Rosenblatt Award in 2019.",
- "Yann LeCun received the IEEE Frank Rosenblatt Award in 2019.",
- "Yann LeCun received the IEEE Frank Rosenblatt Award in 2019.",
- "Yann LeCun received the IEEE Frank Rosenblatt Award in 2019."

**Examples for top right quadrant: CoT classified as uncertain, answer distribution certain**

**Question:** During what year did Tatsuo Miyajima have his first solo exhibition?

**CoT summary:** "The most likely year for Tatsuo Miyajima's first solo exhibition is 1988, as this is the year most commonly cited in reliable sources. However, there is a possibility that it could have been as early as 1986, though this is less well-documented."

**Answer distribution:**

- "1988",
- "Tatsuo Miyajima had his first solo exhibition in 1988.",
- "Tatsuo Miyajima had his first solo exhibition in 1988.",
- "1988",
- "1988",
- "1988",
- "1988",
- "1988",
- "Tatsuo Miyajima had his first solo exhibition in 1988.",
- "1988"

**Question:** At which University of Puerto Rico campus did New York State Representative Nydia Velázquez work as a professor from 1976 to 1981?

**CoT summary:** "The most likely campus where Nydia Velázquez worked as a professor from 1976 to 1981 is the University of Puerto Rico, Río Piedras, given its prominence and alignment with her academic and political background. However, it could also be the University of Puerto Rico, Mayagüez or Cayey, though these are less likely."

**Answer distribution:**

- "University of Puerto Rico, Río Piedras Campus.",
- "University of Puerto Rico, Río Piedras Campus.",
- "University of Puerto Rico, Río Piedras Campus.",
- "University of Puerto Rico, Río Piedras Campus.",
- "University of Puerto Rico, Río Piedras Campus.",
- "University of Puerto Rico, Río Piedras Campus.",
- "University of Puerto Rico, Rio Piedras Campus.",
- "University of Puerto Rico, Río Piedras Campus.",
- "University of Puerto Rico, Río Piedras Campus.",
- "University of Puerto Rico, Río Piedras Campus."

**Examples for bottom left quadrant: CoT classified as certain, answer distribution uncertain**

**Question:** Danger Island, The Micro Ventures, The Arabian Knights and The Three Musketeers were short segments in which classic children's TV programme?

**CoT summary:** The most likely answer is The Banana Splits Adventure Hour, as it featured a mix of different adventure segments that could include titles like Danger Island, The Micro Ventures, The Arabian Knights, and The Three Musketeers.

**Answer distribution:**

- These segments were part of the classic children's TV programme "The Banana Splits Adventure Hour."

- The programme you're referring to is "The Banana Splits Adventure Hour."

- The classic children's TV programme that featured these short segments is "The Banana Splits Adventure Hour."

- These segments were part of the classic children's TV programme "The Banana Splits Adventure Hour."

- The programme you're referring to is "The Quatermass Experiment," but it seems there might be a mix-up. Those segments are actually from the classic children's TV show "Vision On," which aired in the UK from 1964 to 1976.

- These were segments in the children's TV show "The Storyteller."

- These segments were part of the classic children's TV programme "The Banana Splits Adventure Hour."

- These segments were part of the classic children's TV programme "The Dame's House," also known as "Dame's House" or "The hastily assembled Dame's house for children," which aired in the UK in the 1940s and 1950s. However, the most well-known incarnation featuring these segments was likely "The Goon Show," although it's important to note that "Danger Island" and "The Micro Ventures" were specific to "The Dame's House."

- The segments you mentioned were part of the classic children's TV program "The Banana Splits Adventure Hour."

- These segments were part of the classic children's TV programme "The Banana Splits Adventure Hour."

**Question:** Which 'A-road' passes through Preston and Harrogate?

**CoT summary:** There is no single A-road that passes through both Preston and Harrogate. The closest options are the A65, which passes through Harrogate, and the A683, which passes through Preston.

**Answer distribution:**

- The A65 passes through both Preston and Harrogate.

- The A59 road passes through both Preston and Harrogate.

- The A65 road passes through both Preston and Harrogate.

- The A59 road passes through Preston and Harrogate.

- The A65 road passes through both Preston and Harrogate.

- The A65 passes through both Preston and Harrogate.

- The A65 road passes through both Preston and Harrogate.

- The A59 road passes through Preston and Harrogate.

- The A65 passes through Preston and Harrogate.

- The A65 passes through both Preston and Harrogate.

## L   FINETUNING TO GENERATE SELF-REFLECTIVE SUMMARIES

Table 17 includes experiments with post-training to generate summarized uncertainties. Despite testing multiple setups, we have not found a strategy that trains the model to output summaries of its uncertainties. It seems to capture the style, and the results for certain train samples, but fails to find a generalizable mechanism. We see this as a potential for future studies.

Table 17: SelfReflect scores of models that have been supervised finetuned (SFT) and/or direct preference optimized (DPO) on sample-and-summarize summaries with LoRA adapters. Each epoch includes 10,000 example summaries. No approach consistently improves both in- and out-of-distribution over sampling greedily from Qwen 3 8B Instruct.

| Finetuning | Train Natural Q | Eval Natural Q | OOD TriviaQA | Train TriviaQA | Eval TriviaQA | OOD Natural Q |
|---|---|---|---|---|---|---|
| Qwen 3 8B Instruct | 93 | 94 | 95 | 98 | 95 | 94 |
| + SFT (5 epochs), lr 1e-4 | $90_{-3}$ | $92_{-2}$ | $96_{+1}$ | $96_{-2}$ | $96_{+1}$ | $92_{-2}$ |
| + SFT (10 epochs), lr 1e-4 | $89_{-4}$ | $94_{-0}$ | $98_{+3}$ | $94_{-4}$ | $97_{-2}$ | $94_{-0}$ |
| + SFT (15 epochs), lr 1e-4 | $88_{-5}$ | $94_{-0}$ | $97_{+2}$ | $92_{-6}$ | $97_{-2}$ | $94_{-0}$ |
| + SFT (20 epochs), lr 1e-4 | $87_{-6}$ | $93_{-1}$ | $98_{+3}$ | $92_{-6}$ | $98_{-3}$ | $94_{-0}$ |
| + DPO (5 epochs), lr 1e-5 | $95_{+2}$ | $95_{+1}$ | $97_{+2}$ | $99_{+1}$ | $97_{+2}$ | $94_{+0}$ |
| + DPO (10 epochs), lr 1e-5 | $95_{+2}$ | $95_{+1}$ | $97_{+2}$ | $100_{+2}$ | $97_{+2}$ | $95_{+1}$ |
| + DPO (15 epochs), lr 1e-5 | $96_{+3}$ | $96_{+2}$ | $97_{+2}$ | $100_{+2}$ | $97_{+2}$ | $95_{+1}$ |
| + DPO (20 epochs), lr 1e-5 | $96_{+3}$ | $96_{+2}$ | $97_{+2}$ | $99_{+1}$ | $97_{+2}$ | $95_{+1}$ |
| + SFT (5 ep.), lr 1e-4 + DPO (20 ep.), lr 1e-5 | $97_{+4}$ | $97_{+3}$ | $99_{+4}$ | $99_{+1}$ | $99_{+4}$ | $97_{+3}$ |
| + DPO, lr 1e-4 (20 epochs) | $99_{+6}$ | $98_{+4}$ | $102_{+7}$ | $102_{+4}$ | $100_{+5}$ | $98_{+4}$ |
| + DPO, lr 1e-4, $\beta = 0$ (20 epochs) | $93_{-0}$ | $93_{-1}$ | $95_{-0}$ | $98_{-0}$ | $95_{-0}$ | $93_{-1}$ |
| + DPO, lr 1e-5, $\beta = 0.5$ (20 epochs) | $94_{+1}$ | $94_{-0}$ | $96_{+1}$ | $98_{-0}$ | $96_{+1}$ | $93_{-1}$ |
| + SFT (5 epochs), lr 1e-4 + DPO, lr 1e-4 (20 epochs) | $97_{+4}$ | $97_{+3}$ | $99_{+4}$ | $99_{+1}$ | $99_{+4}$ | $97_{+3}$ |
| + SFT (5 epochs), lr 1e-4 + DPO, lr 1e-4, $\beta = 0$ (20 epochs) | $90_{-3}$ | $92_{-2}$ | $96_{+1}$ | $96_{-2}$ | $96_{+1}$ | $92_{-2}$ |
| + SFT (5 epochs), lr 1e-4 + DPO, lr 1e-5 (20 epochs) | $90_{-3}$ | $92_{-2}$ | $96_{+1}$ | $96_{-2}$ | $96_{+1}$ | $92_{-2}$ |
| + SFT (5 epochs), lr 1e-4 + DPO, lr 1e-5, $\beta = 0.5$ (20 epochs) | $92_{-1}$ | $94_{-0}$ | $96_{+1}$ | $96_{-2}$ | $96_{+1}$ | $93_{-1}$ |

## M   RESULTS ON RETRIEVAL-AUGMENTED GENERATION

The datasets we have used in Section 5 are closed-book question-answering datasets. This is to have an experimental setup in which there are sufficient questions in which the LLMs have non-unimodal output distributions and must actually summarize multiple options. However, SelfReflect can measure whether an LLM can summarize its current answer state no matter which previous context it is responding to. To show this, we run an additional experiment on HotpotQA (Yang et al., 2018), a retrieval-augmented generation (RAG) dataset in which each question is given along with background information from an according Wikipedia page.

Table 18: SelfReflect score $\downarrow$ ($\times 10^{-3}$ for readability) on HotpotQA, a RAG dataset. The results in small font are relative to *Greedy*.

| Model | Greedy | Basic | CoT | Sample & Summarize ($N = 20$) |
|---|---|---|---|---|
| Qwen 2.5 1.5B | 76 | $76_{-0}$ | $76_{-0}$ | $73_{-3}$ |
| Qwen 2.5 3B | 80 | $81_{+1}$ | $81_{+1}$ | $79_{-1}$ |
| Qwen 2.5 7B | 79 | $79_{-0}$ | $80_{+1}$ | $78_{-1}$ |
| Qwen 2.5 14B | 78 | $80_{+2}$ | $80_{+2}$ | $76_{-2}$ |
| Qwen 2.5 32B | 79 | $80_{+1}$ | $80_{+1}$ | $78_{-1}$ |
| Qwen 2.5 72B | 78 | $79_{+1}$ | $80_{+2}$ | $76_{-2}$ |
| Microsoft Phi 4 | 79 | $78_{-1}$ | $78_{-1}$ | $76_{-3}$ |
| Ministral 8B 2410 | 86 | $86_{-0}$ | $85_{-1}$ | $84_{-2}$ |
| Llama 3.1 70B | 80 | $79_{-1}$ | $79_{-1}$ | $78_{-2}$ |

The results are similar to the main paper results: Sample & Summarize is the only summary generator that consistently produces better summaries than just giving the greedy answer. Note that the gap to greedy is tighter here than in the main paper. This is because in RAG questions, the answer less uncertain, so that distributions are more often unimodal and no hedging is required.

Table 19: Sample & Summarize summaries that summarize an increasing number of sampled answers. Generated by Qwen 2.5 72B Instruct on TriviaQA questions.

| Answers used in Sample & Summarize | 10 | 20 | 30 | 40 | 50 | 60 | 70 | 80 | 90 | 100 |
|---|---|---|---|---|---|---|---|---|---|---|
| SelfReflect score ↓ ($\times 10^{-3}$) | 84 | 83 | 83 | 82 | 82 | 82 | 82 | 82 | 82 | 82 |
| Length (characters) | 93.33 | 104.37 | 113.22 | 120.46 | 123.97 | 127.27 | 133.82 | 135.28 | 136.42 | 137.58 |

## N  BEHAVIOR OF SAMPLE & SUMMARIZE

To get a better understanding of the best-working approach, sample & summarize, we extend its analysis here. First, we provide qualitative examples to understand how summary strings generated by it look like. We conduct this analysis on Qwen 2.5 72B Instruct on TriviaQA questions.

In 50% of the questions, the LLM is certain about its answer, thus the 50 samples from its answer distribution are (virtually) the same. In this case, the Sample & Summarize string just looks like a normal answer, for example:

*Question: The naval battle of Coronel in World War I took place in which ocean?*
Sample & Summarize (N=50): The naval battle of Coronel took place in the Pacific Ocean.

In the large majority (±45%) of other cases, the LLM is uncertain about its answer, but usually just split between two or three answer options. In this case, the answer remains relatively brief, for example:

*Question: When used by John Major in a speech of October 1993, which three word term was misinterpreted as a re-assertion of traditional moral values?*
Sample & Summarize (N=50): "Back to Basics" is the most likely term, though it could also be "classless society.".

Then there are the high-variance cases where the LLM answers (or hallucinates) many different options. This happens for about 5% of the questions, and we provide two examples below:

*Question: What is the highest award in the Boys Brigade?*
Sample & Summarize (N=50): The highest award in the Boys Brigade is most likely the Queen's Badge, but it could also be the King's Scout Award or the President's Badge. Other possibilities include the Medal of the Order of St. George, the Diamond Efficiency Medal, the Duke of Edinburgh's Gold Award, the King's Badge, the Jubilee Badge, the Diamond Award, the Chief's Medal, and the BB Company Award.

*Question: Which company announced in early 2002 that it would be transferring production from its factory in Malmesbury, Wiltshire to Malaysia?*
Sample & Summarize (N=50): The company that announced the transfer of production from its factory in Malmesbury, Wiltshire to Malaysia in early 2002 was most likely Smiths Aerospace. It could also be British Aerospace, which later became part of BAE Systems, or Dyson. Other possibilities mentioned include CORGI, Swindon Powertrains, Psion, and GKN.

Sample & Summarize method leads to a naive listing behavior in those high-variance cases. We also see this effect when we increase the answers that Sample & Summarize samples up to $N = 100$ in Table 19. The length of the average summary string grows, driven by this subset of questions, but seems to converge. Likewise, the SelfReflect score does not reward longer and longer lists. Instead, it flattens out since the additional examples carry no more information.

It is an interesting question what other answers would faithfully reflect the uncertainty especially in these high-variance listing cases. One could for example give an answer that roughly outlines the distribution ("It was some larger British technology firm, but I am not sure which.") rather than giving exact options. We do not want to impose this way over another in our paper, we take a technical perspective: Whatever the answer may look like, it is supposed to give the same information to the user (or a judge LLM) that we would get if we sampled the LLM multiple times. This is the criterion that the SelfReflect score measures (by this, it also penalizes if there is a list of unrelated options, because they would lower the likelihood of the actually relevant options, and thus move away from a faithful representation of the distribution, see the examples in the next question). We propose SelfReflect to give this measurement tool to the community in the future search for different ways to communicate uncertainties to the user.

Table 20: SelfReflect score of self-critique approach based on initial summary from Sample & summarize (N=10) $\downarrow$ ($\times 10^{-3}$ for readability). The results in subscript font are relative to *Sample & summarize (N=10)*.

| Model | Sample & summarize $N\!=\!10$ | Self-Critique $N\!=\!10$ |
|---|---|---|
| Qwen2.5 0.5B Instruct (Yang et al., 2024a) | 96 | $97_{+1}$ |
| Qwen2.5 1.5B Instruct (Yang et al., 2024a) | 87 | $87_{-0}$ |
| Qwen2.5 3B Instruct (Yang et al., 2024a) | 91 | $92_{+1}$ |
| Qwen2.5 7B Instruct (Yang et al., 2024a) | 91 | $90_{-1}$ |
| Qwen2.5 14B Instruct (Yang et al., 2024a) | 86 | $85_{-1}$ |
| Qwen2.5 32B Instruct (Yang et al., 2024a) | 91 | $91_{-0}$ |
| Qwen2.5 72B Instruct (Yang et al., 2024a) | 85 | $84_{-1}$ |
| Phi 4 (Abdin et al., 2024) | 85 | $84_{-1}$ |
| Ministral 8B Instruct 2410 (Jiang et al., 2024) | 101 | $100_{-1}$ |
| Llama 3.1 70B Instruct (Meta AI, 2024a) | 87 | $87_{-0}$ |
| Llama 3.3 70B Instruct (Meta AI, 2024b) | 89 | $88_{-1}$ |
| Llama 4 Scout 17B 16e Instruct (Meta AI, 2025) | 88 | $86_{-2}$ |
| Gemma 3 1B Instruct (Gemma Team et al., 2025) | 117 | $117_{-0}$ |
| Gemma 3 4B Instruct (Gemma Team et al., 2025) | 101 | $102_{+1}$ |
| Gemma 3 12B Instruct (Gemma Team et al., 2025) | 102 | $99_{-3}$ |
| Gemma 3 27B Instruct (Gemma Team et al., 2025) | 97 | $95_{-2}$ |

## O    RESULTS WITH SELF-CRITIQUE

One additional approach we tried to improve the performance of summaries was self-critique, where the model first generates a summary based on one of the other strategies and is then given the chance to critique and improve on its summary. In Table 20 we show the results for applying it to the Sample & summarize (N=10) strategy, where we give the model access to 10 further samples in the second step of critiquing its original summary. As we see, the self-critique approach does not consistently improve the SelfReflect score. However, some improvements can be seen for the larger Gemma models, making this a potentially interesting approach to pursue further in future work, e.g. in an iterative fashion.

