# OpenReview forum: "SelfReflect: Can LLMs Communicate Their Internal Answer Distribution?"
_ICLR.cc/2026/Conference — ICLR 2026 Poster_

### Official Review · Reviewer_zDzS · 2025-10-21

**Soundness:** 3
**Presentation:** 4
**Contribution:** 3
**Rating:** 6
**Confidence:** 4

**Summary:**

The authors introduce SelfReflect, a novel metric to measure the faithfulness of LLMs expressing their uncertainty about answers. Specifically, they examine a certain type of expression of uncertainty, which entails an explicit natural language "description" of a distribution. The SelfReflect metric is a method to measure the consistency between such a natural language "distribution summary" and the real output distribution of the LLM. They do this by alternating putting the distribution summary or some real samples of the distribution in the context and seeing how the distribution of a masked word changes in these two scenarios. The logic: if conditioning on the distribution summary and the "real distribution" (represented by a set of samples) leads to similar masked word probabilities, then they contain the same information. **This point initially confused me: there is the judge LLM (which is giving the masked word probabilities) and then there is the LLM being measured (which gives the distribution summary and real distribution). These are not the same LLM.**

The findings are that the metric is generally more effective than the baselines they show, especially when it comes to nuanced judgements (judging distribution summaries that are subtly wrong in small ways). They also validate that it correlates well with human judgements.

Having established the effectiveness of the metric, they then proceed to measure various LLMs via this metric to see if they can generate effective distribution summaries in a single inference call, and conclude that *they cannot*. Neither simple prompting, chain-of-thought reasoning, reasoning models (like DeepSeek-R1), nor finetuning successfully get the LLMs to generate faithful summaries. They state that LLMs are effective at writing good distribution summary strings given samples from the distribution (the "Sample & Summarize" approach), but not at actually "introspecting" and generating the distribution summaries based on their own internal state.

**Strengths:**

I think the problem being presented (existing methods like providing a single number representing the certainty of one answer not being rich enough to understand the LLMs internal state) is interesting and well-motivated.

The metric is validated extensively. I especially appreciate the human study.

The theoretical connection to predictive sufficiency is compelling, the proofs look rigorous.

The authors do seem to have tested the negative result extensively, trying many different approaches to have the models be able to introspect. They establish that "model introspection" is an open and very difficult problem.

Results are comprehensive and reproducible, writing quality in the paper is clear and comprehensive.

**Weaknesses:**

I am not sure that the authors effectively justify why a natural language summary of the answer distribution is better than just teaching the model to generate structured output representing is answer distribution: something like [("Paris", 0.75), ("Marseille", 0.15), ("Toulouse", 0.10)].

The example the authors provide in their diagram (Figure 3) is interesting, but it's not at all clear to me what happens when you extrapolate to N=50. What do the summary strings actually look like? Is it an enumeration of 30 different answers in e.g. high variance cases? Is this a useful thing to look at as an end-user? It would be interesting to incorporate some notion of "utility" for an end-user.

What is the "compression strategy" the LLMs are using to generate these distribution summaries? I'm assuming some answers get tossed. Which ones, and why, and are we losing important distributional information this way? It's not clear to me. Like I understand your metric hypothetically should avoid losing important distributional information. But I still really would like to see what the final distribution summaries look like at this level. Appendix E is somewhat informative but not sufficient: it demonstrates the metric, but not what the actual distribution summaries generated from the LLMs look like.

I think a qualitative analysis of some examples of the summaries generated would be extremely useful.

The paper is extremely dense and long, with the appendices. Some important details are buried in the appendices, it's so long I may have missed other details, not sure.

I think the paper may be overreaching in its claims, specifically regarding the second contribution: examining if LLMs are capable of expressing distributional summaries without access to samples from the distribution, and if they can be trained to do so. "With SelfReflect, we make a resounding negative observation: modern LLMs are, across the board, incapable of revealing what they are uncertain about, neither through reasoning, nor chains-of-thoughts, nor explicit finetuning." -- I think this is slight overclaiming, they are not able to estimate their answer distribution, which is one specific way they can make the user aware of their uncertainty.

**Questions:**

I would want to see an Appendix with some qualitative analysis of some examples from the N=50 case. What do these summaries look like?

I would want to see some discussion of if these summaries are actually helpful for an end-user. There is no rigorous discussion of whether distribution summaries in natural language are actually useful, and what the benefit is compared to other ways to summarize the distribution of answers from an LLM.

Can an analysis of summary length vs N be done?

To be honest it feels like especially the sample&summarize method is effectively a form of implicit clustering. Like what I must infer is that the summary is essentially grouping different samples together and assigning some probabilities based on the frequency of certain samples over others. This is explicitly what you're doing in Appendix G it seems? Why is the connection to clustering not made explicit in the paper?

"The negative result shows LLMs can't introspect" why do you think this might be? Is it a training data issue, architectural limitation, or something about how RLHF works? Do you have hypotheses for what would enable introspection?"

How does your Sample & Summarize approach compare to existing methods like Semantic Entropy (Farquhar et al. 2024), which also clusters sampled answers? What's the advantage of natural language output over numerical entropy?

---

> ### Author Response · Authors · 2025-11-21
>
> We would like to thank you for your review and for your interesting questions. We address them below.
>
> > W1: I am not sure that the authors effectively justify why a natural language summary of the answer distribution is better than just teaching the model to generate structured output representing is answer distribution: something like [("Paris", 0.75), ("Marseille", 0.15), ("Toulouse", 0.10)].
>
> In QA settings, such structured output would certainly be a valid way of communicating model’s uncertainty. In fact, we do not restrict the output format too much: SelfReflect would give such a structured output a good score, because the judge LLM can perform the imputation task well based on this output, probably about as well as with a corresponding “It’s 75% Paris, but it could be Marseille (15%) or Toulouse (10%)” string. Ultimately, SelfReflect measures what information the summary implies. Future works could thus both explore generating strings and structured outputs, provided they are faithful to the model’s predictive distribution, which is what SelfReflect measures in both cases.
>
> We choose to evaluate SelfReflect on these simple QA settings, because they provide a tractable playground for the development of this approach: they simplify establishing the desiderata for the metric, and allow us to manually craft various scenarios we evaluated in Section 4.1.
> However, an advantage of expressing the uncertainty in the form of a natural-language statement is that, as LLMs become more powerful, this approach should generalize to more complex tasks, where the possible answers are not discrete mutually-exclusive choices. Thanks to its rigorous information-theoretic foundations, SelfReflect should continue to serve its purpose in those more complex cases as well.
>
> > W2: The example the authors provide in their diagram (Figure 3) is interesting, but it's not at all clear to me what happens when you extrapolate to N=50. What do the summary strings actually look like? Is it an enumeration of 30 different answers in e.g. high variance cases? Is this a useful thing to look at as an end-user? It would be interesting to incorporate some notion of "utility" for an end-user.
>
> Thank you for the suggestion! We have now run Sample & Summarize to summarize N=50 answers (and beyond, after which behavior roughly stays the same, see the results below in response to your first question). We provide some examples below that Qwen 2.5 72B Instruct generated in response to questions from the TriviaQA dataset.
>
> In 50% of the questions, the LLM is certain about its answer, thus the 50 samples from its answer distribution are (virtually) the same. In this case, the Sample & Summarize string just looks like a normal answer, for example:
>
> _Question_: The naval battle of Coronel in World War I took place in which ocean?
>
> _Sample & Summarize (N=50)_: The naval battle of Coronel took place in the Pacific Ocean.
>
> In the large majority (±45%) of other cases, the LLM is uncertain about its answer, but usually just split between two or three answer options. In this case, the answer remains relatively brief, for example:
>
> _Question_: When used by John Major in a speech of October 1993, which three word term was misinterpreted as a re-assertion of traditional moral values ?
>
> _Sample & Summarize (N=50)_: \"Back to Basics\" is the most likely term, though it could also be \"classless society.\".
>
> Then there are the high-variance cases where the LLM answers (/hallucinates) many different options. This happens for about 5% of the questions, and we provide two examples below:
>
> _Question_: What is the highest award in the Boys Brigade?
>
> _Sample & Summarize (N=50)_: The highest award in the Boys Brigade is most likely the Queen's Badge, but it could also be the King's Scout Award or the President's Badge. Other possibilities include the Medal of the Order of St. George, the Diamond Efficiency Medal, the Duke of Edinburgh's Gold Award, the King's Badge, the Jubilee Badge, the Diamond Award, the Chief's Medal, and the BB Company Award.
>
> _Question_: Which company announced in early 2002 that it would be transferring production from its factory in Malmesbury, Wiltshire to Malaysia?
>
> _Sample & Summarize (N=50)_: The company that announced the transfer of production from its factory in Malmesbury, Wiltshire to Malaysia in early 2002 was most likely Smiths Aerospace. It could also be British Aerospace, which later became part of BAE Systems, or Dyson. Other possibilities mentioned include CORGI, Swindon Powertrains, Psion, and GKN.

---

> ### Author Response · Authors · 2025-11-21
>
> So indeed, the Sample & Summarize method leads to a naive listing behavior in those high-variance cases. It is an interesting question what other answers would faithfully reflect the uncertainty in these cases. One could for example give an answer that roughly outlines the distribution (“It was some larger British technology firm, but I am not sure which.”) rather than giving exact options. We do not want to impose this way over another in our paper, we take a technical perspective: Whatever the answer may look like, it is supposed to give the same information to the user (or a judge LLM) that we would get if we sampled the LLM multiple times. This is the criterion that the SelfReflect score measures (by this, it also penalizes if there is a list of unrelated options, because they would lower the likelihood of the actually relevant options, and thus move away from a faithful representation of the distribution, see the examples in the next question). We propose SelfReflect to give this measurement tool to the community in the future search for different ways to communicate uncertainties to the user.
>
> For some comments on the user-study aspect, please also refer to our response to your question on user preferences below.
>
> > W3: What is the "compression strategy" the LLMs are using to generate these distribution summaries? [...] A qualitative analysis of some examples of the summaries generated would be extremely useful.
>
> Here we provide two verbose example questions from TriviaQA, ten i.i.d. sampled responses, and summaries generated using four methods (greedy, basic, sample-and-summarize, CoT). Samples and summaries were generated using Qwen2.5-7B-Instruct. We also report SelfReflect scores in square brackets (lower is better) to show how leaving out information is punished. We will include these examples (plus examples from SimpleQA and Natural Questions) in Appendix D.
>
> **Example 1**
>
> _Question_: Calamity Jane was buried next to which well known western personality?
> Sampled answers:
>
> * Calamity Jane was buried next to Wyatt Earp.
> * Calamity Jane is buried next to Wyatt Earp.
> * Calamity Jane was buried next to Buffalo Bill Cody.
> * Calamity Jane is buried next to Wild Bill Hickok.
> * Calamity Jane was buried next to Wild Bill Hickok.
> * Calamity Jane was buried next to Wild Bill Hickok.
> * Calamity Jane was buried next to Wild Bill Hickok.
> * Calamity Jane was buried next to Buffalo Bill Cody.
> * Calamity Jane is buried next to Wyatt Earp.
> * Calamity Jane is buried next to Specialists Pete Coons at Mount Moriah Cemetery in Deadwood, South Dakota.
>
> _Sample-and-summarize:_ Calamity Jane was buried next to Wild Bill Hickok. Could also be buried next to Wyatt Earp or Buffalo Bill Cody. [SR=96.4]
>
> _Basic:_ Calamity Jane was buried next to Wyatt Earp, who is one of the most well-known personalities from the American Old West. [SR=115.3]
>
> _Greedy:_ Calamity Jane was buried next to Wyatt Earp. [SR=118.2]
>
> _Chain-of-thought:_ Calamity Jane is most likely buried in Deadwood, South Dakota, and could be buried next to a notable figure from that area, such as Seth Bullock. Another possible interpretation is that she could be buried next to Buffalo Bill Cody in Wyoming, though there is no definitive evidence to support this. [SR=116.9]
>
> Here, the sample-and-summarize method does a decent job of capturing the full distribution: In the samples, the most frequent response is Wild Bill Hickok (which happens to be the correct answer), but there is high probability mass on both Wyatt Earp (the greedy answer) and Buffalo Bill Cody. The other methods over-estimate the confidence of the greedy answer (basic and greedy), or introduce out-of-distribution answers (Chain of Thought), and hence obtain worse SelfReflect scores.
>
> **Example 2**
>
> _Question:_ The 1995 film 'Clueless', starring Alicia Silverstone, is based on which Jane Austen work?
> Sampled answers:
>
> * 'Clueless' is based on Jane Austen's novel 'Emma'.
> * 'Clueless' is based on Jane Austen's novel 'Pride and Prejudice'.
> * 'Clueless' is based on Jane Austen's novel 'Emma'.
> * 'Clueless' is based on Jane Austen's 'Emma'.
> * 'Clueless' is based on Jane Austen's 'Emma'.
> * 'Clueless' is based on Jane Austen's novel 'Pride and Prejudice'.
> * 'Clueless' is based on Jane Austen's novel 'Emma'.
> * 'Clueless' is based on Jane Austen's novel 'Emma'.
> * 'Clueless' is based on Jane Austen's novel 'Emma'.
> * 'Clueless' is based on Jane Austen's 'Emma'.

---

> ### Author Response · Authors · 2025-11-21
>
> _Sample-and-summarize:_ 'Clueless' is based on Jane Austen's novel 'Emma'. Could also be based on 'Pride and Prejudice'. [SR=84.9]
>
> _Basic:_ The 1995 film 'Clueless', starring Alicia Silverstone, is based on Jane Austen's novel 'Emma'. [SR=90.2]
>
> _Greedy:_ 'Clueless' is based on Jane Austen's novel 'Emma'. [SR=91.6]
>
> _Chain of Thought:_ The most likely answer is that 'Clueless' is based on Jane Austen's novel 'Emma'. While it is possible that the film is inspired by multiple Austen works, the strong parallels to 'Emma' make it the most probable source. [SR=88.7]
>
> Here, the sample-and-summarize method identifies both the most likely response in the samples, and also the other response in the samples. The chain-of-thought response does indicate some uncertainty, but is less precise, leading to it having a slightly worse score than sample-and-summarize. The basic and greedy summaries only identify the most likely response, leading to a worse score.
>
> **Example 3**
>
> If you are interested, there is another example in Appendix E, where we break down the SelfReflect score word-by-word for another question and set of summaries.
>
> > W5: The paper is extremely dense and long, with the appendices. Some important details are buried in the appendices, it's so long I may have missed other details, not sure.
>
> Thank you for the feedback! With the additional camera-ready page, we are open to bring some explanations into the main paper. We have already added some explanations (highlighted in blue). Let us know if you have parts of the paper you would find especially beneficial.
>
> > W6: I think the paper may be overreaching in its claims, specifically regarding the second contribution: examining if LLMs are capable of expressing distributional summaries without access to samples from the distribution, and if they can be trained to do so. "With SelfReflect, we make a resounding negative observation: modern LLMs are, across the board, incapable of revealing what they are uncertain about, neither through reasoning, nor chains-of-thoughts, nor explicit finetuning." -- I think this is slight overclaiming, they are not able to estimate their answer distribution, which is one specific way they can make the user aware of their uncertainty.
>
> We did indeed not intend to claim that LLMs cannot show any form of uncertainty. In our introduction, we mention that LLMs are capable to output numbers, but that we want a description in words, including knowing not only if the LLM is uncertain, but which particular aspects it is uncertain about. We have updated the above phrase to “Neither explicit prompting, nor reasoning, nor SFT and DPO fine-tuning enable an LLM to faithfully summarize its internal beliefs. Its output may have a summary-style format, but it mentions arbitrary possibilities, not those that the LLM actually believes in.”. We hope that this makes our technical observation clearer (and in the experiment section, where we have more space for details, we relate this statement to the observed benchmark scores).
>
> Note that we substantiate our claims under an exact theoretical framework. The claims that we make about the capabilities of LLMs in being able to quantify their uncertainties use the SelfReflect metric, whose correctness and efficacy follows from Proposition 3.2, which in turn is a necessary consequence of the axiomatization in Figure 2, Definition 3.1, and Appendix A.1, where we also discuss the assumptions and their interpretation. These axiomatize what “capability of uncertainty quantification” and “ideal summarization” we measure. We believe that this axiomatization is fair and captures the desiderata of our problem at hand. Thus, we tend to say that our findings are indeed fair.

---

> ### Author Response · Authors · 2025-11-21
>
> > Q1: Can an analysis of summary length vs N be done?
>
> That’s an interesting question. We provide the length of Sample & Summarize with N=10 and N=20 in Tables 4 and 5, and now extended it up to N=100 for Qwen 2.5 72B on TriviaQA, see the results below. We observe that the majority of summaries do not change in lengths, because there are only one or two options the LLM believes in, so independent of how often we sample its distribution, we only get these options in the summary. However, there are some cases in which the LLM is very uncertain, and here the Sample & Summarize method starts listing options. As can be seen below, this increases summary length, but flattens off. Similarly, the score that SelfReflect assigns to these summaries quickly flattens off.
>
> Intuitively this is because some listing gives an idea of what is possible, but more and more examples do not convey any more information about the actual distribution, hence the SelfReflect score (correctly) does not give it better values. We believe it is an interesting venue for future research to explore how to summarize what answers are roughly about, without explicitly listing them like Sample & Summarize does. We add this as a motivating analysis for future works in the revised version (since our paper is mostly about SelfReflect, not about developing a solution to it). We added this to Appendix O of the revised paper.
>
> | Answers used in Sample & Summarize | 10    | 20     | 30     | 40     | 50     | 60     | 70     | 80     | 90     | 100    |
> |-------------------------|--------|--------|--------|--------|--------|--------|--------|--------|--------|--------|
> | SelfReflect score ($×10^{-3}$) ↓ | 84     | 83     | 83     | 82     | 82     | 82     | 82     | 82     | 82     | 82     |
> | Length (characters)     | 93.33 | 104.37 | 113.22 | 120.46 | 123.97 | 127.27 | 133.82 | 135.28 | 136.42 | 137.58 |
>
> > Q2: I would want to see an Appendix with some qualitative analysis of some examples from the N=50 case. What do these summaries look like?
>
> We provide examples above in response to the weakness.
>
> > Q3: I would want to see some discussion of if these summaries are actually helpful for an end-user. There is no rigorous discussion of whether distribution summaries in natural language are actually useful, and what the benefit is compared to other ways to summarize the distribution of answers from an LLM.
>
> We know that measures of uncertainty can be useful to end-users for establishing trust (e.g., Zhang et al 2020). However, existing measures of uncertainty in the form of numbers have empirically been shown to perform suboptimally at this task (see Corvelo Benz and Rodriguez, 2024, and references therein). One argument is that this is because looking at the probability of the most likely answer loses information about the full distribution (Yona et al, 2022). Another is the fact that humans are, on average, not good at interpreting numeric expressions of probability (e.g., Sanborn and Chater, 2016; Tversky and Kahneman, 1983). Natural language summaries of distributions can address both of these issues.
>
> While a full user study on the uncertainties for human-AI collaborative pipelines is beyond the scope of this work, we provide a user study on our SelfReflect score, where raters have a high inter-rater correlation, indicating that users have a consistent understanding of how well a summary captures a distribution.
>
> > Q4: To be honest it feels like especially the sample&summarize method is effectively a form of implicit clustering. Like what I must infer is that the summary is essentially grouping different samples together and assigning some probabilities based on the frequency of certain samples over others. This is explicitly what you're doing in Appendix G it seems? Why is the connection to clustering not made explicit in the paper?
>
> One can view semantic clustering as one part of the task an LLM needs to do while generating a summary string, see also the response to your question about semantic entropy below.
>
> However, in this paper we do not dive too much into the clustering direction, because clustering is often understood as “Given K clusters, find representatives and assignments, usually based on embeddings”. Our perspective on the problem does not use this setup and comes from a more general information-theory perspective: What information (in the form of a summary string) does one need to communicate such that it triggers the same downstream understanding in an LLM judge / user that another information (in the form of a distribution of strings) would trigger? This is the original motivation of our work, and the theoretical basis for section 3. We compare this to related work in summarization, because it is very close to both the higher-level idea of compression and also to some specific, mathematical proposals in that field (measuring mutual information via a downstream task).

---

> ### Author Response · Authors · 2025-11-21
>
> We hope that this gives the best-understandable way to think about our contribution, but are open to include a discussion on the clustering perspective if you feel it would enrich the paper.
>
> > Q5: "The negative result shows LLMs can't introspect" why do you think this might be? Is it a training data issue, architectural limitation, or something about how RLHF works? Do you have hypotheses for what would enable introspection?"
>
> That’s a good question. We believe it is a mixture of training and architecture. In training, the LLM has probably sometimes seen the format of giving multiple outputs (hence it acts according to that format when prompted to output multiple outputs), but apparently never linked this type of list generation to the internal distribution it has, thus giving an answer that has the right format but not the right content. This might be because content is harder to get right than format. We have qualitatively observed that list-generation in an auto-regressive manner might derail the distribution: The LLM is not talking about the distribution about the initial question, but conditioned on the options it already listed. This is similar to an observation made by Zheng et al., 2025. They explicitly build a mechanism that blacks out the attention to previous generated options. They claim that this is a way to train this into the LLM. Hence, it might be a combination of training data and the training loss / design. We hope that the SelfReflect benchmark allows future papers to explore these hypotheses quantitatively.
>
> > Q6: How does your Sample & Summarize approach compare to existing methods like Semantic Entropy (Farquhar et al. 2024), which also clusters sampled answers? What's the advantage of natural language output over numerical entropy?
>
> We see sample & summarize as similar in spirit to semantic entropy (SE), in that both try to convey how many answer options there are. The bigger difference is the output format: SE produces a number whereas sample & summarize produces a string. While SE can thus be evaluated with classical scores like AUROC, we require the SelfReflect score to pivot into the evaluation of strings that summarize uncertainties.
>
> About the advantage of strings over numerical uncertainty outputs: We see strings as the next step in uncertainty quantification for LLMs. Providing numerical uncertainties (and making them faithful) was the first step, mirroring how uncertainties have been communicated in computer vision and classification. But we believe LLMs can go beyond this: A number alone reveals to some extent how certain an LLM is, but many users implicitly use a threshold at something like 50% to decide whether to trust the LLM. With a full text that tells not only about which parts of the answer the LLM is unsure, but also which other options it deems likely, the user gets a much more detailed and honest insight into the knowledge state of the LLM. Outside user-interaction, such strings can also be the starting point for the LLM to refine its answer. Once it has verbalized its uncertainty, it can launch lookups, tool uses, or follow-up questions to the user to better refine its knowledge (whereas a number would only enable it to decide not to answer, or maybe to allocate more reasoning budget, but without direction). For a reasoning model, we see this as a missing building block; currently even frontier reasoning models get stuck in one reasoning path. Overall, we thus believe that giving an honest description of uncertainties as a string is the required stepping stone to enabling new capabilities in both user interaction and reasoning. We see it as our objective in the uncertainty quantification community to develop this building block.
>
> **We would like to thank you again for your time committed to reviewing.** Please let us know if you have further questions.
>
> ---
> Zhang et al 2020: Effect of Confidence and Explanation on Accuracy and Trust Calibration in AI-Assisted Decision Making, https://arxiv.org/pdf/2001.02114
>
> Corvelo Benz and Rodriguez, 2024: Human-Aligned Calibration for AI-Assisted Decision Making, https://arxiv.org/pdf/2306.00074
>
> Yona et al, 2022: Useful Confidence Measures: Beyond the Max Score, https://arxiv.org/pdf/2210.14070
>
> Sanborn and Chater, 2016: Bayesian Brains without Probabilities, https://www.researchgate.net/publication/309467324_Bayesian_Brains_without_Probabilities
>
> Tversky and Kahneman, 1983: Extensional versus intuitive reasoning: The conjunction fallacy in probability judgment, https://psycnet.apa.org/record/1984-03110-001
>
> Zheng et al., 2025: Parallel-R1: Towards Parallel Thinking via Reinforcement Learning, https://arxiv.org/abs/2509.07980

---

> > ### Comment · Reviewer_zDzS · 2025-11-23
> >
> > I thank the authors for their extremely detailed responses.
> >
> > The naive listing behaviour is indeed what I would expect, thank you for confirming this is what occurs.
> >
> > >We did indeed not intend to claim that LLMs cannot show any form of uncertainty.
> >
> > The rewritten phrasing satisfies me, thank you.
> >
> > >We know that measures of uncertainty can be useful to end-users for establishing trust (e.g., Zhang et al 2020). However, existing measures of uncertainty in the form of numbers have empirically been shown to perform suboptimally at this task (see Corvelo Benz and Rodriguez, 2024, and references therein). One argument is that this is because looking at the probability of the most likely answer loses information about the full distribution (Yona et al, 2022). Another is the fact that humans are, on average, not good at interpreting numeric expressions of probability (e.g., Sanborn and Chater, 2016; Tversky and Kahneman, 1983). Natural language summaries of distributions can address both of these issues.
> >
> > Makes sense, thank you! Not sure if I missed this, but highly recommend incorporating this explanation (specficially about humans not being good at interpreting numeric expressions of probability) directly in the paper.
> >
> > >We added this to Appendix O of the revised paper.
> >
> > Fantastic! I think this analysis is important -- it's hard to envision what occurs at high Ns as the reader, so both this and the specific examples you provided are very helpful for this.
> >
> > >We hope that this gives the best-understandable way to think about our contribution, but are open to include a discussion on the clustering perspective if you feel it would enrich the paper.
> >
> > This makes sense -- I'm using clustering in a more general sense, and I do agree that this is effectively a *component* of what the model must do to generate a summary string, so I would be curious essentially to see what happens if you simply do the clustering in a more explicit fashion rather than leaving it "implied" for the model to do, as currently seems to be the case.
> >
> > Thanks to the authors' excellent responses, I am raising my score a notch.

---

### Official Review · Reviewer_2owx · 2025-10-26

**Soundness:** 3
**Presentation:** 2
**Contribution:** 2
**Rating:** 4
**Confidence:** 2

**Summary:**

The authors introduce SelfReflect, an information-theoretic metric that measures how well a summary string matches an LLM’s true internal answer distribution. The SelfReflect metric measures how faithfully such a summary represents the model’s true internal answer distribution using masked-token prediction and Wasserstein distance. Extensive experiments show that current LLMs cannot genuinely articulate their internal uncertainties.

**Strengths:**

1. The experimental results in Section 4 including Distinguishing Good, Bad, and Almost-Good Summaries, Multiple-choice QA and Alignment with Human Judgments are solid and sufficient.

2. The explanation of why CoT fail on reflecting the LLM’s internal confidence is very interesting.

**Weaknesses:**

1. To be honest, I am not quite familiar with this area but would like to ask a very general question. Why do you think the topic, whether a LLM can accurately express its own confidence through natural language, is important?

2. Based on 1, do we even have a promising metric to reflect LLM’s internal probability distribution yet? If not, the topic of investigating whether a LLM can accurately express its internal probability distribution may not be reliable.

3. For Eq2, I am quite confused why the left hand side is equivalent to the right hand side? Why is the conditional probability measured by mutual information?

4. Could you please explain why we should use 1-Wasserstein distance to measure the difference between distributions instead of JS/KL divergence?

**Questions:**

Please check the Weaknesses. I'm not quite familiar with the background knowledge of this paper, so I'm willing to modify my rating based on the comments from other reviewers and ACs.

---

> ### Author Response · Authors · 2025-11-21
>
> We would like to thank you for your review on our paper in which we explore a new frontier for uncertainty communication in LLMs. We respond to your questions below.
>
> > W1: To be honest, I am not quite familiar with this area but would like to ask a very general question. Why do you think the topic, whether a LLM can accurately express its own confidence through natural language, is important?
>
> Firstly, we believe that accurate uncertainty quantification is important for LLMs. If a model appears confident when it is actually uncertain, this can lead to false reliance on uncertain information, and distrust in LLMs’ outputs.
>
> Second, we believe natural language expression of uncertainty is a richer, and more interpretable, way of conveying distributional uncertainty than existing UQ methods for LLMs. Most of these existing methods output a confidence in a single answer (e.g., verbalized confidence approaches), or provide a single number that conveys the amount of uncertainty in the answer distribution (e.g., semantic entropy). This is a natural first step in UQ for LLMs, and mirrors how uncertainties have been communicated in computer vision and classification.
>
> We believe LLMs can go beyond this. Unless the model has no uncertainty, or it is a simple yes/no question, any single number is necessarily a lossy expression of the model’s distribution over responses. For example, providing simply the probability that a single answer is correct doesn’t tell us anything about how many other options the LLM considers plausible, much less why it is uncertain about the answer. This information can be expressed in natural language, however. When given such a natural-language expression of uncertainty, the user (or the LLM) can look to external tools to refine the distribution (for example, by looking up additional information that could rule in or out possibilities).
>
> > W2: Based on W1, do we even have a promising metric to reflect LLM’s internal probability distribution yet? If not, the topic of investigating whether a LLM can accurately express its internal probability distribution may not be reliable.
>
> This is an interesting question. We define the LLM’s internal probability distribution as the generative distribution over answers $Q(a|c)$ that the LLM generates given some context, if prompted many times. Existing work (e.g., Kadavath et al 2022; Zhu et al 2024; Nakkiran et al 2025) suggests that, in many cases, this sampling distribution approximates the true data distribution $P(a|c)$. Further, $Q(a|c)$ is, by definition, a distribution that the LLM has information about and should be able to summarize.
> We note that, while we focus on approximating $Q(a|c)$ in the paper, the SelfReflect metric can be used to evaluate distance to *any* probability distribution that we have samples from. On a mathematical level, the SelfReflect score is a distance between the information of an arbitrary string and an arbitrary distribution of strings. If future authors wish to evaluate distance to some alternative distribution (e.g., the true data distribution, if it is available), the SelfReflect score is a good choice.
>
> > W3: For Eq2, I am quite confused why the left hand side is equivalent to the right hand side? Why is the conditional probability measured by mutual information?
>
> Thank you for your question. Equation 2 of the paper refers to Proposition 3.1, which is a theorem whose complete formulation and proof are provided in Theorem A.1 of Appendix A.2. We want to provide an intuition below.
>
> The way to understand Proposition 3.1 is as follows.  Under the given graphical model provided in Figure 2 [same as Figure 5 in Appendix A], consider the following two statements:
>
> Statement 1: The information content about a new sample $B$ in given samples $A^{(1:N)}$ is equal to the information content about the same new sample $B$ in the summary $S$ of these samples.
>
> Statement 2: The conditional probability of a new sample $B$ given $A^{(1:N)}$ is equal to the conditional probability of the same sample $B$ given the summary $S$ of these samples.

---

> ### Author Response · Authors · 2025-11-21
>
> What our theorem state is that:
>
> i. [the if part =>] If statement 1 turns out to be true, then statement 2 must be true, AND
>
> ii. [the only if part <=] if statement 2 turns out to be true, then statement 1 must be true.
>
> The intuition is that under our graphical model, the ability of a summary $S$ to hold the same amount of information about a new sample $B$ as the samples $A^{(1:N)}$ turns out to be equivalent to the conditional probability of the new sample $B$ given either the samples $A^{(1:N)}$ or their summary $S$ being equal.
> Appendix A.2 provides a proof of why this holds for our graphical model.
>
> So, we are not saying that “conditional probability is being measured by mutual information”; in fact, in general, such equality doesn’t hold. However, in our graphical model of Figure 2, if one of the equalities about conditional probability or mutual information holds, then the other must also hold.
> We hope this clarifies the confusion. Please let us know if you need further explanation!
>
> > W4: Could you please explain why we should use 1-Wasserstein distance to measure the difference between distributions instead of JS/KL divergence?
>
> Indeed we find that SelfReflect is robust to the choice of the exact distance, as long as it is over the whole vocabulary logit vector. For example, replacing Wasserstein distance with KL distance leads to virtually identical results. We choose the Wasserstein over the KL distance because it generalizes gracefully to settings in which we only have the top-1 word, like in blackbox judge models, in which is reduces to an accuracy (we hinted at this in footnote 2, and have now expanded the explanation in the revised version, highlighted in blue).
>
> This graceful generalization is related to the capability of (1-)Wasserstein distance to provide a finite and meaningful measure of divergence in two distributions even when the distributions may have zero or little overlap in their support. In such cases, KL-divergence may explode and need to be capped and JS-divergence may return almost flat metric values which carry no meaningful signal to indicate how much the involved distributions differ.
>
> When we use even simpler distances, like comparing only the log likelihood of the true masked word instead of the whole vector, we observe drops in performance (this is the PR-logl baseline in Tables 1-3).
>
> **We would like to thank you again for your time.** Let us know if you have further questions.
>
> ---
> Kadavath et al., 2022: Language Models (Mostly) Know What They Know, https://arxiv.org/abs/2207.05221
>
> Zhu et al., 2024: Zhu et al 2024: On the Calibration of Large Language Models and Alignment, https://aclanthology.org/2023.findings-emnlp.654.pdf

---

### Official Review · Reviewer_2DwF · 2025-10-28

**Soundness:** 3
**Presentation:** 3
**Contribution:** 2
**Rating:** 4
**Confidence:** 4

**Summary:**

The paper introduces a new setting of uncertainty estimation where the model is required to provide a comprehensive summary of all of its possible answers.
In order to measure the quality of the output with respect to the summary, a SelfReflect metric is introduced, which uses sufficient statistics arguments to measure the summarisation quality of the provided answer given a variety of answers sampled from a model.
The authors further investigate the quality of the resulting metric on syntetic data.
The metric is then used to determine the extent to which the model's 'internal knowledge' is expressed in its outputs for multiple models and decoding strategies.
The authors anticipate that their findings would improve the ability to communicate the model's uncertainty to the end user as well as help several other uncertainty related tasks in NLG.

**Strengths:**

1. The paper is clearly written, experiments are reasonably documented in the appendix.
2. Theoretical Justficiation of SelfReflect metric appears to be sound, the proofs with sufficient statistics provided in the Appendix A are convincing.
3. Visible good effort to connect the theoretical propositions to experimental evaluation. Authors provide ablations of the SelfReflect metric as well as consider a broad range of LM decoding paradigms, including reasoning.
4. Human feedback studies improve the 'value' of the paper.

**Weaknesses:**

1. Minor (little to no impact on my score):
    1. Antropomorphising the langauge models: while subjective and stylistic, I view it as a minor negative aspect. I.e. line 077: "its internal beliefs", line 82: "making LLMs aware of their internal uncertainties", Line 478: "make LLMs honestly describe", etc.
    2. Line 15: "all options it deems possible" - for a language model all options (i.e. every combinatoric token sequence) are technically "possible" unless -inf is allowed in the logits somehow. A strange way to put it unless some thresholding is in place.
    3. Line 483: "Extracting all output possibilities is also a core necessity for conformal approaches", not sure if this is factually accurate considering that what is meant by all possibilities here is extracting them as a single output string.
2. Major:
    1. Emprical benefits are unclear. The Tab. 1 presents comparison between several metrics as well as ablations of the SelfReflect. It can be seen that in Good vs Bad summaries the LM Judge is only second to the unablated SelfReflect, and even then 0.01% away from not achieving 95% significance. Tab.1 could be presented as a heatmap as well, depicting the performance for all combinations of summaries incorporating *all pairs* for a more wholesome picture.
    2. Some experiments may have unanticipated caveats:
        1. The data for the Tab. 1 was reportedly created by using Qwen2.5 7B for answer generation and Gemini 2.0 Flash for summary creation. Given the tiny margins for the main Good vs Bad summaries experiment, one may want to ablate this by answers and summaries coming from different models / generated by other means. The almost-good summary generation appears to be even less robust, since it is created starting from an already model generated summary. The other pairings also seem somewhat arbitrary.
        2. Fig 4: The certainty / uncertainty of the answers / summaries is ill defined. It therefore hard to pass judgement based on that.
        3. Tables 4 and 5: Reasoning model output lengths appears to be strange: it is hardly twice the length of normal outputs. RLVR tuned models from Tab.5 are not trained for summarization and fact retrieval, but for facing math and logic problems.
    3. Although I do not have an in-depth familiarity with state of the art summarization literature, the paper fails to compare SelfReflect to summarization specific metrics. For example something on the lines of [1] (Which is also a PMI of sorts).
    4. The second part of the contribution, the analysis on the expression of internal knowledge via different decoding schemes lacks clear placement: it is unclear how exactly we can benefit from this knowledge. The guidance in the Outlook is inspecific.

### References:
1. Jung, J. et al. Information-Theoretic Distillation for Reference-less Summarization. Preprint at https://doi.org/10.48550/arXiv.2403.13780 (2024).

**Questions:**

1. Would masking on sentence level be a valid approximation of SelfReflect?
2. Could there be an alternative to using Wasserstein distance?
3. As this work largely addresses on fact recollection by the model, what impact could introducing RAG have on the performances presented?

---

> ### Author Response · Authors · 2025-11-21
>
> We would like to thank the reviewer for the detailed review and the interesting questions. We respond to the questions below.
>
> > W1: Antropomorphising language (interal beliefs, making LLMs aware of their internal uncertainties, all options it deems possible)
>
> Thank you for the remarks. We in no way intend to antropomorphise the LLMs we test; we see them as mathematical object, in particular as distributions over strings. We have chosen these terms due to brevity and to make the sometimes rather technical content easier to understand. To prevent misunderstandings, we have added an explanatory footnote in the revised version (changes are highlighted in blue).
>
> > W2.1: Empirical benefits are unclear. The Tab. 1 presents comparison between several metrics as well as ablations of the SelfReflect. It can be seen that in Good vs Bad summaries the LM Judge is only second to the unablated SelfReflect, and even then 0.01% away from not achieving 95% significance
>
> It is correct that LM Judge is able to tell apart good from bad summaries, like most other baselines and SelfReflect. However, note that the good vs bad case is only a very rough first sanity check. Bad summaries change major content in the summary, mentioning wildly different answers than the LLM actually gives. The more interesting summary pairs are the finer-grained ones like good vs almost-good, percentage vs or-concatenated, etc. We want a benchmark metric to give a better score to a summary that mentions all possibilities rather than just the majority one, and similarly, when presented with two (already very good) summaries that either give all possibilities or all possibilities plus their (correct) percentages, we want the benchmark metric to reward the latter. To make a longer-shot comparison, the generative vision field is shifting away from Frechet distance based on Inception embeddings (FID) towards using Frechet distance based on DINOv2 embeddings (FD-DINOv2), because the former could not tell apart subtle quality differences of strong models anymore. Benchmark metrics require this fine-grained precision to reliably measure progress in the field, and we found that only SelfReflect is able to detect those differences in fine-grained pairs like good vs almost-good, whereas all baseline metrics collapse to close to random performance. This effect is significant and robust across all tested datasets and models.
>
> > W2.1 (cont): Tab.1 could be presented as a heatmap as well, depicting the performance for all combinations of summaries incorporating all pairs for a more wholesome picture.
>
> We intentionally do not show all combinations of summary pairs for two reasons. First, the summaries are computed on different subsplits of the dataset. Good-vs-bad and good-vs-almost-good are computed on the full dataset. Detailed-vs-truncated is only computed on questions where the answers in the answer distribution are all semantically the same, but have varying levels of details. Verbalized-vs-majority etc are computed only on questions where the answer distribution covers multiple answers but those answers are all of the same level of detail. We do this to control for confounders, which allows us to study these two aspects in isolation. We could thus only show all-combinations within each dataset split, but it would make the results harder to understand because they would become three-dimensional (summary 1 vs summary 2 vs benchmark metric) and sparse (due to the data splits). We find the 2D table easier to interpret, since it covers all important comparisons already. We will however highlight the table in heatmap-like colors in the revised version, thank you for the suggestion!
>
> > W2.2.1: The data for the Tab. 1 was reportedly created by using Qwen2.5 7B for answer generation and Gemini 2.0 Flash for summary creation. Given the tiny margins for the main Good vs Bad summaries experiment, one may want to ablate this by answers and summaries coming from different models / generated by other means. The almost-good summary generation appears to be even less robust, since it is created starting from an already model generated summary. The other pairings also seem somewhat arbitrary.
>
> We have conducted these ablations. In Appendix D, Table 7, we report a cross-table of different LLM Judges and LLMs whose answer distributions are summarized (Llama, Qwen, Phi). In Appendix G, Table 8, we use human-written summaries instead of Gemini 2.0 Flash-generated ones. Our findings are robust across all of these experiments. As explained above, the margins are very solid in all experiments on the finer-grained testcases.

---

> ### Author Response · Authors · 2025-11-21
>
> > W2.2.2: Fig 4: The certainty / uncertainty of the answers / summaries is ill defined. It therefore hard to pass judgement based on that.
>
> We define whether an answer distribution / summary is uncertain by whether it mentions more than one (semantic group of) answers, similar to the Semantic Entropy literature. We give the exact prompt in Appendix L.2. We qualitatively find that Gemini 2.0 Flash is reasonably strong in deciding whether there is only one or different answers mentioned in a set of answers / a summary. We’ve added some examples to Appendix L.2 of the revised paper, to allow you and curious readers to have a look.
>
> > W2.2.3: Tables 4 and 5: Reasoning model output lengths appears to be strange: it is hardly twice the length of normal outputs. RLVR tuned models from Tab.5 are not trained for summarization and fact retrieval, but for facing math and logic problems.
>
> The “output length” reported in the table doesn’t count the contents of the generated thinking-block (in-between the special tokens: start_think...end_think), it counts only the final output, after the </think> token. This is why the length of the summary is roughly in the ballpark of the sample-and-summarize method.
>
> Regarding “for facing math and logic problems”: the notable fact about RLVRed models is that while they are indeed trained to reason on math problems, the learned reasoning behavior has been shown to generalize to problems beyond mathematics, see e.g. the “Business” demo at https://openai.com/index/introducing-o3-and-o4-mini/. For example, below we have prompted https://huggingface.co/Qwen/Qwen3-32B to answer the following question from the SimpleQA dataset (you can try yourself with the “Inference Providers” on the hf website):
>
> Prompt:
> > Please respond to the following question 'Who received the IEEE Frank Rosenblatt Award in 2010?'.
> > Your goal is to summarize all possible answers to this question:
> > * If there are multiple possible answers, the summarized answer should mention the main possible answers. However, you do not have to list possibilities that are too unlikely.
> > * If some possibilities are more likely than others, delineate which possibilities are more more likely by using words like "most likely" and "could also be".
> > * The format of the summarized answer should be the same as a normal answer.
> > * If there is only clear answer to the question, just provide that answer, without hedging across possibilities.
> > Please provide the summarized answer.
>
> Response:
> > start_think
> > Okay, let me try to figure out who received the IEEE Frank Rosenblatt Award in 2010.
> > First, I know that the Frank Rosenblatt Award is given by the IEEE Neural Networks Council and the Cognitive and Neural Systems Technical Committee. It's named after Frank Rosenblatt, who was a pioneer in neural networks. The award usually goes to someone for foundational contributions in cognitive and neural systems.
> > I need to check the 2010 awardees. Let me think. If I recall, some notable recipients of this award include Yann LeCun for his work on convolutional networks, Geoffrey Hinton for backpropagation and deep learning, and others. Wait, but Hinton and LeCun were awarded in different years. Let me try to remember or find the exact year. I think Geoff Hinton received the Rosenblatt Award in 2010. But I'm not 100% sure. Alternatively, maybe someone else contributed significantly around that time. Let me cross-verify.
> > Another person who might have received it could be Yoshua Bengio, but I think Bengio received the award later. Wait, no, Bengio might have been a recipient in a different year. Alternatively, maybe someone else who worked on neural networks in the 90s or 2000s. Wait, I've heard that John Hopfield was a recipient, but that might have been earlier.
> > Wait, another angle: the 2010 Rosenblatt award was given to someone for foundational contributions. Let me think if there's any other name associated with that year. I might be mixing up other awards, like the IEEE Neural Networks Pioneer Award. Wait, no, the question is specifically about the Frank Rosenblatt Award.
> > Geoffrey Hinton is a big name in deep learning. I think he did receive this award. Let me try to confirm in my mind. I recall that the 2010 Rosenblatt Award was given to Geoffrey E. Hinton for his work on deep learning and neural networks. However, I might be conflating this with the 2008 or 2009 award. But if I have to choose based on memory, I think it was Hinton. Alternatively, could it be someone else? Let me think again.
> > Wait, I've also heard that Yann LeCun received the Rosenblatt Award, but maybe in another year. For example, Yann LeCun received the award in 2011. So perhaps 2010 went to someone else.
> > Wait, another possibility: maybe another pioneer from the 90s. For example, Jürgen Schmidhuber? No, I think Schmidhuber received the award a bit later. Hmm.

---

> ### Author Response · Authors · 2025-11-21
>
> > Alternatively, maybe it was Demis Hassabis? No, Hassabis is a founder of DeepMind and worked on AlphaGo, but his contributions in neuroscience and AI came more in the 2010s.
> > Wait, I'm getting a bit stuck. Let me try to outline the possibilities:
> > Geoffrey Hinton - a leading figure in deep learning, but when did he receive the Rosenblatt Award? If my initial thought is wrong, perhaps he received it earlier.
> > Yann LeCun - another leading figure, maybe he got the award in 2011.
> > Maybe someone else.
> > Alternatively, perhaps it was someone like Terry Sejnowski?
> > Wait, Terry Sejnowski is a notable neuroscientist and neural network researcher. Did he receive the Rosenblatt Award in 2010? I'm not certain.
> > Alternatively, the award could have been given to multiple people in that year.
> > Given my uncertainty, I need to consider that my initial thought about Geoffrey Hinton might be incorrect. Let me think again.
> > Wait, maybe the 2010 Rosenblatt Award was given to someone for work in the 90s or early 2000s. Geoff Hinton's key work in backpropagation and deep learning was in the 80s and 90s, but he continued developing these ideas into the 2000s. So if the award is given in 2010, maybe they were recognizing that long-term work.
> > Given that, I'm still leaning towards Hinton, but I'm not 100% sure. Alternatively, could it be someone else? Let me check my memory again.
> > Wait, another possibility: I remember that the 2010 IEEE Frank Rosenblatt Award was given to Geoffrey E. Hinton for his contributions to the development of deep learning networks. However, I'm not completely certain, so there's a chance someone else received it that year. But given Hinton's prominence in the field, this seems like a strong possibility.
> > But if there's another possibility, I should consider that. For example, maybe it was awarded to two people? Let me try to recall.
> > Wait, no, I think the award is usually given to one individual each year.
> > So, putting it all together, the most likely answer is Geoffrey Hinton, but I'm not 100% certain. However, since the user wants the most likely answer and to mention if there are others, I should present that.
> > end_think
>
> > The 2010 IEEE Frank Rosenblatt Award was most likely awarded to Geoffrey E. Hinton for his foundational contributions to deep learning and neural networks. While there is no definitive alternative that is widely recognized as equally probable, the award that year could also be associated with other pioneers in the field, though Hinton's receipt is the most plausible based on his prominent role in advancing the field during that period.
>
> > W2.3: Although I do not have an in-depth familiarity with state of the art summarization literature, the paper fails to compare SelfReflect to summarization specific metrics. For example something on the lines of [1] (Which is also a PMI of sorts).
>
> Thank you for providing the InfoSumm baseline. We agree it is similar in nature (a masked-out task intended to measure the PMI between summary and document), though different in the exact estimator, making it interesting to compare against. We have implemented it, with exact details in the new Appendix F.9 and a general discussion in our related works section on summarization. To test it against SelfReflect, we follow the experimental setup of Section 4.1, measuring how often it assigns good summaries better scores than (different kinds of) bad summaries. We report the findings below and in the revised version of the paper. We find that InfoSumm is able to tell apart Good vs bad, but none of the more challenging pairs of summaries. In fact, it prefers summaries that only mention the most likely answer over those that mention all options, similar to the SR-PMI baseline that uses a similar estimator. Note also that besides InfoSumm, we provide two other summarization-specific baselines: The “Summarization” baseline in tables 1, 2, 3 uses consistency, fluency, relevance, and coherence ratings, which are the standard in summarization literature, and LM Judge is prompted to judge the quality of the summary string in one go.

---

> ### Author Response · Authors · 2025-11-21
>
> | Metric             | Good summaries vs bad summaries | Good vs almost-good | Detailed vs truncated | Verbalized uncertainty vs only majority answer | Verbalized vs or-concatenated | Percentage vs or-concatenated |
> |--------------------|---------------------------------|----------------------|------------------------|-----------------------------------------------|--------------------------------|--------------------------------|
> | Summarization      | 93.33% ± 0.89%    | 39.72% ± 1.87%       | 53.05% ± 6.04%         | 19.90% ± 5.66%                                 | 58.12% ± 7.00%    | 64.92% ± 6.77%    |
> | InfoSumm           | 99.87% ± 0.13%                   | 60.81% ± 1.87%       | 49.24% ± 6.05%         | 15.71% ± 5.16%                                 | 27.75% ± 6.35%                 | 10.99% ± 4.44%       |
> | LM Judge           | 98.33% ± 0.46%                   | 47.32% ± 1.91%       | 59.92% ± 5.93%         | 19.37% ± 5.60%                                 | 34.55% ± 6.74%      | 35.08% ± 6.77%  |
> | Opt. Transport     | 80.16% ± 1.43%   | 60.78% ± 1.87%       | 39.69% ± 5.92%         | 48.69% ± 7.09%                                 | 52.88% ± 7.08%                 | 69.11% ± 6.55%  |
> | Embedding          | 96.50% ± 0.66%                   | 65.49% ± 1.82%       | 65.65% ± 5.75%         | 10.99% ± 4.44%                                 | 43.98% ± 7.04%                 | 36.65% ± 6.83%                 |
> | SR-PMI             | 88.40% ± 1.15%                   | 33.64% ± 1.81%       | 53.44% ± 6.04%         | 25.65% ± 6.19%                                 | 14.14% ± 4.94%                 | 20.42% ± 5.72%                 |
> | SelfReflect        | 98.77% ± 0.40%                   | 93.20% ± 0.96%       | 93.13% ± 3.06%         | 85.34% ± 5.02%                                 | 72.77% ± 6.31%                 | 80.10% ± 5.66%                 |
>
> > W2.4: The second part of the contribution, the analysis on the expression of internal knowledge via different decoding schemes lacks clear placement: it is unclear how exactly we can benefit from this knowledge. The guidance in the Outlook is inspecific.
>
> Thank you for your feedback. There are two important takeaways from this contribution:
> 1. A simple sample-and-summarize method performs reasonably well for summarizing the models distribution from samples and hence can be a faithful (w.r.t. the reference distribution of independent samples) way of communicating models’ uncertainty.
> 2. In general, methods that don’t explicitly generate samples cannot be trusted to accurately capture this uncertainty faithfully. At a higher level, this elucidates a difficulty LLMs have with “reflecting” on their own predictive distribution for the original question while trying to produce a summary in a single-decoding.
>
> We are optimistic that future work will uncover ways of obtaining a good summary in a single decoding (ie without drawing samples), and feel this is an important open direction for the field. SelfReflect provides a way to evaluate the success of such attempts, and the sample-and-summarize approach offers a strong baseline.
>
> > Q1: Would masking on sentence level be a valid approximation of SelfReflect?
>
> We tested this; we call this the SR-PMI baseline, described in lines 256-258 and Appendix F.3. We find that this yields a less fine-grained metric than SelfReflect, probably because it loses too much context and increases the noise-to-signal ratio of what responses are possible. The masked-out task which apparently makes the context clearer to the judge. To reduce the amount of computations, we found it better to skip stopwords, or to reduce the number of masked-out tasks (see Appendix B.1 and the new experiments in Appendix B.2).
>
> > Q2: Could there be an alternative to using Wasserstein distance?
>
> Yes, indeed we find that SelfReflect is robust to the choice of the exact distance, as long as it is over the whole vocabulary logit vector. For example, replacing Wasserstein distance with KL distance leads to virtually identical results. We choose the Wasserstein over the KL distance because it generalizes gracefully to settings in which we only have the top-1 word, like in blackbox judge models, in which is reduces to an accuracy (we hinted at this in footnote 2, but have now expanded the discussion in the revised paper).
> This graceful generalization is related to the capability of (1-)Wasserstein distance to provide a finite and meaningful measure of divergence in two distributions even when the distributions may have zero or little overlap in their support. In such cases, KL-divergence may explode or need to be capped and JS-divergence may return almost flat metric values which carry no meaningful signal to indicate how much the involved distributions differ.  When we use even simpler distances, like comparing only the log likelihood of the true masked word instead of the whole vector, we observe drops in performance (this is the PR-logl baseline in Tables 1-3).

---

> ### Author Response · Authors · 2025-11-21
>
> > Q3: As this work largely addresses on fact recollection by the model, what impact could introducing RAG have on the performances presented?
>
> We do not expect a difference and have now added an experiment. In the submitted paper, we use fact recollection datasets because they give a setup where models are unsure about answers and actually have to summarize different answer options (whereas in RAG many answer distributions are collapsed to only one answer), but the general idea persists even outside of it: No matter what the context is, it might be just a question or include a RAG document, a model’s summary should be able to summarize the model’s output distribution based on the current context. SelfReflect is able to judge the consistency between the answer distribution and the summary string agnostic of how or on which context they were generated.
>
> To test this, we’ve now added HotpotQA, a dataset where the model is given some paragraphs of Wikipedia as a context to answer a question. We show the results below. The trends are the same as in the paper, sample and summarize is the only method that consistently gives better (lower) SelfReflect scores than Greedy. The gap to Greedy is smaller in all methods due to the output distributions that are less diverse, as predicted above. We have added this experiment to the revised version of the paper, Appendix N.
>
> | Model              | Greedy | Basic        | CoT          | Sample & Summarize (N=20) |
> |--------------------|--------|--------------|--------------|---------------------------|
> | Qwen 2.5 1.5B      | 76     | 76 (-0)      | 76 (-0)      | 73 (-3)                   |
> | Qwen 2.5 3B        | 80     | 81 (+1)      | 81 (+1)      | 79 (-1)                   |
> | Qwen 2.5 7B        | 79     | 79 (-0)      | 80 (+1)      | 78 (-1)                   |
> | Qwen 2.5 14B       | 78     | 80 (+2)      | 80 (+2)      | 76 (-2)                   |
> | Qwen 2.5 32B       | 79     | 80 (+1)      | 80 (+1)      | 78 (-1)                   |
> | Qwen 2.5 72B       | 78     | 79 (+1)      | 80 (+2)      | 76 (-2)                   |
> | Microsoft Phi 4    | 79     | 78 (-1)      | 78 (-1)      | 76 (-3)                   |
> | Ministral 8B 2410  | 86     | 86 (-0)      | 85 (-1)      | 84 (-2)                   |
> | Llama 3.1 70B      | 80     | 79 (-1)      | 79 (-1)      | 78 (-2)                   |
>
> **We would like to thank you again for the detailed review.** We hope that the above experiments and clarifications address your concerns. Let us know if you have further questions.

---

> > ### Comment · Reviewer_2DwF · 2025-11-28
> >
> > I thank the authors for answering the bulk of my questions. Some points like addition of a summarization specific baseline to comparison as well as clarifications provided definitely improve my assessment of the paper.
> >
> > I therefore would raise my score to 6- (carefully leaning in the positive direction). The rationale that prevents me from further increase is the following:
> >
> > W2.1: While I agree with the authors rationale for good vs almost good samples, I am not fully convinced in the means of obtaining them. There are several iterations of model generated examples which is inherently complex and may require a whole separate work to analyze its robustness (even though they are well documented in the paper).
> > W2.2.3: Indeed, I agree that reasoning models are generalists even though they are trained for math problems. At the same time, the models specifically trained for summarization might be more capable in producing summaries.

---

> > > ### Author Response · Authors · 2025-12-02
> > >
> > > We would like to thank you for coming back, and for your positive reception of our rebuttal experiments so far. We are happy to offer more experiments below to address your last questions.
> > >
> > > > W2.1: While I agree with the authors rationale for good vs almost good samples, I am not fully convinced in the means of obtaining them. There are several iterations of model generated examples which is inherently complex and may require a whole separate work to analyze its robustness (even though they are well documented in the paper).
> > >
> > > To study the robustness of our experiment setup in Table 1, we have now also added almost-good summaries that are generated in a one-step process. So instead of first deriving a good summary and then changing it into an almost-good one, we now directly generate an almost-good summary by providing the sample distribution. You can find the results below (on Natural Questions and Qwen 2.5 7B in this case).
> > >
> > > They agree with what we report in the paper: Most baseline metrics are able to distinguish a good from an almost-good summary in 72% to 94% of the cases. SelfReflect is more fine-grained and can distinguish, in this dataset, every single good summary from their mid counterpart (this one-step way of generating almost-good summaries seems to make them slightly easier to distinguish than adapting the good summary as we do in the main experiments). Note also that besides this ablation, we also have an ablation where humans write good and almost-good summaries in Appendix G, Table 9, with agreeing results. With these agreeing results under the ablated experimental setups, we are quite certain about the robustness of our findings.
> > >
> > > | Metric              | Good summaries vs mid summaries (one-step generated) |
> > > |---------------------|---------------------------------|
> > > | Summarization       | 93.98% ± 2.70%                  |
> > > | InfoSumm            | 71.91% ± 5.09%                  |
> > > | LM Judge            | 84.28% ± 4.13%                  |
> > > | Opt. Transport      | 75.92% ± 4.85%                  |
> > > | Embedding           | 82.94% ± 4.26%                  |
> > > | SR-PMI              | 84.28% ± 4.13%                  |
> > > | SR-sampling-free    | 89.30% ± 3.50%                  |
> > > | SelfReflect         | 100% ± 0.00%                     |
> > >
> > >
> > >
> > > > W2.2.3: Indeed, I agree that reasoning models are generalists even though they are trained for math problems. At the same time, the models specifically trained for summarization might be more capable in producing summaries.
> > >
> > > We believe we should further clarify the nuance about the promise of the reasoning models!
> > > Sorry for a long explanation, but we feel it’s beneficial for the sake of explanation if we are explicit and comprehensive here.
> > >
> > > Producing a good summary of the model's internal uncertainties basically requires two components: 1) Being able to list out or be in some sense aware of the different possible options in the internal distribution, and 2) summarizing them into a consistent answer.
> > > The motivation for evaluating the reasoning models is the potential promise (which we formed based on our qualitative observations about their behavior of the kind exemplified with Qwen3-32B in our previous response) that, unlike the non-reasoning models, the reasoning-models might have an ability to use their reasoning-block to "brainstorm" different possible answers in a way that coincides with their probability distribution when drawing independent samples (which is the reference for the SelfReflect score). This would imply that the reasoning-models might perform significantly better in terms of producing a single-decoding summary of their predictive distribution, because they perform better at the first component above.
> > > This is evaluated with prompt "Basic" of Table 4, which for reasoning models is denoted as "Reasoning" in Table 5. You can see the exact prompt in the example with Qwen3-32B in our earlier response.
> > > Surprisingly, we found that the SelfReflect scores for the Reasoning method for reasoning-models were worse than for the Basic method for non-reasoning-models. The reasoning block often comes up with different options than the model actually thinks of, failing at component 1.
> > >
> > > The second component, summarizing given options, seems to work consistently across models, including reasoning models. Our results suggest that the performance of Sample&Summarize method for reasoning models are about as good as non-reasoning, e.g. compare Sample&Summarize numbers for Qwen-2.5-32B vs DeepSeek-R1-Distill-Qwen-2.5-32B: for N=10, 91 vs 91, and for N=20, 91 vs 90, respectively. These results suggest that the summarization component alone is not a bottleneck in the process. (Note: for “summarization component”, the options to be summarized are provided in context here, in contrast to the Basic/Reasoning prompt, where the model has to reflect on and summarize the different options in one decoding step).

---

> > > > ### Author Response · Authors · 2025-12-02
> > > >
> > > > As such, evaluating performing the second step with models specifically trained for summarization will not elucidate any new qualitative effects, which is the primary purpose of this paper.
> > > > We suspect that performing the summarization-step for Sample&Summarize with such models might perform marginally better in absolute terms, but comes at the inconvenience and computational overhead of having to do inference using two separate models for each query.
> > > > For the time being, we leave the question of tuning these details to practitioners who want to gain absolute maximum performance from their system.
> > > >
> > > > That said, we have one new extra experimental result that might cast some light on the question you’re asking. By the suggestion of Reviewer j1CR Q7, we evaluated a self-critiquing approach which tries to improve upon the Sample&Summarize-N=10 by allowing the (same) model to improve upon its original summary. We have included the results in Appendix P of the manuscript. In short, for some models (the larger Gemma3 models), this approach improves the SelfReflect scores marginally (by 2-3 points), as expected, in line with our comment above. See our recent response to Reviewer’s j1CR Q7 for details.
> > > >
> > > > We hope that these explanations and the experiment address your remaining comments.

---

### Official Review · Reviewer_j1CR · 2025-11-02

**Soundness:** 3
**Presentation:** 3
**Contribution:** 4
**Rating:** 8
**Confidence:** 4

**Summary:**

**Summary**: This paper investigates the problem of communicating uncertainty of the distribution over answers to a question, advocating for the generation of all possible answers and their likelihood when generating an answer to a question (instead of generating the most likely answer and a confidence score). It then proceeds with (1) proposing and validating a metric–**SelfReflect**– to quantify the faithfulness of a summary to a LLMs internal distribution over answer, (2) empirically investigating whether through various prompting techniques LLMs are able to produce such descriptive summaries of their internal distribution.

**Main contributions**:

- Novel paradigm for uncertainty quantification, requiring an answer that is descriptive of the LLM’s internal uncertainty.
- Proposal of a theoretically grounded metric, SelfReflect, that measures faithfulness between a descriptive answer (dubbed *summary*) and the LLM’s internal distribution over answers.
- Extensive experimentation with 16 popular LLMs across 3 datasets to assess their abilities in generating descriptive summaries of their internal uncertainty either through prompting, finding them to be inapt.

**Strengths:**

- Novel contributions for uncertainty quantification in LLMs, offering a fresh perspective on uncertainty quantification and designing new set of metrics and evaluations;
- Proposal of a new metric that is grounded in information theory concepts, such as mutual information.
- Experimental design allows for the controlled assessment of the capabilities of the proposed metric, comparing it against numerous baselines including LLM-as-a-judge approaches.
- Benchmark 10+ models in 3+ datasets (TriviaQA, NaturalQuestions, SimpleQA, MMLU) and 5 different prompting strategies and 2 fine-tuning approaches to assess model’s abilities in generating descriptive summaries.
- Various appendices supporting various analyses and arguments in the paper – e.g., providing a convergence analysis of SelfReflect metric.
- Interesting findings, including the model’s inability (even after SFT or DPO) to provide descriptive summaries of its own internal uncertainty.
- Well-written and well-organized.

**Weaknesses:**

W1. **Controlled experiments in section 4 concern different models**, raising questions about the generalization of the results (see Questions).
W2. Some sections are a bit confusing or not clearly explained (see Questions for details)

**Questions:**

**Questions**:

1. If I understand correctly, according to _Section 3.1 Summaries as Predictive Sufficient Statistics_, an ideal summary should capture the relative frequency and diversity of the answers. This aligns well with the example in Figure 1 for which the answer distribution concerns confident-sounding answers. However, I’m curious to understand how the definition of an _ideal summary_ would account for cases  where some of the answers have uncertainty in themselves, e.g., “It’s likely to be Paris” or “It’s possible that the answer is Toulouse”?
2. The paper mentions the computational requirements of the SelfReflect metric in lines 203-208 and convergence of the SelfReflect score under various computational budgets in Appendix B (Figures 6-10). This convergence analysis is based on the SelfReflect score itself. However, I was interested in knowing how exactly Self-Reflect compares to other baselines such as LLM-as-a-judge in some controlled experiment similar to study 1 in Section 4.1. Can the authors comment on whether there’s a minimum number of samples to ensure the faithfulness of SelfReflect (i.e., if we have less than N sampling budget, we’re better off using other baseline metric?)
3. Concerning the methodology, I’d like to clarify how is the b sentence selected? For example, in Figure 3, is b selected as the most frequent generation, or does SelfReflect eventually iterate over different sampled sequences $a^{(i)}$?
4. Line 293 mentions “fine-grained quality differences” when discussing the superior behavior of the SelfReflect metric on the Good vs Almost Good setting. I could not find the details of how almost good summaries are generated. How do you ensure that these summaries differ by only fine-grained differences and what kind of differences are these? Was there any human validation performed to ensure this is the case?
5. Line 298 states “matched only twice by its own SR-P(True) ablation”. Should it be “three times” instead of “twice”, since performance values between the SelfReflect and SR-P(True) ablation are also overlapping in the “Verbalized vs or-concatenated” setting?
6. Could the authors motivate the selection of different models across controlled experiments in Section 4?
7. Section 5 includes experiments with numerous prompting and fine-tuning strategies. I was wondering if an iterative self-critiquing approach would improve upon simple prompting or “sample and summarize”?
8. In Section 5.2 the authors suggest that despite fine-tuning and providing examples of descriptive summaries to the models, they seem to be unable to perform the task and that “model memorizes individual summaries rather than learning a general mechanism”. Could this be due to data quality issues (since the data is generated synthetically) and/or hyperparameter tuning?




**Writing style**:

- Line 40: More of a subjective preference but expressions “We believe we can do better than this.” seem to be unnecessary and a bit informal for a conference style paper. I’d suggest replacing it with something more formal.
- Line 351: Please provide a citation for the Krippendorf’s alpha. This can be for a particular implementation or a relevant paper.


** Clarity**:
- The experimental design for study 2 is unclear for me. Here are a few terms and expressions that were ambiguous:
     - line 314: “true ratio of answers”: what is the true ratio of answers computed over? Is it over the whole test set or based on the normalized logits for each question?
     - line 319: “compute the true Wasserstein distance between the distribution described in the summary and that of the test-set answers” → Could you clarify what the test-set answers distribution is in the context of a MMLU and how that is computed in a per question vs whole dataset basis? Why is the reference-metric a reference metric? What’s the intuition behind it?

**Typo**:
- Line 374: “decoding” → “decoding pass”
- Line 448: “as example case” → “As an example”
- Line 378: “Developing” → “developing”
- Table 2 caption: “Mean $\pm 95\%$.” → “Mean $\pm 95\%$ CI.”

---

> ### Author Response · Authors · 2025-11-21
>
> We would like to thank the reviewer for the detailed review and the enthusiasm about our work on this new frontier of uncertainty communication. We respond to the questions below.
>
> > W1. Controlled experiments in section 4 concern different models, raising questions about the generalization of the results (see Questions).
>
> In section 4, we test whether our SelfReflect benchmark score works across multiple models, both when using the models as judges and when judging the distributions of those models. We do this to test the robustness of our method. We did not test every combination of models due to limitations of our compute availability and the feasible amount of human annotation & quality-assurance (sections 4.1 and 4.3). Please see our answers to your questions below.
>
> > W2. Some sections are a bit confusing or not clearly explained (see Questions for details)
>
> Thank you for the feedback, we updated the paper to address your feedback, and uploaded a new revision (changes are in blue). We hope that our changes resolve any inclarity. We go into detail in the questions below.
>
> > Q1: I’m curious to understand how the definition of an ideal summary would account for cases where some of the answers have uncertainty in themselves, e.g., “It’s likely to be Paris” or “It’s possible that the answer is Toulouse”?
>
> This is an interesting question. Our metric bases on predictive sufficiency, i.e., a summary is ideal if, when given to an LLM judge, the LLM judge will predict the same way it would predict as when given the distribution of answers. When the answers in the distribution contain hedging statements themselves, it depends on how the LLM judge interprets them when filling in the masked-out tasks. We have observed that LLMs (judges) implicitly assign distributions to statements like “It’s likely”. For example, when prompting Qwen 2.5 7B with “Most likely A, but could be B or C”, and then asking to choose one of them in a SelfReflect-style masked-out task, it produces 50% A, 25% B, 25% C. So models generate quite reasonable logits for masked-out tokens even when conditioned with context containing “It’s likely…” etc. The ideal summary would need to make the LM judge output the same distribution, i.e., it would need to reflect the same uncertainties.
>
> > Q2: I was interested in knowing how exactly Self-Reflect compares to other baselines such as LLM-as-a-judge in some controlled experiment similar to study 1 in Section 4.1. Can the authors comment on whether there’s a minimum number of samples to ensure the faithfulness of SelfReflect (i.e., if we have less than N sampling budget, we’re better off using other baseline metric?)
>
> Thank you for the suggestion! In Appendix B, we ablate both the number of answers in the reference distribution (in the judge context) and the number of answers that we use for our MC estimation (the number of masked-out tasks). We now added an experiment where we keep the reference distribution constant (and thus equal to the context provided to the baselines), and only vary the number of answers we calculate the masked-out tasks over, from 1 answer to 50 answers (our default). We report results for Qwen 2.5 7B on Natural Questions and all baselines below and add them to the paper in Appendix B.2. The performance is kept up from n=50 down to n=5, allowing to save compute budget. At n=2 and n=1, performance starts to decrease slightly, especially on harder cases like good vs almost-good and percentage vs or-concatenated, but stays above that of the baselines. Note that this was not the case in Appendix B; when reducing the answers _in the reference distribution / context_ down to 1 the metric becomes unfaithful by design (it is better to report the greedy answer). Reducing only the number of samples in the masked-out tasks is thus a better way to save on compute budget while staying faithful, thank you for suggesting the experiment!

---

> ### Author Response · Authors · 2025-11-21
>
> | Metric             | Good summaries vs bad summaries | Good vs almost-good | Detailed vs truncated | Verbalized uncertainty vs only majority answer | Verbalized vs or-concatenated | Percentage vs or-concatenated |
> |--------------------|---------------------------------|----------------------|------------------------|-----------------------------------------------|--------------------------------|--------------------------------|
> | Summarization      | 97.40% ± 0.99%                   | 38.70% ± 3.02%       | 53.55% ± 7.85%         | 11.57% ± 5.70%                                 | 57.02% ± 8.82%                 | 65.29% ± 8.48%                 |
> | LM Judge           | 98.33% ± 0.46%                   | 47.32% ± 1.91%       | 59.92% ± 5.93%         | 19.37% ± 5.60%                                 | 34.55% ± 6.74%                 | 35.08% ± 6.77%                 |
> | Opt. Transport     | 80.16% ± 1.43%                   | 60.78% ± 1.87%       | 39.69% ± 5.92%         | 48.69% ± 7.09%                                 | 52.88% ± 7.08%                 | 69.11% ± 6.55%                 |
> | Embedding          | 96.50% ± 0.66%                   | 65.49% ± 1.82%       | 65.65% ± 5.75%         | 10.99% ± 4.44%                                 | 43.98% ± 7.04%                 | 36.65% ± 6.83%                 |
> | SelfReflect M=1    | 99.00% ± 0.62%                   | 94.23% ± 1.48%       | 96.13% ± 3.04%         | 84.30% ± 6.48%                                 | 71.07% ± 8.08%                 | 78.51% ± 7.32%                 |
> | SelfReflect M=2    | 99.60% ± 0.39%                   | 96.12% ± 1.23%       | 98.06% ± 2.17%         | 90.91% ± 5.12%                                 | 77.69% ± 7.42%                 | 73.55% ± 7.86%                 |
> | SelfReflect M=5    | 99.70% ± 0.34%                   | 97.90% ± 0.91%       | 98.71% ± 1.78%         | 90.91% ± 5.12%                                 | 71.90% ± 8.01%                 | 80.99% ± 6.99%                 |
> | SelfReflect M=10   | 99.80% ± 0.28%                   | 98.22% ± 0.84%       | 98.06% ± 2.17%         | 92.56% ± 4.68%                                 | 76.03% ± 7.61%                 | 82.64% ± 6.75%                 |
> | SelfReflect M=20   | 99.80% ± 0.28%                   | 98.64% ± 0.74%       | 98.06% ± 2.17%         | 95.04% ± 3.87%                                 | 71.07% ± 8.08%                 | 84.30% ± 6.48%                 |
> | SelfReflect M=50   | 99.90% ± 0.28%                   | 98.74% ± 0.71%       | 98.06% ± 2.17%         | 95.04% ± 3.87%                                 | 74.38% ± 7.78%                 | 83.47% ± 6.62%                 |
>
> > Q3: Concerning the methodology, I’d like to clarify how is the b sentence selected? For example, in Figure 3, is b selected as the most frequent generation, or does SelfReflect eventually iterate over different sampled sequences ?
>
> We sample the M=50 sentences (or generally, answers, might have multiple sentences) for the masked out tasks B from the model at temperature 1 and iterate over each of them, and each word in them (except stopwords). We sketched this in lines 203-207, but have now added the above explanation, see the text highlighted in blue in the revised version.
>
> > Q4: Line 293 mentions “fine-grained quality differences” when discussing the superior behavior of the SelfReflect metric on the Good vs Almost Good setting. I could not find the details of how almost good summaries are generated. How do you ensure that these summaries differ by only fine-grained differences and what kind of differences are these? Was there any human validation performed to ensure this is the case?
>
> We generate them by prompting Gemini 2.0 Flash to alter the good summary by leaving out details or options, we report the exact prompt in Appendix G, page 34. We found qualitatively that this works as intended: The model either removes some answer possibilities that the good summary mentions or some additional details. To support this with a quantitative analysis, we also reran the experiment from Table 1 by letting humans write good and almost-good summaries. This is reported in Table 8 in Appendix G, and the results are equivalent to when using Gemini-generated summaries.
>
> > Q5: Line 298 states “matched only twice by its own SR-P(True) ablation”. Should it be “three times” instead of “twice”, since performance values between the SelfReflect and SR-P(True) ablation are also overlapping in the “Verbalized vs or-concatenated” setting?
>
> We agree, we have updated this in the revised version.

---

> > ### Author Response · Authors · 2025-11-21
> >
> > > Q6: Could the authors motivate the selection of different models across controlled experiments in Section 4?
> >
> > We started experiments in Section 4 with Qwen 2.5 7B at the time of conception of the project, since it was one of the best open-weight LLMs available. We then decided to diversify and test the robustness of our results on other LLMs, and aimed to have a range of architectures and providers, thus using Google’s Gemma 3 12B and Microsoft’s Phi-4 14B, and Meta’s Llama 3.1 8B. Some experiments have a cross-test over all combinations of models (such as in Appendix D). In Table 1, we instead tested on multiple datasets and in Table 3, conducted a human user study; in these cases including all model combinations would have been excessive in cost.
> >
> > > Q7: Section 5 includes experiments with numerous prompting and fine-tuning strategies. I was wondering if an iterative self-critiquing approach would improve upon simple prompting or “sample and summarize”?
> >
> > Thanks for the suggestion! We have experimented with multi-step refinements of summaries, where we first ask the model to give a summary and then prompt it again to see if there is any other options. The results were similar to CoT, with the model listing options, but those options deviating from what it is actually uncertain about. Hence, the SelfReflect score also did not improve. We have not run these experiments on all model and dataset combinations yet, which is why we did not include them in the paper. If you are interested, we can run this and include the results.
> >
> > > Q8: In Section 5.2 the authors suggest that despite fine-tuning and providing examples of descriptive summaries to the models, they seem to be unable to perform the task and that “model memorizes individual summaries rather than learning a general mechanism”. Could this be due to data quality issues (since the data is generated synthetically) and/or hyperparameter tuning?
> >
> > We do not believe so. After submission, we continued our RL experiments, with additional hyperparameters (Appendix M), by using a 10x larger (still synthetic) train dataset, and by training on human-written summaries from Appendix G. We found that the behavior stays the same. Even on the best hyperparameters, SFT training leads to slow improvements on the train dataset that generalize neither to the (same-dataset) eval set nor to the OOD data. We are thus convinced that at this point, naive fine-tuning only leads to learning the specific examples in and out and does not train the underlying (more complex) mechanism of thinking about the possible solutions and then summarizing them in the LLM. We believe that this “algorithm” might need to be made more explicit for the LLM to learn, similar to how it is in Sample & Summarize.
> >
> > > Writing style
> >
> > Thank you for the suggestions! We now link the Krippendorf Alpha implementation we used (https://github.com/grrrr/krippendorff-alpha/tree/master) and fixed the typos in the revised version.
> >
> > > Clarity: What's the intuition behind the reference metric in Section 4.2?
> >
> > We agree that this may not have been clear in the submitted version. We changed it to the following in the revised version (highlighted in blue): Since MMLU is a multiple-choice dataset, we can sample the LLM multiple times to obtain a simple categorical distribution. We then create summaries that either talk about this distribution (``The answer is most likely C (54% sure), but it could also be B (32% sure) or A (14% sure).'') or that mention the most likely answer only, are overconfident, or give random percentages. This gives a range of different-quality summaries that the benchmark metrics have to tell apart. The categorical setup of MMLU also allows to calculate a reference benchmark metric, namely the Wasserstein distance between the percentages mentioned in the summary and that of the real distribution. This lets us test if the SelfReflect metric and the other baselines agrees with the true distance in this special case. Specifically, we report the rank correlation between them and the reference metric.
> >
> > **We would like to thank you again for your time committed to reviewing.** Let us know if you have further questions or suggestions.

---

> > > ### Comment · Reviewer_j1CR · 2025-11-27
> > >
> > > Thank you for all the effort and time put into this rebuttal, including the new ablation experiments in Appendix B and additional clarifications. I think it’s a strong paper and I’m overall satisfied with their responses. I’ll keep my score as is.
> > >
> > > Note: I was going over Appendix B.2 and there is a small typo: “M can be reduces down” → “ M can be reduced down”.

---

> ### Author Response · Authors · 2025-12-02
>
> Thank you for your response and your strong score! We've fixed the typo in the revised version. Also, our experiment results for your final question are in:
>
> > Q7: Section 5 includes experiments with numerous prompting and fine-tuning strategies. I was wondering if an iterative self-critiquing approach would improve upon simple prompting or “sample and summarize”?
>
> Based on the reviewer’s suggestion, we inspected whether self-critique could help improve the performance of summaries. To this end, we let the model first generate a summary based on the Sample & summarize (N=10) strategy, and then give it the chance to critique and improve on its summary. While this self-critique approach does not consistently improve the SelfReflect score, it does give some marginal improvements (2-3 points) on the larger Gemma models, making this a potentially interesting approach to pursue further in future work. We have included the results in Appendix P of the manuscript.
>
> | Model                           | Sample & summarize N=10 | Self-Critique N=10 |
> |---------------------------------|-------------------------|---------------------|
> | Qwen2.5 0.5B Instruct           | 96                      | 97 (+1)             |
> | Qwen2.5 1.5B Instruct           | 87                      | 87 (-0)             |
> | Qwen2.5 3B Instruct             | 91                      | 92 (+1)             |
> | Qwen2.5 7B Instruct             | 91                      | 90 (-1)             |
> | Qwen2.5 14B Instruct            | 86                      | 85 (-1)             |
> | Qwen2.5 32B Instruct            | 91                      | 91 (-0)             |
> | Qwen2.5 72B Instruct            | 85                      | 84 (-1)             |
> | Phi 4                           | 85                      | 84 (-1)             |
> | Ministral 8B Instruct 2410      | 101                     | 100 (-1)            |
> | Llama 3.1 70B Instruct          | 87                      | 87 (-0)             |
> | Llama 3.3 70B Instruct          | 89                      | 88 (-1)             |
> | Llama 4 Scout 17B 16e Instruct  | 88                      | 86 (-2)             |
> | Gemma 3 1B Instruct             | 117                     | 117 (-0)            |
> | Gemma 3 4B Instruct             | 101                     | 102 (+1)            |
> | Gemma 3 12B Instruct            | 102                     | 99 (-3)             |
> | Gemma 3 27B Instruct            | 97                      | 95 (-2)             |

---

### Author Response · Authors · 2025-12-02

Dear Reviewers and Area Chairs,

We would like to thank you again for the time you spent on reviewing our work. We are happy to see the enthusiasm of the reviewers about the new avenue for communicating uncertainty that we envision for LLMs, namely to reflect on and summarize all possible outputs to a query when a model is uncertain.

To give a quick summary of the rebuttal period, here are the updates we have integrated into the revised version of the manuscript (all marked in blue in the uploaded revision):
* We compare the SeflReflect metric to an additional baseline metric, InfoSumm, which it outperforms similar to the related SR-PMI metric we benchmarked against.
* We analyze model behavior on an additional dataset, HotpotQA, with a retrieval-augmented generation setup. It behaves similar, but with a lower variance in answers, to our remaining datasets.
* We add a setup where almost-good summaries are generated in a one-step process to ensure robustness of our findings.
* We investigate SelfReflect’s estimation efficiency on lower numbers of masked-out tasks, finding that we can reduce the MC-estimate down to about 5 samples.
* We study the behavior of Sample & Summarize on high numbers of samples.
* We add qualitative examples of summaries and their corresponding SelfReflect scores.
* We add a self-critiquing version of sample-&-summarize.
* With the additional page, we will extend our explanations of some experiment setups and the interesting discussions on user trust and clustering works.

---

### Meta-Review · Area_Chair_GoMy · 2025-12-06

**Summary:**

This paper studies how large language models should communicate uncertainty by summarizing their full internal distribution over possible answers, rather than providing only a single response with a confidence score. The authors propose SelfReflect, an information-theoretic metric that evaluates how faithfully a generated summary represents the model’s true answer distribution, measured using masked-token prediction and Wasserstein distance. They validate the metric on synthetic data and then apply it to multiple models and decoding strategies. Experimental results show that current LLMs struggle to accurately articulate their internal uncertainty, highlighting challenges and future opportunities for uncertainty communication in natural language generation.

The reviewers agree the importance of the paper and the concerns were clearly addressed. There were one concern raised by a reviewer whether the topics of uncertainty quantification is important or not. Also, the following question "" seems to be valid to me. The response by the authors was Okay, but, it would be good to discuss this point further in the final version. Overall, I find the paper is interesting and good to be publish at ICLR;  I also vote for acceptance.

The reviewers agree that this paper addresses an important problem, and most concerns were clearly resolved in the rebuttal. One reviewer questioned the importance of uncertainty quantification and raised the valid point: “Do we even have a promising metric to reflect LLM’s internal probability distribution yet? If not, the topic of investigating whether a LLM can accurately express its internal probability distribution may not be reliable.” The authors’ response was acceptable, but it would be beneficial to further discuss this issue in the final version. Overall, I find the paper interesting and suitable for publication at ICLR, and I recommend acceptance.

**Reviewer Concerns:**

There were several requests for technical details and clarifications from Reviewer j1CR and Reviewer 2DwF, such as the comparison between Self-Reflect and other baselines like LLM-as-a-judge. The authors provided additional experiments that largely addressed these concerns.

Reviewer 2owx appears to be less experienced, but raised a valid point regarding the importance of the research topic itself: “Do we even have a promising metric to reflect LLM’s internal probability distribution yet? If not, the topic of investigating whether a LLM can accurately express its internal probability distribution may not be reliable.” The authors responded that they assume such a distribution exists, but this may not fully resolve the question. Nevertheless, given that theoretical understanding of LLMs is still limited, the empirical contributions of this work remain valuable. This issue could be sufficiently addressed by adding further discussion in the camera-ready version.

**Reviewer Scores:**

Most reviewers indicated that they would raise their scores, while one reviewer did not or could not respond during the discussion phase. Based on the discussion, I will keep my original score. However, I believe the paper has more merits than drawbacks overall.

---

### Decision · Program_Chairs · 2026-01-26

Accept (Poster)